# SOReL and TOReL: Two Methods for Fully Offline Reinforcement Learning

## Abstract

Sample efficiency remains a major obstacle for real world adoption of reinforcement learning (RL): success has been limited to settings where simulators provide access to essentially unlimited environment interactions, which in reality are typically costly or dangerous to obtain. Offline RL in principle offers a solution by exploiting offline data to learn a near-optimal policy before deployment. In practice, however, current offline RL methods rely on extensive online interactions for hyperparameter tuning, and have no reliable bound on their initial online performance. To address these two issues, we introduce two algorithms. Firstly, SOReL: an algorithm for **s**afe **o**ffline **re**inforcement **l**earning. *Using only offline data* our Bayesian approach infers a posterior over environment dynamics to obtain a reliable estimate of the online performance via the posterior predictive uncertainty. Crucially, all hyperparameters are also tuned fully offline. Secondly, we introduce TOReL: a **t**uning for **o**ffline **re**inforcement **l**earning algorithm that extends our information rate based offline hyperparameter tuning methods to general offline RL approaches. Our empirical evaluation confirms SOReL's ability to accurately estimate regret in the Bayesian setting whilst TOReL's offline hyperparameter tuning achieves competitive performance with the *best online hyperparameter tuning* methods *using only offline data*. Thus, SOReL and TOReL make a significant step towards safe and reliable offline RL, unlocking the potential for RL in the real world. Our implementations are publicly available: ANONYMISED.

## 1 Introduction

Offline RL (Lange et al., 2012; Levine et al., 2020; Murphy, 2024) promises to unlock the potential for agents to act autonomously, successfully and safely from the moment they are deployed into an environment. However, existing offline RL methods (Tarasov et al., 2023; Kostrikov et al., 2021; Kidambi et al., 2020; Yu et al., 2021) are yet to fulfil this promise due to two key issues that have been largely ignored by the community. **Issue I: there are currently no *offline* metrics to tune hyperparameters or choose between approaches**. Existing methods rely on *online* samples to carry out the extensive hyperparameter tuning required to achieve high performance (Zhang et al., 2021; Jackson et al., 2025). As we sketch in Fig. 1a, this results in cycles of training offline, deployment, failure online, further hyperparameter tuning and/or model selection, re-training offline and redeployment until the online performance of the agent is acceptable. Without a method for offline tuning, existing methods suffer from *high online sample complexity*, which is precisely the problem offline RL intends to solve.

For many problem settings, we also need reliable online performance guarantees *before* the agent is deployed online. Precisely, **issue II: current methods offer no reliable *offline* method to approximate true *online* regret**, i.e. the difference between expected returns of an optimal policy and a policy trained using offline data. This is concerning from an AI safety perspective, as without a reliable regret bound, we cannot deploy agents into the real world where agent failure presents a serious hazard to human life; for these settings it is essential that agents are deployed with near-zero regret. For less sensitive domains, users still need some guarantee of optimality on deployment; there will often be a clear cost associated with deviation from optimal policy in terms of regret, for example in settings where the degree to which a product can fail will result in financial loss that can be determined before deployment using regret and kept in a tolerable range.

To tackle these two essential issues, we develop a Bayesian framework where the posterior (conditioned on the offline data) is used as a *prior* for a Bayesian RL problem (Martin, 1967; Duff, 2002).

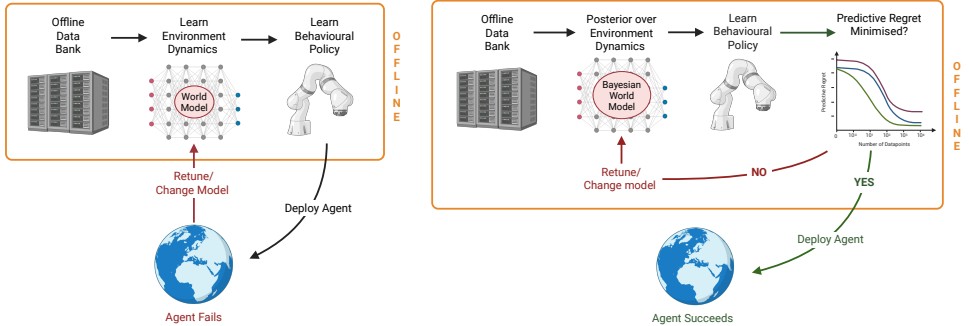

(a) Existing Model-Based Offline RL         (b) SOReL and TOReL (Our Approach)

Figure 1: Existing model-based offline approaches rely on online interactions for hyperparameter tuning and verifying accurate model learning before they can achieve good performance, leading to poor online sample efficiency. In contrast, in SOReL/TOReL, model tuning and world model learning is carried out fully offline using the predictive regret as a tuning signal. Only then is the agent trained and deployed.

Our analysis reveals that the regret of the corresponding Bayes-optimal policy is controlled by the posterior information loss (PIL) - that is the expected posterior KL divergence between the model and true dynamic. The change in PIL, known as the *information rate*, measures how much information the model has gained from an incremental amount of offline data. Crucially, the PIL can be estimated and tracked during offline training, allowing us to monitor performance and tune hyperparameters completely offline.

For our first method, we develop SOReL, a theoretically grounded framework for model-based **s**afe **o**ffline **re**inforcement **l**earning which resolves both key issues. By using offline data to infer a posterior over environment dynamics, we approximate regret using the predictive variance and median of policy rollouts *prior to deployment*, directly tackling **issue II**. Moreover, both the predictive regret and the PIL can be used as a signal to tune hyperparameters offline, thereby tackling **issue I** (see Fig. 1b). Only then is the trained agent deployed safely, making SOReL (to the authors' knowledge) the first fully offline RL approach with reliable performance guarantees once deployed. Our experiments support this claim empirically, showing that in the standard offline RL MuJoCo control tasks (Yu et al., 2020; Kidambi et al., 2020; Ball et al., 2021; Lu et al., 2022b; Sun et al., 2023; Sims et al., 2024), SOReL's *offline* regret approximation accurately tracks the true regret once deployed *online*.

As SOReL is a general Bayesian approach, existing methods may outperform it in specific domains as many have been designed to exploit heuristics tailored to specific datasets and tasks. To address this, we also extend SOReL's offline hyperparameter tuning methods to address **issue I** for existing model-free and model-based offline RL approaches (Yu et al., 2020; Kidambi et al., 2020; Ball et al., 2021; Lu et al., 2022b; Sun et al., 2023; Tarasov et al., 2023; Sims et al., 2024) when accurate regret estimation isn't required, greatly improving their sample efficiency. Using this insight, we develop TOReL, an algorithm for **t**uning **o**ffline **re**inforcement **l**earning that uses the PIL signal and tracks a *regret metric correlated to the true regret* for tuning. To test our method, we apply TOReL to IQL (Kostrikov et al., 2021), ReBRAC (Tarasov et al., 2023), MOPO (Yu et al., 2020) and MOReL (Kidambi et al., 2020) to carry out hyperparameter tuning in Adroit gymnasium and the standard offline RL MuJoCo control tasks. Using only *offline* data, TOReL achieves similar performance to existing methods that carry out *online* hyperparameter tuning. Notably, when combined with ReBRAC, TOReL consistently finds a hyperparameter combination with near-zero regret, outperforming all hyperparameters for all other algorithms. When comparing TOReL's offline hyperparameter tuning to a recent online UCB approach (Jackson et al., 2025), we see that UCB typically requires about a dataset's worth of *online* samples to match TOReL's performance. We summarise our key contributions:

I  In Section 4, we develop a Bayesian framework for model-based offline RL;

II  In Section 4.3 we carry out a regret analysis for our framework, demonstrating regret is controlled by the PIL and provide a strong frequentist justification for our Bayesian approach;

III  In Section 5.1 we develop SOReL, a method for approximating true regret using predictive uncertainty, which can achieve a desired and safe level of true regret once deployed;

IV  In Section 5.2 we introduce TOReL, adapting SOReL's offline hyperparameter/model tuning approach to general offline model-based and model-free RL;

V  In Section 6 we empirically confirm that TOReL addresses **issue I** for existing methods and SOReL's abilty to address **issues I and II** for more conservative applications.

## 2 PRELIMINARIES

Let $X$ be a $\mathcal{X} \subseteq \mathbb{R}^n$-valued random variable. We denote a distribution as $P_X$ with density (if it exists) as $p(x)$. We denote the set of all distributions over $\mathcal{X}$ as $\mathcal{P}(\mathcal{X})$. We introduce the notation $\mathcal{G}(p)$ to represent the geometric distribution and $\mathcal{AG}(p)$ to represent the arithmetico-geometric distribution, with probability mass functions: $P_{\mathcal{G}}(i) \coloneqq (1-p)p^i$ and $P_{\mathcal{AG}}(i) \coloneqq (1-p)^2 p^i (i+1)$ respectively, for $i \in \mathbb{N}_0^+$ and parameter $p \in [0, 1)$. We denote the uniform distribution over $\{0, 1, \dots i\}$ as $\mathcal{U}_i$ and the multivariate normal distribution with mean vector $\mu$ and covariance matrix $\Sigma$ as $\mathcal{N}(\mu, \Sigma)$.

### 2.1 OFFLINE REINFORCEMENT LEARNING

For our offline RL setting, an agent is tasked with solving the learning problem in an infinite-horizon, discounted Markov decision process (Bellman, 1956; 1958; Sutton and Barto, 2018; Puterman, 1994; Szepesvári, 2010): $\mathcal{M}^\star \coloneqq \langle \mathcal{S}, \mathcal{A}, P_0, P_S^\star(s, a), P_R^\star(s, a), \gamma \rangle$, with state space $\mathcal{S}$, action space $\mathcal{A}$ and discount factor $\gamma$. At time $t = 0$, an agent starts in an initial state allocated according the the initial state distribution: $s_0 \sim P_0$. At every timestep $t$, an agent in state $s_t$ takes an action according to a policy $a_t \sim \pi(h_t)^1$, receives a scalar reward $r_t \sim P_R^\star(s_t, a_t)$ and transitions to a new state $s_{t+1} \sim P_S^\star(s_t, a_t)$ where $h_t \coloneqq \{s_0, a_0, r_0, s_1, a_1, r_1, \dots a_{t-1}, r_{t-1}, s_t\} \in \mathcal{H}_t$ is the observed a history of interactions with the environment. Here $\mathcal{H}_t \coloneqq \mathcal{S} \times (\mathcal{A} \times \mathbb{R} \times \mathcal{S})^{\times t}$ denotes the corresponding product space. We assume rewards are bounded with $r_t \in [r_{\min}, r_{\max}] \subset \mathbb{R}$ where $r_{\min}$ and $r_{\max}$ denote the minimum and maximum reward values respectively. For convenience, we often write the joint state transition-reward distribution as $P_{R,S}^\star(s, a)$. We denote the distribution over history $h_t$ as $P_{t,\pi}^\star$. The goal of an agent is to learn an optimal policy $\pi^\star \in \Pi^\star$ where $\Pi^\star \coloneqq \arg\max_\pi J^\pi(\mathcal{M}^\star)$ is the set of policies that maximise the expected discounted return $J^\pi(\mathcal{M}^\star) \coloneqq \mathbb{E}_{h_\infty \sim P_{\infty,\pi}^\star} \left[ \sum_{i=0}^\infty \gamma^i r_i \right]$. It suffices to consider only optimal policies that condition only on the most recent state (i.e. $\pi^\star(s_t)$) as, in a fully observable MDP, any optimal history-conditioned policy will never take an action that cannot be taken by an optimal policy that conditions only on most recent state.

In the learning setting, the true state transition distribution $P_S^\star(s, a)$ and reward distribution $P_R^\star(s, a)$ are assumed unknown a priori. Once deployed, the agent is faced with the exploration/exploitation dilemma in that it must balance exploring to learn about the unknown environment dynamics with exploiting. In offline RL (Lange et al., 2012; Levine et al., 2020; Murphy, 2024), an agent has access to a dataset of histories of various lengths collected from the true environment. The policies used to collect the data may vary and not be optimal. In the zero-shot model-based offline RL setting (Jackson et al., 2025), the dataset is used to learn the unknown environment dynamics from which a policy is trained prior to any interaction with the environment. The agent is then deployed at test time $t = 0$ and its performance evaluated. The goal of offline RL is to take advantage of offline data so that the deployed policy will be near-optimal from the outset.

## 3 RELATED WORK

Developing reliable model-based offline RL algorithms remains an open challenge for several reasons; most existing methods are not fully offline, requiring extensive online interactions for tuning. Only limited attention has been given to solving key **issues I and II** introduced in Section 1, which is the focus of this work: Paine et al. (2020) introduce a method for estimating online value and partial hyperparameter tuning of offline model-free algorithms, however their method neither approximates regret, nor is an accurate proxy for true value, resulting in significant overestimation in most domains. As noted by Smith et al. (2023), their approach relies on offline policy evaluation, which is a challenging and provably difficult problem (Wang et al., 2020) whose hyperparameters require tuning online. Moreover, as noted by Jackson et al. (2025), their framework is limited to behavioural cloning and two model-free critic-based methods that have since been outperformed by modern algorithms. Smith et al. (2023) introduce a method for offline hyperparameter tuning, but are limited to the model-free imitation learning setting and offer no regret estimation. Finally, Wang et al. (2022) introduce a method for offline hyperparameter tuning to pre-select hyperparameters for online methods, but do not learn optimal policies offline or provide regret approximation. In contrast to all of these approaches, to the authors' knowledge, our method is the first offline RL method to reliably approximate regret and carry out *all hyperparameter* tuning for general methods using *only offline data*. Finally, understanding of offline RL from a Bayesian perspective is limited. To the authors' knowledge, only Chen et al. (2024) have framed solving offline model-based RL as solving a

---

[1] Policies condition on history as we work within a Bayesian paradigm, using methods such as RNN-PPO (Schulman et al., 2017)

BAMDP, however no regret analysis of the Bayes-optimal policy is carried out, a continuous BAMCP (Guez et al., 2014) approximation is used to learn behavioural policies and the algorithm still suffers from a lack of regret approximation, hence relying on online data for tuning.

In addition, the performance of offline RL approaches is particularly dependent on the ability to accurately model transition dynamics as errors in a dynamics model can compound over several timesteps for the long-horizon problems encountered in RL (see also our analysis in Section 4.3) and many datasets used to benchmark methods contain missing datapoints in critical regions of state-action space, which poses a generalisation challenge. We note that both these problems are *orthogonal* to solving key **issues I and II**, which we focus on in this work. Most existing offline RL methods focus on tackling the missing data problem by introducing a form of reward pessimism based on the model uncertainty (Yu et al., 2020; Kidambi et al., 2020; Kumar et al., 2020; Yu et al., 2021; Fujimoto and Gu, 2021; Kostrikov et al., 2021; An et al., 2021; Ball et al., 2021; Lu et al., 2022b; Sun et al., 2023; Tarasov et al., 2023; Sims et al., 2024). Unifloral (Jackson et al., 2025) is a recent framework that unites these offline RL approaches into a single algorithmic space with lightweight and high-performing implementations, as well as providing a clarifying benchmarking protocol. Our implementations and evaluation methods follow this framework.

## 4 BAYESIAN OFFLINE RL

We now introduce our Bayesian RL framework, which constitutes learning an (approximate) posterior from offline data before solving a Bayesian RL problem with the posterior acting as the prior. We provide an introductory primer on Bayesian RL in Section B and all proofs for all theorems can be found in Section D.

### 4.1 LEARNING A POSTERIOR WITH OFFLINE DATA

A Bayesian epistemology characterises the agent's uncertainty in the MDP through distributions over any unknown variable (Martin, 1967; Duff, 2002). We first specify a parametric model $p(r_t, s_{t+1}|s_t, a_t, \theta)$, $P_{R,S}(s_t, a_t, \theta)$, $\theta \in \Theta \subset \mathbb{R}^d$, over the unknown state transitions and reward distributions, with each $\theta \in \Theta \subseteq \mathbb{R}^d$ representing a hypothesis about the MDP $\mathcal{M}^\star$. As we show in Section 4.3, our results can easily be generalised to non-parametric methods like Gaussian process regression (Rasmussen and Williams, 2006; Wiener, 1923; Krige, 1951). A prior distribution over the parameter space $P_\Theta$ is specified, which represents the initial *a priori* belief in the true value of $P_{R,S}^\star(s, a)$ before the agent has observed any transitions.

We denote an offline dataset of $N$ state-action-state-reward transition observations as: $\mathcal{D}_N = \{(s_i, a_i, s_i', r_i)\}_{i=0}^{N-1}$, all collected from a single MDP $\mathcal{M}^\star$. Datapoints may be collected from several policies and non-Markovian sampling. Given the dataset $\mathcal{D}_N$, the prior $P_\Theta$ with density $p(\theta)$ is updated to posterior $P_\Theta(\mathcal{D}_N)$ with density $p(\theta|\mathcal{D}_N)$, using Bayes' rule:

$$p(\theta|\mathcal{D}_N) = \frac{p(\mathcal{D}_N|\theta)p(\theta)}{p(\mathcal{D}_N)} = \frac{\prod_{i=0}^{N-1} p(r_i, s_i'|s_i, a_i, \theta)p(\theta)}{\int_\Theta \prod_{i=0}^{N-1} p(r_i, s_i'|s_i, a_i, \theta)p(\theta)d\theta}. \tag{1}$$

The posterior represents the agent's belief in the unknown environment dynamics once $\mathcal{D}_N$ has been observed. We now detail how a Bayes-optimal policy is learned using the posterior as the initial belief in the environment dynamics.

### 4.2 LEARNING A BAYES-OPTIMAL POLICY

It is well known that solving a Bayesian RL problem exactly is intractable for all but the simplest models (Martin, 1967; Duff, 2002; Guez et al., 2012; 2013; Zintgraf et al., 2020; Fellows et al., 2024). Inferring the posterior in Eq. (1) is typically infeasible for dynamics models of interest (for example, nonlinear Gaussian world models). This is because there is no analytic solution for the posterior density and the cost of carrying out integration required to evaluate the evidence $p(\mathcal{D}_N)$ grows exponentially in parameter dimensions $d$. Fortunately, there exist tractable methods to learn an approximate posterior $\hat{P}_\Theta(\mathcal{D}_N) \approx P_\Theta(\mathcal{D}_N)$; in this paper we use randomised priors (Osband and Van Roy, 2017; Osband et al., 2018; Ciosek et al., 2020) (RP) and provide details in Section F.2. In addition, a planning problem must be solved for every conceivable history that an agent could encounter. In our offline RL setting, we ease intractability by replacing the prior $P_\Theta$ with a highly informative posterior $P_\Theta(\mathcal{D}_N)$, significantly reducing the hypothesis space in the Bayesian RL problem.

Let $P_{\infty,\pi}(\theta)$ denote the corresponding model distribution over $h_\infty$ for policy $\pi(h_t)$. To obtain an (approximately) Bayes-optimal policy, we use the Bayesian RL objective in the meta-

learning form (Zintgraf et al., 2020; Beck et al., 2024) (i.e. as an expectation using $\hat{P}_\Theta(\mathcal{D}_N)$) so that a simple RL$^2$(Duan et al., 1987) style algorithm can be applied: $J^\pi_{\text{Bayes}}(\hat{P}_\Theta(\mathcal{D}_N)) :=$ $\mathbb{E}_{\theta \sim \hat{P}_\Theta(\mathcal{D}_N)}\left[\mathbb{E}_{h_\infty \sim P_{\infty,\pi}(\theta)}\left[\sum_{i=0}^\infty \gamma^i r_i\right]\right]$. Solving the Bayesian RL objective is known as solving a Bayes-adaptive MDP (BAMDP) (Duff, 2002). We optimise the objective $J^\pi_{\text{Bayes}}(\hat{P}_\Theta(\mathcal{D}_N))$ by sampling a hypothesis environment from the approximate posterior $\theta \sim \hat{P}_\Theta(\mathcal{D}_N)$ then rolling out the policy in the sampled environment dynamics. The Bayes-optimal policy $\pi^\star_{\text{Bayes}} \in \arg\max_\pi J^\pi_{\text{Bayes}}(\hat{P}_\Theta(\mathcal{D}_N))$ is learned using RNN-PPO (Schulman et al., 2017) as a BAMDP solver on the rollouts. Complete implementation details can be found in Section F.

In addition to having excellent exploration/exploitation properties, a Bayesian approach affords access to epistemic uncertainty in the returns via the variance of predictive rollouts. Uncertainty estimation is essential for tackling our two keys **issues I and II**; firstly, as we show in Section 6.2, the predictive variance and predictive median of policy returns can be used to estimate the true regret at test time. Secondly, monitoring the decay of predictive variance and regret is a powerful tool for diagnosing issues with the BAMPD planner offline. Finally, we remark that a Bayesian approach is relatively simple compared to existing model-based approaches in Section 3 as it does not rely on hand-crafted heuristics tailored to specific problem settings and *scales as well as existing state-of-the-art offline RL methods* (Jackson et al., 2025) which also rely on ensembling methods for uncertainty quantification.

### 4.3 FREQUENTIST JUSTIFICATION OF BAYESIAN OFFLINE RL

We now carry out a frequentist asymptotic regret analysis for Bayesian offline RL. For finite $N$, we can measure how far the performance of the Bayes optimal policy is from an optimal policy using the *true regret*, which is the difference between the expected return $J^{\pi^\star_{\text{Bayes}}}(\mathcal{M}^\star, \mathcal{D}_N)$ of the Bayes-optimal policy $\pi^\star_{\text{Bayes}}$ given a posterior $P_\Theta(\mathcal{D}_N)$, all in the true MDP $\mathcal{M}^\star$: $\text{Regret}(\mathcal{M}^\star, \mathcal{D}_N) :=$ $J^{\pi^\star}(\mathcal{M}^\star) - J^{\pi^\star_{\text{Bayes}}}(\mathcal{M}^\star, \mathcal{D}_N)$. The goal of this analysis twofold; firstly, we show that the rate of regret decreases for a Bayes-optimal policy is characterised by an easy to estimate quantity known as the *posterior information loss* (PIL). Secondly, our goal is show (for the first time) that the asymptotic regret of offline Bayesian RL converges at the $\mathcal{O}\left(1/\sqrt{N}\right)$ rate found in prior work (Yu et al., 2020; Kidambi et al., 2020), which is known to be the optimal asymptotic convergence rate for nonlinear models in general MDPs for offline RL (Agarwal et al., 2022). This provides a strong frequentist justification for our Bayesian offline RL framework. Our key result shows that regret can be bounded using the PIL, defined as: $\mathcal{I}^\pi_N := \mathbb{E}_{\theta \sim P_\Theta(\mathcal{D}_N)}\left[\mathbb{E}_{s,a \sim \rho^\star_\pi}\left[\text{KL}\left(P^\star_{R,S}(s,a)\|P_{R,S}(s,a,\theta)\right)\right]\right]$. Here $\rho^\star_\pi := \mathbb{E}_{i \sim \mathcal{A}\mathcal{G}(\gamma)}\left[\mathbb{E}_{j \sim \mathcal{U}_i}\left[P^\star_{j,\pi}\right]\right]$ is the arithmetico-geometric ergodic state-action distribution, which places mass over state-action pairs according to how much errors in the model influence the regret at each state. Regions of state-action space that require more timesteps to reach from initial states are weighted significantly less than those that are encountered earlier and more frequently, as state errors encountered early accumulate in each prediction from that timestep onwards. The PIL has an intuitive information-geometric interpretation which we discuss further in Section D.2.

**Theorem 1.** *Let $\mathcal{R}_{\max} := \frac{(r_{max} - r_{min})}{1-\gamma}$ denote the maximum possible regret for the MDP. Using the PIL: $\mathcal{I}^\pi_N$, the true regret is bounded as:* $\text{Regret}(\mathcal{M}^\star, \mathcal{D}_N) \leq 2\mathcal{R}_{\max} \cdot \sup_\pi \sqrt{1 - \exp\left(-\frac{\mathcal{I}^\pi_N}{1-\gamma}\right)}$

We observe from Theorem 1 that the rate at which regret decreases with $N$ is governed by the rate the PIL decreases, which is known as the *information rate*. This measures how much information the model has gained from an incremental amount of data. Fast information rates imply highly informative posteriors can be learned using minimal data as regret will decrease at least as fast. How fast the information rate is depends on the exact model specification, prior and underlying MDP. Formulating our bound in terms of the PIL ties the regret to the KL divergence over the reward-state model: $\text{KL}\left(P^\star_{R,S}(s,a)\|P_{R,S}(s,a,\mathcal{D}_N)\right)$. Not only is this mathematically more convenient, yielding a simpler bound, but the PIL is *easy to estimate in practice* meaning the information rate can be monitored offline as a proxy for downstream regret and to carry out hyperparameter tuning associated with the model and approximate inference method, partially resolving key **issue I**. Using Theorem 1, we can study the PIL $\mathcal{I}^\pi_N$ for different classes of models which allows us to understand how regret will evolve given the model choice. This also provides a frequentist justification for many Bayesian approaches. We now characterise the information rate for parametric models, which allows us to recover the optimal $\mathcal{O}\left(1/\sqrt{N}\right)$ regret convergence rate for Bayesian offline RL:

**Theorem 2.** *Let the data be drawn from the underlying true distribution $\mathcal{D}_N \sim P_{Data}^\star$. Under standard local asymptotic normality assumptions (see Assumption 1 in Section D.3), there exists some constant $0 < C < \infty$ such that for sufficiently large $N$: $\mathbb{E}_{\mathcal{D}_N \sim P_{Data}^\star}\left[\text{Regret}(\mathcal{M}^\star, \mathcal{D}_N)\right] \leq$*

$$2\mathcal{R}_{\max} \cdot \sqrt{1 - \exp\left(-\frac{Cd}{(1-\gamma)N}\right)} = \mathcal{O}\left(\frac{1}{\sqrt{(1-\gamma)N}}\right).$$

### 4.4 GAUSSIAN WORLD MODELS

Many methods specify Gaussian reward and state transition models of the form:

$$P_R(s, a, \theta) = \mathcal{N}(r_\theta(s, a), \sigma_r^2(s, a)), \quad P_S(s, a, \theta) = \mathcal{N}(s_\theta'(s, a), I\sigma_s^2(s, a)), \tag{2}$$

with isotropic variance characterised by $\sigma_r^2$ and $\sigma_s^2$, mean reward function $r_\theta(s, a)$ and mean state transition function $s_\theta'(s, a)$. Using a Gaussian world model, we find the PIL takes a convenient and intuitive form. Let $r(s, a, \mathcal{D}_N) := \mathbb{E}_{\theta \sim P_\Theta(\mathcal{D}_N)}\left[r_\theta(s, a)\right]$ and $s'(s, a, \mathcal{D}_N) := \mathbb{E}_{\theta \sim P_\Theta(\mathcal{D}_N)}\left[s_\theta(s, a)\right]$ denote the Bayesian mean reward and state transition functions and $r^\star(s, a)$ and $s^{\star'}(s, a)$ denote the true mean functions. We define the mean squared error between the true and Bayesian mean functions as:

$$\mathcal{E}(\mathcal{D}_N, \mathcal{M}^\star) := \mathbb{E}_{(s,a)\sim\rho_\pi^\star}\left[\frac{\|r(s, a, \mathcal{D}_N) - r^\star(s, a)\|_2^2}{2\sigma_r^2(s, a)} + \frac{\|s'(s, a, \mathcal{D}_N) - s^{\star'}(s, a)\|_2^2}{2\sigma_s^2(s, a)}\right], \tag{3}$$

and the predictive variance as:

$$\mathcal{V}(\mathcal{D}_N) := \mathbb{E}_{(s,a)\sim\rho_\pi^\star}\left[\mathbb{E}_{\theta \sim P_\Theta(\mathcal{D}_N)}\left[\frac{\|r(s, a, \mathcal{D}_N) - r_\theta(s, a)\|_2^2}{2\sigma_{r,}^2(s, a)} + \frac{\|s'(s, a, \mathcal{D}_N) - s_\theta'(s, a)\|_2^2}{2\sigma_s^2(s, a)}\right]\right]. \tag{4}$$

We now re-write the PIL for the Gaussian world model using these two terms:

**Proposition 1.** *Using the Gaussian world model in Eq. (2), it follows: $\mathcal{I}_N^\pi = \mathcal{E}(\mathcal{D}_N, \mathcal{M}^\star) + \mathcal{V}(\mathcal{D}_N)$.*

Proposition 1 shows that the PIL is governed by i) the MSE of the point estimate $\mathcal{E}(\mathcal{D}_N, \mathcal{M}^\star)$, which characterises how quickly the Bayesian mean function converges to the true function; and ii) the predictive variance $\mathcal{V}(\mathcal{D}_N)$, which characterises the epistemic uncertainty in the model. For frequentist methods using point estimates like the MLE, there is no characterisation of epistemic uncertainty, meaning $\mathcal{V}(\mathcal{D}_N) = 0$. The PIL can easily be estimated by estimating $\mathcal{E}(\mathcal{D}_N, \mathcal{M}^\star)$ using the empirical MSE with offline data and estimating $\mathcal{V}(\mathcal{D}_N)$ using posterior sampling.

## 5 FROM THEORY TO PRACTICE

Our frequentist analysis in Section 4.3 provides valuable intuition about how we might expect regret to change depending on the choice of model, however it cannot address key **issues I and II** from Section 1. This is because asymptotic results are only theoretical guarantees that apply with high probability in the limit of large data across a space of MDPs; their main use is a theoretical object to provide a way to compare asymptotic behaviour of algorithms rather than for obtaining reliable regret estimation or a being a metric to tune algorithms as they cannot actually be calculated. As a concrete example, the regret bound of MOReL (Kidambi et al., 2020) is: $\text{Regret}_N \leq C/\sqrt{N} + m$ which is impossible to calculate in the offline RL setting because 1) $C$ is not specified, it is just proved to exist; 2) the constant $m$ requires knowing: the hitting time of all unknown states in the original MDP of the optimal policy, the smallest nonzero elements of the initial state distribution and the smallest elements of state-transition distribution; and 3) the bound only holds for large enough $N$, which is unspecified. Even if these quantities were known, we would only have a bound and not an estimate of the true regret. For these reasons, tackling **issues I and II** is a challenging problem to solve.

### 5.1 SOReL

We now introduce SOReL in Algorithm 1, our algorithm for reliable regret estimation and offline hyperparameter tuning. In our SOReL framework, there are three sets of hyperparameters: $\phi_I$ the model (such as the architecture for a neural-network function approximator); $\phi_{II}$ the approximate inference method (such as the number of ensemble members for RP); and $\phi_{III}$ the BAMDP solver (the hyper-parameters of a Bayesian meta-learning algorithm like RNN-PPO). Sets $\phi_I$ and $\phi_{II}$ are tuned jointly to both minimise the PIL and ensure a roughly even split between the predictive variance and MSE loss terms. Set $\phi_{III}$ is then tuned to minimise approximate regret based on the now-fixed model and approximate posterior: for each combination of hyper-parameters, we learn a policy using the BAMDP solver, and choose the combination whose policy leads to the lowest approximate regret. $\mathcal{R}_{\text{Deploy}}$ denotes the desired level of regret of the deployed policy.

**Fixing Issue I - PIL Monitoring:** To tune sets $\phi_I$ and $\phi_{II}$, we monitor the change in PIL: $\mathcal{I}_N^\pi$. Our goal is to select hyperparameters that minimises the PIL whilst ensuring the the MSE term (c.f. Eq. (3)) closely matches the predictive variance term (c.f. Eq. (4)): $\mathcal{E}(\mathcal{D}_N, \mathcal{M}^\star) \approx \mathcal{V}(\mathcal{D}_N)$. Misalignment of predictive variance and MSE indicates either an overfitting/underfitting issue with model hyperparameters in set $\phi_I$ and/or an issue with uncertainty esti-

---

**Algorithm 1** SOReL($P_\Theta, \mathcal{D}_N, \mathcal{R}_{\text{Deploy}}$)

---

$R_N \leftarrow \hat{\mathcal{R}}_{\max}$
**while** $R_N > \mathcal{R}_{\text{Deploy}}$ **do**
    *Hyperparameter tuning:*
    $\phi_I, \phi_{II} \leftarrow \arg\min_{\phi_I, \phi_{II}} \text{PIL}(\phi_I, \phi_{II}, \mathcal{D}_N)$
        s.t. $\mathcal{E}(\mathcal{D}_N, \mathcal{M}^\star) \approx \mathcal{V}(\mathcal{D}_N)$
    $\phi_{III} \leftarrow \arg\min_{\phi_{III}} \text{ApproxRegret}(\phi_I, \phi_{II}, \phi_{III}, \mathcal{D}_N)$
    *Policy Learning and Regret Approximation:*
    $\pi_{\text{Bayes}}^\star \leftarrow \text{SolveBAMDP}(\phi_I, \phi_{II}, \phi_{III}, \mathcal{D}_N)$
    $R_N \leftarrow \text{ApproxRegret}(\phi_I, \phi_{II}, \phi_{III}, \mathcal{D}_N)$
**end while**
**return** $\pi_{\text{Bayes}}^\star$

---

mation due to approximate inference hyperparameters in set $\phi_{II}$. Moreover, under/overestimating uncertainty will lead to poor regret estimation, which is why we tune sets $\phi_I$ and $\phi_{II}$ first in Algorithm 1. Our empirical evaluations in Section 6.2 confirm that when $\mathcal{E}(\mathcal{D}_N, \mathcal{M}^\star) \approx \mathcal{V}(\mathcal{D}_N)$, the approximate regret aligns strongly with true regret.

**Fixing Issues I and II - Regret Approximation:** A simple approach for bounding regret would be to estimate the PIL from the offline data and then apply our theoretical upper bound in Theorem 1. This is likely to be too conservative for most applications as it protects against the worst case MDP that the agent could encounter. In particular, it is very sensitive to errors in the model, especially as $\gamma \to 1$, which is an artifact of model errors accumulating over all future timesteps in the regret analysis. Instead, we approximate the regret using the posterior predictive median:

$$\text{Regret}(\mathcal{M}^\star, \mathcal{D}_N) \approx \hat{R}_{\max} - \hat{\mathbb{M}}_{\theta \sim P_\Theta(\mathcal{D}_N), h_\infty \sim P_\infty^\pi(\theta)} [R(h_\infty)], \quad (5)$$

where $\hat{\mathbb{M}}_{\theta \sim P_\Theta(\mathcal{D}_N), h_\infty \sim P_\infty^\pi(\theta)} [R(h_\infty)]$ denotes the median predictive return based on sampling from the (approximate) posterior and rolling out the Bayes-optimal policy and $\hat{R}_{\max}$ is estimated from the maximum return in the offline dataset - full details and an overview of alternative metrics that can be derived from the posterior to approximate the regret with varying degrees of conservatism are found in Section C.2. We hypothesise that the sample median offers a good compromise: neither overly conservative nor overly susceptible to being skewed by a policy that performs well on only a subset of posterior samples. Our empirical evaluations support this hypothesis in Section 6.2. The posterior predictive median also allows us to tune hyperparameter set $\phi_{III}$, selecting hyperparameters to learn a policy that achieves the lowest approximate regret as shown in Algorithm 1.

## 5.2 TOReL

SOReL's offline hyperparameter tuning methods are directly applicable to general offline RL approaches, allowing us to address **issue I** for existing offline methods. We now adapt these methods to derive a general **t**uning for **o**ffline **re**inforcement **l**earning approach called TOReL, shown in Algorithm 2. A policy is learned offline us-

---

**Algorithm 2** TOReL($P_\Theta, \mathcal{D}_N$)

---

$\phi_I, \phi_{II} \leftarrow \arg\min_{\phi_I, \phi_{II}} \text{PIL}(\phi_I, \phi_{II}, \mathcal{D}_N)$
  $[\text{ s.t. } \mathcal{E}(\mathcal{D}_N, \mathcal{M}^\star) \approx \mathcal{V}(\mathcal{D}_N), \text{(model-based)}]$
$\phi_{III} \leftarrow \arg\min_{\phi_{III}} \text{RegretMetric}(\phi_I, \phi_{II}, \phi_{III}, \mathcal{D}_N)$
$\pi_{\text{TOReL}}^\star \leftarrow \begin{cases} \text{ORL}(\phi_{III}, \mathcal{D}_N), & \text{(model-free)} \\ \text{ORL}(\phi_I, \phi_{II}, \phi_{III}, \mathcal{D}_N), & \text{(model-based)} \end{cases}$
**return** $\pi_{\text{TOReL}}^\star$

---

ing a planning algorithm, denoted by ORL. There thus exists a corresponding set of hyperparameters associated with $\phi_{III}$ the offline planner. For model-based methods with uncertainty estimation like MOReL Kidambi et al. (2020) and MOPO Yu et al. (2020), we can exactly adapt SOREL's PIL tuning method to the parameters associated with: $\phi_I$ the dynamics model and $\phi_{II}$ uncertainty estimation. For all other methods, we introduce and learn a dynamics model and an approximate inference method like in SOReL and jointly tune the corresponding hyperparameters $\phi_I$ and $\phi_{II}$ to minimise the PIL without requiring the even split between the predictive variance and MSE loss terms. Since the policy learned with ORL is typically neither Bayes-Optimal nor robust to model uncertainty, we expect that applying SOReL's regret approximation method to more general methods in *TOReL will not yield an accurate estimate of the regret in terms of its absolute value*. Instead, we treat the approximate regret in Eq. (5) as a *regret metric* that is positively correlated with true regret, and use this to tune ORL parameters $\phi_{III}$. Our empirical evaluations in Section 6.1 support this hypothesis. We note that in model-free methods, the dynamics model and an approximate inference method are not used in policy learning, only to aid regret metric calculation.

## 6 EXPERIMENTS

The goal of experimental section is to confirm that our methods successfully solves key **issues I and II**. In Section 6.1, we confirm that TOReL resolves **issue I** for existing ORL algorithms, consistently identifying hyperparameters with a lower regret than the average regret of randomly chosen hyperparameters using *offline data alone*. In Section 6.2, we confirm that SOReL is the first approach that can resolve **issue II**, accurately approximating regret over a range of regret curves.

### 6.1 TOReL IS AN EFFECTIVE OFFLINE HYPERPARAMETER TUNER FOR ORL

For this experiment, our goal is to use TOReL to identify hyperparameters for ReBRAC Tarasov et al. (2023) and IQL Kostrikov et al. (2021) (two model-free algorithms), and MOPO Yu et al. (2020) and MOReL Kidambi et al. (2020) (two model-based algorithms). We use Unifloral's online hyperparameter tuning framework as it matches or achieves better performance than papers originally report through its sophisticated online UCB hyperparameter search methods Jackson et al. (2025, Section C). Details are given in Section F.4. In Fig. 2 and Fig. 6 in Section E, we compare the regret of the TOReL-selected hyperparameter combination to the true regret, which we define as the expected regret over all possible hyperparameter combinations. We also compare against the oracle regret: the minimum regret achieved by any hyperparameter combination. We evaluate each algorithm on 8 offline datasets: 200K randomly sampled transitions from each of our three brax datasets, and in the D4RL Fu et al. (2020) Adroit (pen-expert and hammer-expert) and locomotion datasets (halfcheetah-medium-expert, hopper-medium and walker2d-medium-replay) suggested by Jackson et al. (2025). Full results are in Table 3 of

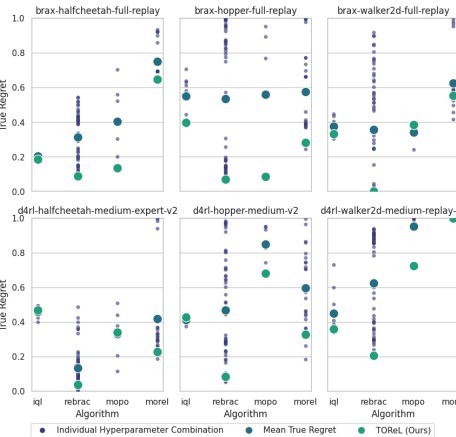

Figure 2: TOReL-selected hyperparameter regret versus mean hyperparameter regret.

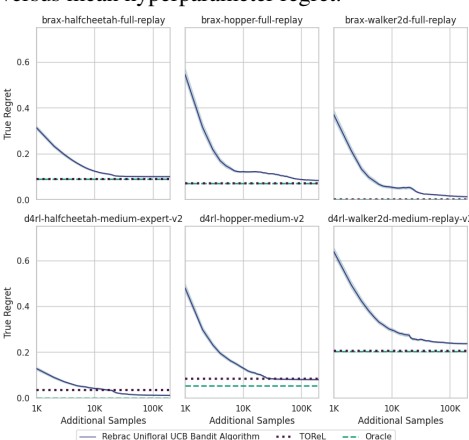

Figure 3: TOReL compared to UCB bandit-based online hyperparameter selection. The x-axis shows the additional samples required during online tuning. Note that he size of the D4RL datasets are 1998K, 1000K and 302K (left to right). UCB 95th percentile confidence interval shaded.

Section E. Table 1 shows ReBRAC+TOReL as a consistently high-achieving combination, reaching near-oracle performance on every dataset. For two-thirds of the tasks and algorithms there is statistically significant ($p < 0.05$), strong ($r > |0.5|$) positive ($r > 0$) Pearson correlation between the ensemble median regret metric and the true regret (Table 5 and Fig. 9 in Section E). Where no strong positive correlation is observed (possibly due to limited hyperparameter coverage) the average TOReL regret (0.433) is still lower than the corresponding true regret (0.458).

Our final experiment analyses the number of samples saved using TOReL rather than the UCB bandit-based online hyperparameter selection algorithm proposed by Jackson et al. (2025). We tune hyperparameters for ReBRAC, as ReBRAC achieves the lowest regret across all tasks and algorithms. Results for the D4RL and brax datasets are depicted in Fig. 3. TOReL offers significant savings in terms of sample complexity compared to existing online hyperparameter tuning methods: for the D4RL datasets, 20K to >200K additional online samples are spared, while for the brax datasets >200K are spared, essentially preventing a doubling of the size of the offline dataset.

### 6.2 SOReL IS A SAFE ALGORITHM FOR ORL

We demonstrate how SOReL can be implemented as a *safe* ORL algorithm. Our goal is test whether SOReL accurately approximates regret over a range of datasets and collection policies including transitions from poor, medium and expert regions of performance to produce high, medium and low regret curves. We evaluate in 5 environments: two gymnax environments and three brax environments. Referring back to Fig. 1b, we progressively include more offline data to learn a policy until a safe level of approximate regret is achieved. For our implementation, we use a variation of the standard Gaussian world model presented in Section 4.4, randomised priors (Osband et al., 2018) for

| Task | Algo. | Oracle | TOReL | Oracle Mean | TOReL Mean | True |
|------|-------|--------|-------|-------------|------------|------|
| brax-halfcheetah-full-replay | ReBRAC | 0.089 | **0.089** | 0.262 | **0.264** | 0.417 |
| brax-hopper-full-replay | ReBRAC | 0.070 | **0.070** | 0.193 | **0.209** | 0.554 |
| brax-walker-full-replay | ReBRAC | 0.000 | **0.000** | 0.241 | 0.317 | 0.425 |
| d4rl-halfcheetah-medium-expert-v2 | ReBRAC | 0.000 | **0.036** | 0.176 | 0.268 | 0.336 |
| d4rl-hopper-medium-v2 | ReBRAC | 0.053 | **0.083** | 0.380 | 0.323 | 0.580 |
| d4rl-walker2d-medium-replay-v2 | ReBRAC | 0.204 | **0.206** | 0.567 | **0.572** | 0.757 |
| d4rl-pen-expert-v1 | ReBRAC | 0.033 | 0.188 | 0.510 | 0.564 | 0.570 |
| d4rl-hammer-expert-v1 | ReBRAC | 0.086 | 0.159 | 0.585 | **0.604** | 0.684 |

Table 1: TOReL Regret Summary Statistics (lower is better): bold indicates where TOReL is within 5% of the corresponding Oracle. Left: algorithm chosen if the Oracle can choose over both hyperparameters *and* algorithms; corresponding oracle regret and regret of the TOReL-chosen hyperparameters for that algorithm. Middle: oracle and TOReL regrets averaged over all algorithms. Right: true regret averaged over all algorithms. For these tasks, ReBRAC+TOReL is consistently the best.

approximate inference and RNN-PPO (Schulman et al., 2017) to solve the BAMDP. Implementation and dataset details are found in Section F.

In practice, each time additional offline data is incorporated, the model, approximate inference and BAMDP hyperparameters should be newly tuned. To avoid too high a computational burden in our experiments, we use fixed model and approximate inference hyperparameters, highlighting in red the region where the approximate regret may be unreliable, and only tune the BAMDP hyperparameters using the approximate regret for one seed and offline dataset size (Fig. 26 in Section E.2). While we deploy each policy in the true environment to validate our approximate regret, in practice, the policy would only be deployed once the approximate regret is sufficiently low. Fig. 4, showing results for halfcheetah-full-replay, along with all other results found in Section E.2, confirm that *SOReL's approximate regret is a good proxy for the true regret*, allowing for the safe deployment of the Bayes-optimal policy. Using the regret and PIL, all hyperparameters can be tuned entirely offline and the practitioner can identify any issues (whether with the offline-dataset, the approximate inference

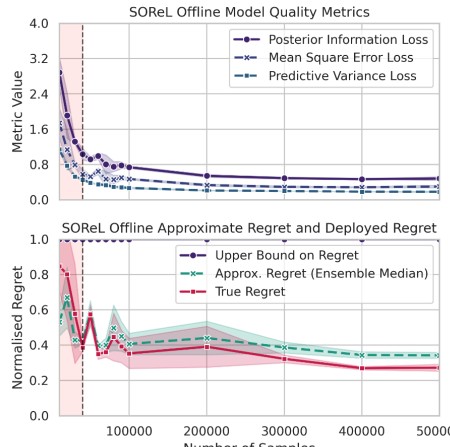

Figure 4: SOReL applied to brax-halfcheetah-full-replay to identify when the policy can be deployed. Shaded red indicates where $\mathcal{E}(\mathcal{D}_N, \mathcal{M}^\star) \not\approx \mathcal{V}(\mathcal{D}_N)$ (for a threshold of 0.25), and hence the approximate regret may be unreliable. Mean and standard deviation given over 3 seeds.

method, or the model) prior to deployment. We also highlight the *generalisability* of our algorithm: while the policy used to collect the halfcheetah dataset achieves an expected episodic return of around 1800 (Fig. 30 in Section E), SOReL's policy (learned on a subset of the offline dataset) achieves a normalised regret of around 0.28 in the true environment (bottom of Fig. 4), corresponding to an undiscounted episode return of just under 2500. As expected (Section 5.1), our experiments show that the utility of the upper bound depends critically on the model being accurate enough relative to the discount factor. More details on a non-trivial upper-bound, along with results for gymnax and the remaining brax environments and ablations of different ensemble metrics that can be used to approximate regret with varying degrees of conservatism are found in Section E.

## 7 CONCLUSION

High online sample complexity and lack of performance guarantees of existing methods present a major barrier to the widespread adoption of offline RL. In this paper, we introduce SOReL and TOReL, two theoretically grounded approaches to tackle these core issues. For SOReL, we introduce a model-based Bayesian approach for offline RL and exploit predictive uncertainty to approximate regret. To tune hyperparameters and ensure accurate regret quantification, we minimise the PIL. In TOReL, we extend our fully offline hyperparameter tuning algorithm to general offline RL methods. Our empirical evaluations confirm SOReL is a reliable method for safe offline RL with accurate regret quantification and TOReL achieves near-oracle performance with *offline data alone*, resulting in significant savings in online samples for hyperparameter tuning without sacrificing performance.

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

## A   BROADER IMPACT

This paper presents work whose goal is to improve the safety and efficacy of offline RL. Our work therefore takes significant steps towards the development of safe offline RL methods with accurate regret guarantees that act in ways that are predictable. We also hope our paper helps to re-frame the discourse on offline RL to focus on safety, practical efficacy and sample efficiency.

More generally, any advancement in RL should be seen in the context of general advancements to machine learning. Whilst machine learning has the potential to develop useful tools to benefit humanity, it must be carefully integrated into underlying political and social systems to avoid negative consequences for people living within them. A discussion of this complex topic lies beyond the scope of this work.

## B   PRIMER ON BAYESIAN RL

A Bayesian epistemology characterises the agent's uncertainty in the MDP through distributions over any unknown variable (Martin, 1967; Duff, 2002). In our learning problem, a Bayesian first specifies a model $P_{R,S}(s_t, a_t)$ over the unknown state transition and reward distribution, representing a hypothesis space of possible environment dynamics. We focus on a parametric model: $p(r_t, s_{t+1}|s_t, a_t, \theta)$ with each $\theta \in \Theta \subseteq \mathbb{R}^d$ representing a hypothesis about the MDP $\mathcal{M}^\star$, however our results can easily be generalised to non-parametric methods like Gaussian process regression (Rasmussen and Williams, 2006; Wiener, 1923; Krige, 1951). A prior distribution over the parameter space $P_\Theta$ is specified, which represents the initial *a priori* belief in the true value of $P^\star_{R,S}(s, a)$ before the agent has observed any transitions. Priors are a powerful aspect of Bayesian RL, allowing practitioners to provide the agent with any information about the MDP and transfer knowledge between agents and domains. Given a history $h_t$, the prior is updated to a posterior $P_\Theta(h_t)$, representing the agent's beliefs in the MDP's dynamics once $h_t$ has been observed. For each history, the posterior is used to *marginalise* across all hypotheses according to the agent's uncertainty, yielding the predictive state transition-reward distribution $P_{R,S}(h_t, a_t) = \mathbb{E}_{\theta \sim P_\Theta(h_t)}[P_{R,S}(s_t, a_t, \theta)]$ which characterise the epistemic and aleatoric uncertainty in $P^\star_{R,S}(s_t, a_t)$. Given $P_{R,S}(h_t, a_t)$, we reason over counterfactual future trajectories using the predictive distribution over trajectories $P^\pi_t$ and define the BRL objective as:

$$J^\pi_{\text{Bayes}}(P_\Theta) := \mathbb{E}_{h_\infty \sim P^\pi_\infty}\left[\sum_{i=0}^\infty \gamma^i r_i\right].$$

Let $\Pi_\mathcal{H}$ denote the space of all history-conditioned policies. A corresponding optimal policy is known as a Bayes-optimal policy, which we denote as $\pi^\star_{\text{Bayes}}(\cdot) \in \Pi^\star_{\text{Bayes}}(P_\Theta) := \arg\max_{\pi \in \Pi_\mathcal{H}} J^\pi_{\text{Bayes}}(P_\Theta)$. Unlike in frequentist RL, Bayesian variables depend on histories obtained through posterior marginalisation; hence the posterior is often known as the *belief state*, which augments each ground state $s_t$ like in a partially observable Markov decision process (Drake, 1962; Åström, Karl Johan, 1965; Smallwood and Sondik, 1973; Kaelbling et al., 1998). Analogously to the state-transition distribution in frequentist RL, we define a *belief transition* distribution $P_\mathcal{H}(h_t, a_t)$ using the predictive state transition-reward distribution, which yields *Bayes-adaptive MDP* (BAMDP) (Duff, 2002): $\mathcal{M}_{\text{Bayes}}(P_\Theta) := \langle \mathcal{H}, \mathcal{A}, P_0, P_\mathcal{H}(h, a), \gamma \rangle$. The BAMDP is solved using planning methods to obtain a Bayes-optimal policy, which naturally balances exploration with exploitation: after every timestep, the agent's uncertainty is characterised via the posterior conditioned on the history $h_t$, which includes all future trajectories to marginalise over. Via the belief transition, the BRL objective accounts for how the posterior evolves on every timestep, and hence any Bayes-optimal policy $\pi^\star_{\text{Bayes}}$ is optimal not only according to the epistemic uncertainty of a fixed belief but accounts for how the epistemic uncertainty evolves at every future timestep, decaying according to the discount factor.

## C   DERIVATIONS

### C.1   NEGATIVE LOG LIKELIHOOD LOSS FUNCTION WITH APPROXIMATE INFERENCE

Assume a dataset of $N$ input-output pairs:
$$\mathcal{D}_N := \{(x_0, y_0), (x_1, y_1), \ldots (x_{N-1}, y_{N-1})\},$$
and a multivariate Gaussian regression model:
$$p(y|x, \theta) = \frac{1}{(2\pi)^{\frac{D}{2}}}|\Sigma_\theta(x)|^{\frac{1}{2}} \exp\left(-\frac{1}{2}(y - \mu_\theta(x))\Sigma_\theta^{-1}(y - \mu_\theta(x))^T\right),$$

where $D$ is the number of dimensions. Here our model $\text{NN}_\theta : \mathcal{X} \to \mathcal{P}(\mathcal{Y})$ is a neural network parametrised by $\theta \in \Theta$ that outputs a Gaussian distribution $\mathcal{N}(\mu_\theta, \Sigma_\theta)$ over $\mathcal{Y}$. Assuming independent dimensions, such that the covariance matrix is diagonal:

$$p(y|x,\theta) = \prod_{d=0}^{D-1} \frac{1}{\sqrt{2\pi\sigma_{\theta_d}^2(x)}} \exp\left(-\frac{1}{2\sigma_{\theta_d}^2(x)}(y_d - \mu_{\theta_d}(x))^2\right).$$

We can then fit our model by minimising the negative log likelihood loss:

$$\mathcal{L}(\text{NLL}(\theta))$$
$$:= -\log p(\mathcal{D}_N|\theta),$$
$$= \sum_{i=0}^{N-1}\left(\frac{D}{2}\log(2\pi) + \frac{1}{2}\sum_{d=0}^{D-1}\left(\log\sigma_{\theta_d}^2(x_i) + \frac{(y_{i_d} - \mu_{\theta_d}(x_i))^2}{\sigma_{\theta_d}^2(x_i)}\right)\right),$$
$$= \left[\sum_{i=0}^{N-1}\frac{1}{N}\left(\frac{D}{2}\log(2\pi) + \frac{1}{2}\sum_{d=0}^{D-1}\left(\log\sigma_{\theta_d}^2(x_i) + \frac{(y_{i_d} - \mu_{\theta_d}(x_i))^2}{\sigma_{\theta_d}^2(x_i)}\right)\right)\right],$$
$$= \mathbb{E}_{i\sim\mathcal{U}_N}\left[\frac{D}{2}\log(2\pi) + \frac{1}{2}\sum_{d=0}^{D-1}\left(\log\sigma_{\theta_d}^2(x_i) + \frac{(y_{i_d} - \mu_{\theta_d}(x_i))^2}{\sigma_{\theta_d}^2(x_i)}\right)\right],$$
$$\overset{c}{=} \mathbb{E}_{i\sim\mathcal{U}_N}\left[\sum_{d=0}^{D-1}\left(\log\sigma_{\theta_d}^2(x_i) + \frac{(y_{i_d} - \mu_{\theta_d}(x_i))^2}{\sigma_{\theta_d}^2(x_i)}\right)\right],$$

where recall $\mathcal{U}_N$ is the uniform distribution over $\{0, 1, \ldots N-1\}$. Our final line means equality up to a constant, as we can ignore the $\frac{D}{2}\log(2\pi)$ term for optimisation because it is independent of $\theta$.

We use RP ensembles for our approximate posterior (Osband et al., 2018; Ciosek et al., 2020); here an ensemble of $M$ separate model weights $\{\theta_0, \theta_i, \ldots \theta_{M-1}\}$ are randomly initialised and are optimised in parallel, summing over the corresponding negative log likelihoods. When training, we optimise the log-variance rather than the variance for numerical stability and to ensure that the variance remains positive. This allows us to simultaneously optimise maximum and minimum log-variance parameters for each dimension across the ensemble, which we use to soft-clamp the log-variances output by individual models, preventing any individual model becoming overly confident or too uncertain in one dimension. Our final loss function is then given by:

$$\mathcal{L}(\theta, \mathcal{D}_N) = \sum_{j=0}^{M-1}\left(\mathbb{E}_{i\sim\mathcal{U}_N}\left[\sum_{d=0}^{D-1}\left(\xi_{\theta_{d_j}}(x_i) + \frac{(y_{i_d} - \mu_{\theta_{d_j}}(x_i))^2}{\exp\left(\xi_{\theta_{d_j}}(x_i)\right)}\right)\right]\right)$$
$$+ c \cdot \sum_{d=0}^{D-1}\left(\xi_{\theta_{d_{\max}}} - \xi_{\theta_{d_{\min}}}\right),$$

where $\xi_{\theta_{d_j}} = \xi_{\theta_{d_{\min}}} + \left[1 + \exp\left(\xi_{\theta_{d_{\max}}} - \left[1 + \exp\left(\log\sigma_{\theta_{d_j}}^2 - \xi_{\theta_{d_{\max}}}\right)\right] - \xi_{\theta_{d_{\min}}}\right)\right]$ and $\xi_{\theta_{d_{\min}}}$ and $\xi_{\theta_{d_{\max}}}$ are respectively the minimum and maximum log-variance parameters optimised across the ensemble, $c$ is the log-variance difference coefficient used to control the clamping term, and $M$ is the number of models in the ensemble. $\mathcal{L}(\theta, \mathcal{D}_N)$ can be minimised by using Monte Carlo minibatch gradient descent with a minibatch $\mathcal{M}_n$ of $n < N$ samples drawn uniformly from $\mathcal{D}_N$.

## C.2 Regret Approximators

**Predictive Variance:** We now show how the true regret can be approximated using the Bayesian predictive variance of returns. We start with the bound on from Ineq. 9. Defining the discounted return $R(h_\infty) := \sum_{i=0}^\infty \gamma^i r_i$:

$$\left|J^\pi(\mathcal{M}^\star) - J_{\text{Bayes}}^\pi(P_\Theta(\mathcal{D}_N))\right| = \left|J^\pi(\mathcal{M}^\star) - J_{\text{Bayes}}^\pi(P_\Theta(\mathcal{D}_N))\right|,$$
$$= \left|J^\pi(\mathcal{M}^\star) - \mathbb{E}_{\theta\sim P_\Theta(\mathcal{D}_N), h_\infty\sim P_\infty^\pi(\theta)}[R(h_\infty)]\right|,$$
$$= \left|\mathbb{E}_{\theta\sim P_\Theta(\mathcal{D}_N), h_\infty\sim P_\infty^\pi(\theta)}[J^\pi(\mathcal{M}^\star) - R(h_\infty)]\right|,$$
$$= \sqrt{\left(\mathbb{E}_{\theta\sim P_\Theta(\mathcal{D}_N), h_\infty\sim P_\infty^\pi(\theta)}[J^\pi(\mathcal{M}^\star) - R(h_\infty)]\right)^2}.$$

Applying Jensen's inequality:

$$\left| J^\pi(\mathcal{M}^\star) - J^\pi_{\text{Bayes}}(P_\Theta(\mathcal{D}_N)) \right| \leq \sqrt{\mathbb{E}_{\theta \sim P_\Theta(\mathcal{D}_N), h_\infty \sim P^\pi_\infty(\theta)} \left[ (J^\pi(\mathcal{M}^\star) - R(h_\infty))^2 \right]}. \quad (6)$$

We now recognise that the mean squared error term in Eq. (6) relies on knowing the true MDP dynamics $J^\pi(\mathcal{M}^\star)$. We can approximate this term using the using the predictive variance over returns:

$$\mathbb{E}_{\theta \sim P_\Theta(\mathcal{D}_N), h_\infty \sim P^\pi_\infty(\theta)} \left[ (J^\pi(\mathcal{M}^\star) - R(h_\infty))^2 \right]$$
$$\approx \mathbb{E}_{\theta \sim P_\Theta(\mathcal{D}_N), h_\infty \sim P^\pi_\infty(\theta)} \left[ \left( J^\pi_{\text{Bayes}}(\mathcal{D}_N) - R(h_\infty) \right)^2 \right],$$
$$= \mathbb{V}_{\theta \sim P_\Theta(\mathcal{D}_N), h_\infty \sim P^\pi_\infty(\theta)} \left[ R(h_\infty) \right],$$

which can be estimated using the dataset $\mathcal{D}_N$, yielding:

$$\left| J^\pi(\mathcal{M}^\star) - J^\pi_{\text{Bayes}}(P_\Theta(\mathcal{D}_N)) \right| \leq \sqrt{\mathbb{V}_{\theta \sim P_\Theta(\mathcal{D}_N), h_\infty \sim P^\pi_\infty(\theta)} \left[ R(h_\infty) \right]}.$$

Finally, using Ineq. 9 this justifies our approximation for estimating the regret:

$$\text{Regret}(\mathcal{M}^\star, \mathcal{D}_N) \approx 2\sqrt{\mathbb{V}_{\theta \sim P_\Theta(\mathcal{D}_N), h_\infty \sim P^\pi_\infty(\theta)} \left[ R(h_\infty) \right]}.$$

To improve the approximation, we conservatively upper-bound the regret based on alternate ensemble statistics with varying degrees of conservatism to prevent associating a low regret with a policy that performs equally poorly in all members of the ensemble:

$$\text{Regret}(\mathcal{M}^\star, \mathcal{D}_N) \approx \max \left[ 2\sqrt{\mathbb{V}_{\theta \sim P_\Theta(\mathcal{D}_N), h_\infty \sim P^\pi_\infty(\theta)} \left[ R(h_\infty) \right]}, \right.$$
$$\left. \hat{R}_{\max} - \hat{\mathbb{M}}_{\theta \sim P_\Theta(\mathcal{D}_N), h_\infty \sim P^\pi_\infty(\theta)} \left[ R(h_\infty) \right] \right],$$

for example, here $\hat{\mathbb{M}}_{\theta \sim P_\Theta(\mathcal{D}_N), h_\infty \sim P^\pi_\infty(\theta)} \left[ R(h_\infty) \right]$ denotes the median predictive returns based on sampling from the (approximate) posterior and rolling out the Bayes-optimal policy. $\hat{R}_{\max}$ is estimated from the maximum return in the offline dataset. In Fig. 27 and Fig. 29, we plot different ensemble statistics (the ensemble mean, median, maximum and minimum regrets) that can be used to inform the approximate regret: we shade the regret based on the range of these statistics in purple. As long as the true environment falls in the space spanned by the posterior (model ensemble), the true regret is guaranteed to lie within this range. By ensuring that the (normalised) predictive variance is at least as large as the (normalised) MSE in the PIL, the space spanned by the approximate posterior via model ensemble is approximately large enough, relative to the model error. Below we order different ensemble statistics from least to most conservative:

$$\text{Regret}(\mathcal{M}^\star, \mathcal{D}_N) \approx \hat{R}_{\max} - \hat{R}_{\theta \sim P_\Theta(\mathcal{D}_N), h_\infty \sim P^\pi_\infty(\theta)} \left[ R(h_\infty) \right],$$
$$\leq \hat{R}_{\max} - \hat{\mathbb{E}}_{\theta \sim P_\Theta(\mathcal{D}_N), h_\infty \sim P^\pi_\infty(\theta)} \left[ R(h_\infty) \right],$$
$$\approx \hat{R}_{\max} - \hat{\mathbb{M}}_{\theta \sim P_\Theta(\mathcal{D}_N), h_\infty \sim P^\pi_\infty(\theta)} \left[ R(h_\infty) \right],$$
$$\leq \hat{R}_{\max} - \hat{r}_{\theta \sim P_\Theta(\mathcal{D}_N), h_\infty \sim P^\pi_\infty(\theta)} \left[ R(h_\infty) \right].$$

Here $\hat{R}_{\theta \sim P_\Theta(\mathcal{D}_N), h_\infty \sim P^\pi_\infty(\theta)} \left[ R(h_\infty) \right]$, $\hat{\mathbb{E}}_{\theta \sim P_\Theta(\mathcal{D}_N), h_\infty \sim P^\pi_\infty(\theta)} \left[ R(h_\infty) \right]$, and $\hat{r}_{\theta \sim P_\Theta(\mathcal{D}_N), h_\infty \sim P^\pi_\infty(\theta)} \left[ R(h_\infty) \right]$, respectively denote the maximum, mean, and minimum predictive returns based on sampling from the (approximate) posterior and rolling out the Bayes-optimal policy. We note that the minimum predictive return leads to the maximum predictive regret: "Ensemble Max" in Fig. 27 and Fig. 29 refer to the maximum predictive *regrets* rather than maximum predictive *returns*. Empirically, we find that the ensemble median alone is a good proxy for the true regret, being neither overly conservative, nor overly susceptible to being skewed by a policy that performs well on only a subset of posterior samples (as the mean might be). Using the variance, a policy with high variance but also high mean will be associated with a high approximate regret, which is the case for brax-hopper-full-replay (Fig. 29c), where the variance actually overestimates the regret.

# D PROOFS

## D.1 PRIMER ON TOTAL VARIATIONAL DISTANCE

We measure distance between two probability distributions $P_X$ and $Q_X$ using the total variational (TV) distance, defined as:

$$\text{TV}(P_X \| Q_X) := \sup_E |P_X(E) - Q_X(E)|.$$

The TV distance takes the supremum over all events $E$ to find the event that gives rise to the maximum difference in probability between two distributions. A key property of the TV distance is: $0 \le \text{TV}(P_X \| Q_X) \le 1$. If $\text{TV}(P_X \| Q_X) = 0$, then $P_X = Q_X$ as there is no event that both distributions don't assign the same probability mass to. If $\text{TV}(P_X \| Q_X) = 1$, then the distributions assign completely different mass to at least one event. The TV distance can be related to the Kullback-Leibler (KL) divergence using the Bretagnolle-Huber (Bretagnolle and Huber, 1978) inequality: $\text{TV}(P_X \| Q_X) \le \sqrt{1 - \exp(-\text{KL}(P_X \| Q_X))} \le 1$, which preserves the property that $0 \le \text{TV}(P_X \| Q_X) \le 1$. The TV distance can be shown (Sriperumbudur et al., 2009) to be equivalent to the integral probability metric under the $\infty$-norm, which we will make use of in our theorems:

$$\text{TV}(P_X \| Q_X) = \frac{1}{2} \sup_{f \in \mathcal{F}: \mathcal{X} \to [-1,1]} |\mathbb{E}_{x \sim P_X}[f(x)] - \mathbb{E}_{x \sim Q_X}[f(x)]|, \tag{7}$$

In this form, the supremum is taken over the space of all functions that are bounded by unity, that is $\|f\|_\infty = 1$.

## D.2 PROOF OF THEOREM 1

Let the predictive distribution over history $h_t$ using the posterior $P_\Theta(\mathcal{D}_N)$ be $P_{t,\pi}(\mathcal{D}_N)$, which has density:

$$p_\pi(h_t, \mathcal{D}_N) := p_0(s_0) \prod_{i=0}^{t-1} \pi(a_i|h_i) p(r_i|h_i, a_i, \mathcal{D}_N) p(s_{i+1}|h_i, a_i, \mathcal{D}_N).$$

According to the Bernstein-von Mises theorem (Doob, 1949; Le Cam, 1953; Vaart, 1998), as the posterior becomes more informative it concentrates around a smaller (and more tractable) subset of the hypothesis space. Not only does this ease the computational burden of solving the BRL objective, but in the limit $N \to \infty$, the Bayesian RL objective using the true posterior will approach the true expected discounted return for the MDP: $J^\pi(P_\Theta(\mathcal{D}_N)) \xrightarrow[N \to \infty]{} J^\pi(\mathcal{M}^\star)$. In this limit, any Bayes-optimal policy will be an optimal policy for the true MDP, achieving the highest expected returns once deployed. To make progress towards quantifying how much offline data we need to achieve an acceptable level of regret, we first relate the true regret $\text{Regret}(\mathcal{M}^\star, \mathcal{D}_N)$ to the TV distance between the true $P^\star_{i,\pi}$ and predictive $P_{i,\pi}(\mathcal{D}_N)$ history distributions:

**Lemma 1.** *Let* $\mathcal{R}_{\max} := \frac{(r_{max} - r_{min})}{1-\gamma}$ *denote the maximum possible regret for the MDP. For a prior* $P_\Theta(\mathcal{D}_N)$, *the true regret can be bounded as:*

$$\text{Regret}(\mathcal{M}^\star, \mathcal{D}_N) \le 2\mathcal{R}_{\max} \cdot \sup_\pi \mathbb{E}_{i \sim \mathcal{G}(\gamma)}\left[TV\left(P^\star_{i+1,\pi} \| P^\pi_{i+1}(\mathcal{D}_N)\right)\right]. \tag{8}$$

*Proof.* We start from the definition of the true regret:

$$\text{Regret}(\mathcal{M}^\star, \mathcal{D}_N) := J^{\pi^*}(\mathcal{M}^\star) - J^{\pi^*_{\text{Bayes}}}(\mathcal{M}^\star, \mathcal{D}_N).$$

We now bound the difference between $J^{\pi^*}(\mathcal{M}^\star)$ and $J^{\pi^*_{\text{Bayes}}}(\mathcal{M}^\star, \mathcal{D}_N)$ in terms of the difference between $J^\pi(\mathcal{M}^\star)$ and $J^\pi_{\text{Bayes}}(\mathcal{D}_N)$:

$$\begin{aligned}
&\text{Regret}(\mathcal{M}^\star, \mathcal{D}_N) \\
&= J^{\pi^*}(\mathcal{M}^\star) - J^{\pi^*}_{\text{Bayes}}(P_\Phi(\mathcal{D}_N)) + J^{\pi^*}_{\text{Bayes}}(P_\Phi(\mathcal{D}_N)) - J^{\pi^*_{\text{Bayes}}}(\mathcal{M}^\star, \mathcal{D}_N), \\
&\le J^{\pi^*}(\mathcal{M}^\star) - J^{\pi^*}_{\text{Bayes}}(P_\Phi(\mathcal{D}_N)) + J^{\pi^*_{\text{Bayes}}}_{\text{Bayes}}(P_\Phi(\mathcal{D}_N)) - J^{\pi^*_{\text{Bayes}}}(\mathcal{M}^\star, \mathcal{D}_N), \\
&\le \sup_\pi \left| J^\pi(\mathcal{M}^\star) - J^\pi_{\text{Bayes}}(P_\Theta(\mathcal{D}_N)) \right| + \sup_\pi \left| J^\pi_{\text{Bayes}}(P_\Theta(\mathcal{D}_N)) - J^\pi(\mathcal{M}^\star) \right|, \\
&= 2 \sup_\pi \left| J^\pi(\mathcal{M}^\star) - J^\pi_{\text{Bayes}}(P_\Theta(\mathcal{D}_N)) \right|, \tag{9}
\end{aligned}$$

where the second line follows from $J_{\text{Bayes}}^{\pi^\star}(P_\Phi(\mathcal{D}_N)) \leq J_{\text{Bayes}}^{\pi_{\text{Bayes}}^\star}(P_\Phi(\mathcal{D}_N))$ by definition. Now our goal is to bound $\left| J^\pi(\mathcal{M}^\star) - J_{\text{Bayes}}^\pi(P_\Theta(\mathcal{D}_N)) \right|$:

$$
\left| J^\pi(\mathcal{M}^\star) - J_{\text{Bayes}}^\pi(P_\Theta(\mathcal{D}_N)) \right| = \left| \mathbb{E}_{h_\infty \sim P_{\infty,\pi}^\star} \left[ \sum_{i=0}^\infty \gamma^i r_i \right] - \mathbb{E}_{h_\infty \sim P_\infty^\pi(\mathcal{D}_N)} \left[ \sum_{i=0}^\infty \gamma^i r_i \right] \right|,
$$

$$
= \left| \sum_{i=0}^\infty \gamma^i \mathbb{E}_{h_{i+1} \sim P_{i+1,\pi}^\star} [r_i] - \sum_{i=0}^\infty \gamma^i \mathbb{E}_{h_{i+1} \sim P_{i+1}^\pi(\mathcal{D}_N)} [r_i] \right|,
$$

$$
= \left| \sum_{i=0}^\infty \gamma^i \left( \mathbb{E}_{h_{i+1} \sim P_{i+1,\pi}^\star} [r_i] - \mathbb{E}_{h_{i+1} \sim P_{i+1}^\pi(\mathcal{D}_N)} [r_i] \right) \right|,
$$

$$
\leq \sum_{i=0}^\infty \gamma^i \left| \mathbb{E}_{h_{i+1} \sim P_{i+1,\pi}^\star} [r_i] - \mathbb{E}_{h_{i+1} \sim P_{i+1}^\pi(\mathcal{D}_N)} [r_i] \right|.
$$

Using Ineq. 10 from Lemma 2, we now bound each difference $\left| \mathbb{E}_{h_{i+1} \sim P_{i+1,\pi}^\star} [r_i] - \mathbb{E}_{h_{i+1} \sim P_{i+1}^\pi(\mathcal{D}_N)} [r_i] \right|$ in terms of total variational distance between $P_{i+1,\pi}^\star$ and $P_{i+1}^\pi(\mathcal{D}_N)$:

$$
\left| J^\pi(\mathcal{M}^\star) - J_{\text{Bayes}}^\pi(P_\Theta(\mathcal{D}_N)) \right| \leq (r_{\max} - r_{\min}) \cdot \sum_{i=0}^\infty \gamma^i \text{TV}\left( P_{i+1,\pi}^\star \| P_{i+1}^\pi(\mathcal{D}_N) \right),
$$

$$
= \frac{r_{\max} - r_{\min}}{1 - \gamma} \cdot \sum_{i=0}^\infty (1 - \gamma) \gamma^i \text{TV}\left( P_{i+1,\pi}^\star \| P_{i+1}^\pi(\mathcal{D}_N) \right),
$$

$$
= \frac{r_{\max} - r_{\min}}{1 - \gamma} \cdot \mathbb{E}_{i \sim \mathcal{G}(\gamma)} \left[ \text{TV}\left( P_{i+1,\pi}^\star \| P_{i+1}^\pi(\mathcal{D}_N) \right) \right],
$$

$$
= \mathcal{R}_{\max} \cdot \mathbb{E}_{i \sim \mathcal{G}(\gamma)} \left[ \text{TV}\left( P_{i+1,\pi}^\star \| P_{i+1}^\pi(\mathcal{D}_N) \right) \right],
$$

where $\mathcal{G}(\gamma)$ is the geometric distribution. Finally, substituting into Ineq. 9 yields our desired result:

$$
\text{Regret}(\mathcal{M}^\star, \mathcal{D}_N) \leq 2 \sup_\pi \left| \left( \mathcal{R}_{\max} \cdot \mathbb{E}_{i \sim \mathcal{G}(\gamma)} \left[ \text{TV}\left( P_{i+1,\pi}^\star \| P_{i+1}^\pi(\mathcal{D}_N) \right) \right] \right) \right|,
$$

$$
= 2 \mathcal{R}_{\max} \cdot \sup_\pi \mathbb{E}_{i \sim \mathcal{G}(\gamma)} \left[ \text{TV}\left( P_{i+1,\pi}^\star \| P_{i+1}^\pi(\mathcal{D}_N) \right) \right].
$$

$\square$

We remark that Lemma 1 holds for any general reward-transition model given bounded rewards. The bound in Ineq. 8 proves the true regret is governed by the geometric average of TV distances: $\mathbb{E}_{i \sim \mathcal{G}(\gamma)} \left[ \text{TV}\left( P_{i+1,\pi}^\star \| P_{i+1}^\pi(\mathcal{D}_N) \right) \right]$. As each term $\text{TV}\left( P_{i+1,\pi}^\star \| P_{i+1}^\pi(\mathcal{D}_N) \right)$ measures the distance between the true and predictive distributions over history $h_i$ of length $i$, the discounting factor $\gamma$ determines how much long term histories contribute to regret.

Intuitively, the more mass the posterior places close to the true value $\theta^\star \in \Theta^\star$, the smaller each TV distance becomes, with regret tending to zero for $P_{i+1,\pi}(\mathcal{D}_N) \approx P_{i+1,\pi}^\star \implies \text{TV}\left( P_{i+1,\pi}^\star \| P_{i+1}^\pi(\mathcal{D}_N) \right) \approx 0$. Conversely, a strong but highly incorrect prior will concentrate mass around MDPs whose dynamics oppose the true dynamics, yielding $\text{TV}\left( P_{i+1,\pi}^\star \| P_{i+1}^\pi(\mathcal{D}_N) \right) \approx 1$ for all $i$, achieving the highest possible regret: $\mathcal{R}_{\max} := (r_{\max} - r_{\min})/1 - \gamma$. The resulting Bayes-optimal policy would choose actions that encourage negative reward-seeking behaviour, being farthest from optimal in terms of expected returns.

Our next lemma

**Lemma 2.** *For bounded reward functions:*

$$
\left| \mathbb{E}_{h_{i+1} \sim P_{i+1,\pi}^\star} [r_i] - \mathbb{E}_{h_{i+1} \sim P_{i+1}^\pi(\mathcal{D}_N)} [r_i] \right| \leq (r_{\max} - r_{\min}) \cdot \text{TV}\left( P_{i+1,\pi}^\star \| P_{i+1}^\pi(\mathcal{D}_N) \right). \quad (10)
$$

*Proof.* We start by subtracting and adding $\frac{r_{\max} + r_{\min}}{2}$ to the left hand side of Ineq. 10:

$$\left| \mathbb{E}_{h_{i+1} \sim P^{\star}_{i+1,\pi}} [r_i] - \mathbb{E}_{h_{i+1} \sim P^{\pi}_{i+1}(\mathcal{D}_N)} [r_i] \right|$$

$$= \left| \mathbb{E}_{h_{i+1} \sim P^{\star}_{i+1,\pi}} \left[ r_i - \frac{r_{\max} + r_{\min}}{2} \right] - \mathbb{E}_{h_{i+1} \sim P^{\pi}_{i+1}(\mathcal{D}_N)} \left[ r_i - \frac{r_{\max} + r_{\min}}{2} \right] \right|,$$

$$= \left| \mathbb{E}_{h_{i+1} \sim P^{\star}_{i+1,\pi}} \left[ \frac{2r_i - (r_{\max} + r_{\min})}{2} \right] - \mathbb{E}_{h_{i+1} \sim P^{\pi}_{i+1}(\mathcal{D}_N)} \left[ \frac{2r_i - (r_{\max} + r_{\min})}{2} \right] \right|,$$

$$= \frac{(r_{\max} - r_{\min})}{2} \cdot \left| \mathbb{E}_{h_{i+1} \sim P^{\star}_{i+1,\pi}} \left[ \frac{2r_i - (r_{\max} + r_{\min})}{r_{\max} - r_{\min}} \right] - \mathbb{E}_{h_{i+1} \sim P^{\pi}_{i+1}(\mathcal{D}_N)} \left[ \frac{2r_i - (r_{\max} + r_{\min})}{r_{\max} - r_{\min}} \right] \right|,$$

$$= \frac{(r_{\max} - r_{\min})}{2} \cdot \left| \mathbb{E}_{h_{i+1} \sim P^{\star}_{i+1,\pi}} [r_{\text{norm}}(h_{i+1})] - \mathbb{E}_{h_{i+1} \sim P^{\pi}_{i+1}(\mathcal{D}_N)} [r_{\text{norm}}(h_{i+1})] \right|, \tag{11}$$

where:

$$r_{\text{norm}}(h_{i+1}) := \frac{2r_i - (r_{\max} + r_{\min})}{r_{\max} - r_{\min}}.$$

Now, as $r_{\text{norm}} : \mathcal{H}_{i+1} \to [-1, 1]$, we can bound Eq. (11) using the integral probability metric form of the TV distance (see Eq. (7)), yielding our desired result:

$$\left| \mathbb{E}_{h_{i+1} \sim P^{\star}_{i+1,\pi}} [r_i] - \mathbb{E}_{h_{i+1} \sim P^{\pi}_{i+1}(\mathcal{D}_N)} [r_i] \right|$$

$$\leq \frac{r_{\max} - r_{\min}}{2} \cdot \sup_{f \in \mathcal{F} : \mathcal{H}_{i+1} \to [-1,1]} \left| \mathbb{E}_{h_{i+1} \sim P^{\star}_{i+1,\pi}} [f(h_{i+1})] - \mathbb{E}_{h_{i+1} \sim P^{\pi}_{i+1}(\mathcal{D}_N)} [f(h_{i+1})] \right|,$$

$$= (r_{\max} - r_{\min}) \cdot \text{TV} \left( P^{\star}_{i+1,\pi} \| P^{\pi}_{i+1}(\mathcal{D}_N) \right).$$

$\square$

We proved in Lemma 1 that the rate of convergence of the sum of discounted TV distances between the true and predictive history distributions governs the rate of decrease in regret decreases with increasing data. Using the Bretagnolle-Huber inequality (see Section D.1), we now relate the sum of discounted TV distances to a sum of KL divergences, allowing us to control the expected regret using the PIL.

**Theorem 1.** *Using the PIL $\mathcal{I}^{\pi}_N$, the true regret is bounded as:*

$$\text{Regret}(\mathcal{M}^{\star}, \mathcal{D}_N) \leq 2\mathcal{R}_{\max} \cdot \sup_{\pi} \sqrt{1 - \exp\left( -\frac{\mathcal{I}^{\pi}_N}{1 - \gamma} \right)}$$

*Proof.* Starting with the bounded derived in Ineq. 8 of Lemma 1, we apply the Bretagnolle-Huber inequality (Bretagnolle and Huber, 1978) (see Section D.1) to bound the TV distance terms using the KL divergence:

$$\text{Regret}(\mathcal{M}^{\star}, \mathcal{D}_N) \leq 2\mathcal{R}_{\max} \cdot \sup_{\pi} \mathbb{E}_{i \sim \mathcal{G}(\gamma)} \left[ \text{TV} \left( P^{\star}_{i+1,\pi} \| P^{\pi}_{i+1}(\mathcal{D}_N) \right) \right],$$

$$\leq 2\mathcal{R}_{\max} \cdot \sup_{\pi} \mathbb{E}_{i \sim \mathcal{G}(\gamma)} \left[ \sqrt{1 - \exp\left( -\text{KL} \left( P^{\star}_{i+1,\pi} \| P^{\pi}_{i+1}(\mathcal{D}_N) \right) \right)} \right]. \tag{12}$$

We make two observations. Firstly, as the KL divergence is convex in its second argument and $P_{i+1,\pi}(\mathcal{D}_N) = \mathbb{E}_{\theta \sim P_{\Theta}(\mathcal{D}_N)} \left[ P^{\pi}_{i+1}(\theta) \right]$, we can bound each KL divergence term using Jensen's inequality as:

$$\text{KL} \left( P^{\star}_{i+1,\pi} \| P^{\pi}_{i+1}(\mathcal{D}_N) \right) \leq \mathbb{E}_{\theta \sim P_{\Theta}(\mathcal{D}_N)} \left[ \text{KL} \left( P^{\star}_{i+1,\pi} \| P^{\pi}_{i+1}(\theta) \right) \right].$$

Secondly, as the function $f(x) = \sqrt{1 - \exp(-x)}$ is monotonically increasing in $x$, it follows that $f(x) \leq f(x')$ for any $x \leq x'$, hence:

$$\sqrt{1 - \exp\left( -\text{KL} \left( P^{\star}_{i+1,\pi} \| P^{\pi}_{i+1}(\mathcal{D}_N) \right) \right)} \leq \sqrt{1 - \exp\left( -\mathbb{E}_{\theta \sim P_{\Theta}(\mathcal{D}_N)} \left[ \text{KL} \left( P^{\star}_{i+1,\pi} \| P^{\pi}_{i+1}(\theta) \right) \right] \right)}$$

Applying this bound to Ineq. 12 yields:

$$\mathbb{E}_{i \sim \mathcal{G}(\gamma)} \left[ \sqrt{1 - \exp\left(-\mathrm{KL}\left(P^\star_{i+1,\pi} \| P^\pi_{i+1}(\mathcal{D}_N)\right)\right)} \right]$$

$$\leq \mathbb{E}_{i \sim \mathcal{G}(\gamma)} \left[ \sqrt{1 - \exp\left(-\mathbb{E}_{\theta \sim P_\Theta(\mathcal{D}_N)}\left[\mathrm{KL}\left(P^\star_{i+1,\pi} \| P^\pi_{i+1}(\theta)\right)\right]\right)} \right].$$

As the function $f(x) = \sqrt{1 - \exp(-x)}$ is concave in $x$, we can apply Jensen's inequality, yielding:

$$\mathbb{E}_{i \sim \mathcal{G}(\gamma)} \left[ \sqrt{1 - \exp\left(-\mathrm{KL}\left(P^\star_{i+1,\pi} \| P^\pi_{i+1}(\mathcal{D}_N)\right)\right)} \right]$$

$$\leq \sqrt{1 - \exp\left(-\mathbb{E}_{i \sim \mathcal{G}(\gamma)}\left[\mathbb{E}_{\theta \sim P_\Theta(\mathcal{D}_N)}\left[\mathrm{KL}\left(P^\star_{i+1,\pi} \| P^\pi_{i+1}(\theta)\right)\right]\right]\right)}. \tag{13}$$

Examining the KL divergence term:

$$\mathrm{KL}\left(P^\star_{i+1,\pi} \| P^\pi_{i+1}(\theta)\right)$$

$$= \mathbb{E}_{h_{i+1} \sim P^\star_{i+1,\pi}} \left[ \log\left( \frac{d_0(s_0) \prod_{j=0}^{i} \pi(a_j|h_j) p^\star(r_j|s_j, a_j) p^\star(s_{j+1}|s_j, a_j)}{d_0(s_0) \prod_{j=0}^{i} \pi(a_j|h_j) p(r_j|s_j, a_j, \theta) p(s_{j+1}|s_j, a_j, \theta)} \right) \right],$$

$$= \mathbb{E}_{h_{i+1} \sim P^\star_{i+1,\pi}} \left[ \log\left( \frac{\prod_{j=0}^{i} p^\star(r_j|s_j, a_j) p^\star(s_{j+1}|s_j, a_j)}{\prod_{j=0}^{i} p(r_j|s_j, a_j, \theta) p(s_{j+1}|s_j, a_j, \theta)} \right) \right],$$

$$= \mathbb{E}_{h_{i+1} \sim P^\star_{i+1,\pi}} \left[ \sum_{j=0}^{i} \left( \log p^\star(r_j|s_j, a_j) - \log p(r_j|s_j, a_j, \theta) \right. \right.$$

$$\left. \left. + \log p^\star(s_{j+1}|s_j, a_j) - \log p(s_{j+1}|s_j, a_j, \theta) \right) \right],$$

$$= \sum_{j=0}^{i} \mathbb{E}_{h_j \sim P^\star_{j,\pi}} \left[ \left( \log p^\star(r_j|s_j, a_j) - \log p(r_j|s_j, a_j, \theta) \right. \right.$$

$$\left. \left. + \log p^\star(s_{j+1}|s_j, a_j) - \log p(s_{j+1}|s_j, a_j, \theta) \right) \right],$$

$$= \sum_{j=0}^{i} \mathbb{E}_{s_j, a_j \sim P^\star_{j,\pi}} \left[ \mathbb{E}_{r_j, s_{j+1} \sim P^\star_{R,S}(s_j, a_j)} \left[ \left( \log p^\star(r_j|s_j, a_j) - \log p(r_j|s_j, a_j, \theta) \right. \right. \right.$$

$$\left. \left. \left. + \log p^\star(s_{j+1}|s_j, a_j) - \log p(s_{j+1}|s_j, a_j, \theta) \right) \right] \right],$$

$$= \sum_{j=0}^{i} \mathbb{E}_{s_j, a_j \sim P^\star_{j,\pi}} \left[ \mathbb{E}_{r_j, s_{j+1} \sim P^\star_{R,S}(s_j, a_j)} \left[ \left( \log p^\star(r_j, s_{j+1}|s_j, a_j) \right. \right. \right.$$

$$\left. \left. \left. - \log p(r_j, s_{j+1}|s_j, a_j, \theta) \right) \right] \right],$$

$$= \sum_{j=0}^{i} \mathbb{E}_{s,a \sim P^\star_{j,\pi}} \left[ \mathrm{KL}(P^\star_{R,S}(s, a) \| P_{R,S}(s, a, \theta)) \right],$$

hence:

$$\mathbb{E}_{i\sim\mathcal{G}(\gamma)}\left[\mathbb{E}_{\theta\sim P_\Theta(\mathcal{D}_N)}\left[\text{KL}\left(P^\star_{i+1,\pi}\|P^\pi_{i+1}(\theta)\right)\right]\right]$$

$$= \mathbb{E}_{\theta\sim P_\Theta(\mathcal{D}_N)}\left[\mathbb{E}_{i\sim\mathcal{G}(\gamma)}\left[\sum_{j=0}^{i}\mathbb{E}_{s,a\sim P^\star_{j,\pi}}\left[\text{KL}(P^\star_{R,S}(s,a)\|P_{R,S}(s,a,\theta))\right]\right]\right],$$

$$= \mathbb{E}_{\theta\sim P_\Theta(\mathcal{D}_N)}\left[\sum_{i=0}^{\infty}(1-\gamma)\gamma^i\sum_{j=0}^{i}\mathbb{E}_{s,a\sim P^\star_{j,\pi}}\left[\text{KL}(P^\star_{R,S}(s,a)\|P_{R,S}(s,a,\theta))\right]\right],$$

$$= \mathbb{E}_{\theta\sim P_\Theta(\mathcal{D}_N)}\left[\sum_{i=0}^{\infty}(1-\gamma)\gamma^i(i+1)\sum_{j=0}^{i}\frac{1}{i+1}\mathbb{E}_{s,a\sim P^\star_{j,\pi}}\left[\text{KL}(P^\star_{R,S}(s,a)\|P_{R,S}(s,a,\theta))\right]\right],$$

$$= \frac{1}{1-\gamma}\mathbb{E}_{\theta\sim P_\Theta(\mathcal{D}_N)}\left[\sum_{i=0}^{\infty}(1-\gamma)^2\gamma^i(i+1)\mathbb{E}_{j\sim\mathcal{U}_i}\left[\mathbb{E}_{s,a\sim P^\star_{j,\pi}}\left[\text{KL}(P^\star_{R,S}(s,a)\|P_{R,S}(s,a,\theta))\right]\right]\right],$$

$$= \frac{1}{1-\gamma}\mathbb{E}_{\theta\sim P_\Theta(\mathcal{D}_N)}\left[\mathbb{E}_{i\sim\mathcal{AG}(\gamma)}\left[\mathbb{E}_{j\sim\mathcal{U}_i}\left[\mathbb{E}_{s,a\sim P^\star_{j,\pi}}\left[\text{KL}(P^\star_{R,S}(s,a)\|P_{R,S}(s,a,\theta))\right]\right]\right]\right]. \quad (14)$$

Now, as $\rho^\star_\pi = \mathbb{E}_{i\sim\mathcal{AG}(\gamma)}\left[\mathbb{E}_{j\sim\mathcal{U}_i}\left[P^\star_{j,\pi}\right]\right]$ is the arithemetico-geometric ergodic state-action distribution, we can simplify Eq. (14) to yield:

$$\mathbb{E}_{i\sim\mathcal{G}(\gamma)}\left[\mathbb{E}_{\theta\sim P_\Theta(\mathcal{D}_N)}\left[\text{KL}\left(P^\star_{i+1,\pi}\|P^\pi_{i+1}(\theta)\right)\right]\right]$$

$$= \frac{1}{1-\gamma}\mathbb{E}_{\mathcal{D}_N\sim P_{\text{Data}}}\left[\mathbb{E}_{\theta\sim P_\Theta(\mathcal{D}_N)}\left[\mathbb{E}_{s,a\sim\rho^\star_\pi}\left[\text{KL}(P^\star_{R,S}(s,a)\|P_{R,S}(s,a,\theta))\right]\right]\right],$$

$$= \frac{1}{1-\gamma}\mathbb{E}_{\mathcal{D}_N\sim P_{\text{Data}}}\left[\mathbb{E}_{s,a\sim\rho^\star_\pi}\left[\mathbb{E}_{\theta\sim P_\Theta(\mathcal{D}_N)}\left[\text{KL}(P^\star_{R,S}(s,a)\|P_{R,S}(s,a,\theta))\right]\right]\right],$$

$$= \frac{1}{1-\gamma}\mathcal{I}^\pi_N,$$

hence, substituting into Ineq. 13, we obtain:

$$\mathbb{E}_{i\sim\mathcal{G}(\gamma)}\left[\sqrt{1-\exp\left(-\text{KL}\left(P^\star_{i+1,\pi}\|P^\pi_{i+1}(\mathcal{D}_N)\right)\right)}\right] \leq \sqrt{1-\exp\left(-\frac{\mathcal{I}^\pi_N}{1-\gamma}\right)}.$$

Finally, substituting into Eq. (12) yields our desired result:

$$\text{Regret}(\mathcal{M}^\star,\mathcal{D}_N) \leq 2\mathcal{R}_{\max}\cdot\sup_\pi\left|\mathbb{E}_{i\sim\mathcal{G}(\gamma)}\left[\sqrt{1-\exp\left(-\text{KL}\left(P^\star_{i+1,\pi}\|P^\pi_{i+1}(\mathcal{D}_N)\right)\right)}\right]\right|,$$

$$\leq 2\mathcal{R}_{\max}\cdot\sup_\pi\sqrt{1-\exp\left(-\frac{\mathcal{I}^\pi_N}{1-\gamma}\right)}.$$

$\square$

The PIL has an intuitive information-geometric interpretation: the inner expectation $\mathbb{E}_{s,a\sim\rho^\star_\pi}\left[\text{KL}\left(P^\star_{R,S}(s,a)\|P_{R,S}(s,a,\theta)\right)\right]$ measures the distance between the model and the true distribution in terms of the information lost when approximating $P^\star_{R,S}(s,a)$ with $P_{R,S}(s,a,\theta)$, averaged across all states. The PIL thus measures how close the posterior's belief is to the truth according to the average information lost under the posterior expectation. We observe that via Jensen's inequality, the PIL is an upper bound on the classic KL risk (sometimes known as expected relative entropy) from Bayesian asymptotics and regret analysis (Aitchison, 1975; Clarke and Barron, 1990; Komaki, 1996; Hartigan, 1998; Barron, 1988; 1999; Yang and Barron, 1999; van der Vaart and van Zanten, 2011; Aslan, 2006; Alaa and van der Schaar, 2018; Bilodeau et al., 2021).

By substituting in our definition of the Gaussian world model, we now find a convenient form for the PIL:

**Proposition 2.** *Using the Gaussian world model in Eq.* (2)*, it follows:*

$$\mathcal{I}^\pi_N = \mathcal{E}(\mathcal{D}_N,\mathcal{M}^\star) + \mathcal{V}(\mathcal{D}_N).$$

*Proof.* We substitute the Gaussian world model into the KL divergence to yield:

$$
\begin{aligned}
&\mathrm{KL}\left(P_{R,S}^{\star}(s,a)\|P_{R,S}(s,a,\theta)\right) \\
&=\mathbb{E}_{r,s'\sim P_{R,S}^{\star}(s,a)}\left[\log\left(\exp\left(-\frac{\|r^{\star}(s,a)-r\|_2^2}{2\sigma_r^2}\right)\exp\left(-\frac{\|s^{\star\prime}(s,a)-s'\|_2^2}{2\sigma_s^2}\right)\right)\right] \\
&\quad-\mathbb{E}_{r,s'\sim P_{R,S}^{\star}(s,a)}\left[\log\left(\exp\left(-\frac{\|r_\theta(s,a)-r\|_2^2}{2\sigma_r^2}\right)\exp\left(-\frac{\|s'_\theta(s,a)-s'\|_2^2}{2\sigma_s^2}\right)\right)\right], \\
&=\mathbb{E}_{r,s'\sim P_{R,S}^{\star}(s,a)}\left[\frac{\|r_\theta(s,a)-r\|_2^2-\|r^{\star}(s,a)-r\|_2^2}{2\sigma_r^2}\right. \\
&\quad\left.+\frac{\|s'_\theta(s,a)-s'\|_2^2-\|s^{\star\prime}(s,a)-s'\|_2^2}{2\sigma_s^2}\right], \\
&=\mathbb{E}_{r,s'\sim P_{R,S}^{\star}(s,a)}\left[\frac{r_\theta(s,a)^2-2rr_\theta(s,a)-r^{\star}(s,a)^2+2rr^{\star}(s,a)}{2\sigma_r^2}\right. \\
&\quad\left.+\frac{\|s'_\theta(s,a)\|_2^2-2s'^{\top}s'_\theta(s,a)-\|s^{\star\prime}(s,a)\|_2^2+2s'^{\top}s^{\star\prime}(s,a)}{2\sigma_s^2}\right], \\
&=\frac{r_\theta(s,a)^2-2r^{\star}(s,a)r_\theta(s,a)-r^{\star}(s,a)^2+2r^{\star}(s,a)^2}{2\sigma_r^2} \\
&\quad+\frac{\|s'_\theta(s,a)\|_2^2-2s^{\star\prime}(s,a)^{\top}s'_\theta(s,a)-\|s^{\star\prime}(s,a)\|_2^2+2\|s^{\star\prime}(s,a)\|_2^2}{2\sigma_s^2}, \\
&=\frac{r_\theta(s,a)^2-2r^{\star}(s,a)r_\theta(s,a)+r^{\star}(s,a)^2}{2\sigma_r^2} \\
&\quad+\frac{\|s'_\theta(s,a)\|_2^2-2s^{\star\prime}(s,a)^{\top}s'_\theta(s,a)+\|s^{\star\prime}(s,a)\|_2^2}{2\sigma_s^2}.
\end{aligned}
$$

Now, taking expectations with respect to the posterior:

$$
\begin{aligned}
&\mathbb{E}_{\theta\sim P_\Theta(\mathcal{D}_N)}\left[\mathrm{KL}\left(P_{R,S}^{\star}(s,a)\|P_{R,S}(s,a,\theta)\right)\right] \\
&\qquad=\mathbb{E}_{\theta\sim P_\Theta(\mathcal{D}_N)}\left[\frac{r_\theta(s,a)^2-2r^{\star}(s,a)r_\theta(s,a)+r^{\star}(s,a)^2}{2\sigma_r^2}\right] \\
&\qquad+\mathbb{E}_{\theta\sim P_\Theta(\mathcal{D}_N)}\left[\frac{\|s'_\theta(s,a)\|_2^2-2s^{\star\prime}(s,a)^{\top}s'_\theta(s,a)+\|s^{\star\prime}(s,a)\|_2^2}{2\sigma_s^2}\right], \\
&\qquad=\frac{\mathbb{E}_{\theta\sim P_\Theta(\mathcal{D}_N)}\left[r_\theta(s,a)^2\right]-2r^{\star}(s,a)r(s,a,\mathcal{D}_N)+r^{\star}(s,a)^2}{2\sigma_r^2} \\
&\qquad+\frac{\mathbb{E}_{\theta\sim P_\Theta(\mathcal{D}_N)}\left[\|s'_\theta(s,a)\|_2^2\right]-2s^{\star\prime}(s,a)^{\top}s'(s,a,\mathcal{D}_N)+\|s^{\star\prime}(s,a)\|_2^2}{2\sigma_s^2}.
\end{aligned}
$$

Now, we use the variance identity for both the reward and state functions: $\mathbb{E}_{\theta\sim P_\Theta(\mathcal{D}_N)}\left[r_\theta(s,a)^2\right]=\mathbb{V}_{\theta\sim P_\Theta(\mathcal{D}_N)}\left[r_\theta(s,a)\right]+r_\theta(s,a,\mathcal{D}_N)^2$ and $\mathbb{E}_{\theta\sim P_\Theta(\mathcal{D}_N)}\left[\|s'_\theta(s,a)\|_2^2\right]=\mathbb{V}_{\theta\sim P_\Theta(\mathcal{D}_N)}\left[\|s'_\theta(s,a)\|_2\right]+$

$\|s'(s, a, \mathcal{D}_N)\|_2^2$ yielding:

$$\mathbb{E}_{\theta \sim P_\Theta(\mathcal{D}_N)} \left[ \mathrm{KL} \left( P_{R,S}^\star(s, a) \| P_{R,S}(s, a, \theta) \right) \right]$$

$$= \frac{\mathbb{V}_{\theta \sim P_\Theta(\mathcal{D}_N)} \left[ r_\theta(s, a) \right] + r_\theta(s, a, \mathcal{D}_N)^2 - 2r^\star(s, a)r(s, a, \mathcal{D}_N) + r^\star(s, a)^2}{2\sigma_r^2}$$

$$+ \frac{\mathbb{V}_{\theta \sim P_\Theta(\mathcal{D}_N)} \left[ \|s'_\theta(s, a)\|_2 \right] + \|s'(s, a, \mathcal{D}_N)\|_2^2 - 2s^{\star\prime}(s, a)^\top s'(s, a, \mathcal{D}_N) + \|s^{\star\prime}(s, a)\|_2^2}{2\sigma_s^2},$$

$$= \frac{\mathbb{V}_{\theta \sim P_\Theta(\mathcal{D}_N)} \left[ r_\theta(s, a) \right] + (r_\theta(s, a, \mathcal{D}_N)^2 - r^\star(s, a))^2}{2\sigma_r^2}$$

$$+ \frac{\mathbb{V}_{\theta \sim P_\Theta(\mathcal{D}_N)} \left[ \|s'_\theta(s, a)\|_2 \right] + \|s'(s, a, \mathcal{D}_N) - s^{\star\prime}(s, a)\|_2^2}{2\sigma_s^2},$$

$$= \frac{\mathbb{V}_{\theta \sim P_\Theta(\mathcal{D}_N)} \left[ r_\theta(s, a) \right]}{2\sigma_r^2} + \frac{\mathbb{V}_{\theta \sim P_\Theta(\mathcal{D}_N)} \left[ \|s'_\theta(s, a)\|_2 \right]}{2\sigma_s^2}$$

$$+ \frac{(r_\theta(s, a, \mathcal{D}_N) - r^\star(s, a))^2}{2\sigma_r^2} + \frac{\|s'(s, a, \mathcal{D}_N) - s^{\star\prime}(s, a)\|_2^2}{2\sigma_s^2},$$

$$= \mathcal{E}(\mathcal{D}_N, \mathcal{M}^\star) + \mathcal{V}(\mathcal{D}_N),$$

and hence:

$$\mathcal{I}_N^\pi = \mathbb{E}_{\theta \sim P_\Theta(\mathcal{D}_N)} \left[ \mathrm{KL} \left( P_{R,S}^\star(s, a) \| P_{R,S}(s, a, \theta) \right) \right],$$

$$= \mathcal{E}(\mathcal{D}_N, \mathcal{M}^\star) + \mathcal{V}(\mathcal{D}_N).$$

$\square$

### D.3 Proof of Theorem 2

We first introduce some simplifying notation for the expected cross entropy, log likelihood and corresponding gradients and Hessian:

$$\ell(\theta) := \mathbb{E}_{s, a \sim \rho_\pi^\star, r, s' \sim P_{R,S}^\star(s, a)} \left[ \log p(r, s'|s, a, \theta) \right],$$

$$\ell^\star := \max_{\theta \in \Theta} \ell(\theta) = \mathbb{E}_{s, a \sim \rho_\pi^\star, r, s' \sim P_{R,S}^\star(s, a)} \left[ \log p^\star(r, s'|s, a) \right],$$

$$\ell_N(\theta) := \frac{1}{N} \sum_{i=0}^{N-1} \log p(r_i, s'_i|s_i, a_i, \theta),$$

$$g_{i,N}^\star := \sqrt{N} \nabla_\theta \ell_N(\theta) \big|_{\theta = \theta_i^\star},$$

$$H_i^\star := \nabla_\theta^2 \ell(\theta) \big|_{\theta = \theta_i^\star}.$$

We now introduce key regularity assumptions for our parametric model that are required to derive the convergence rate for PIL. They are relatively mild and commonplace in the asymptotic statistics literature (Le Cam, 1953; Barron, 1988; Clarke and Barron, 1990; Komaki, 1996; Hartigan, 1998; Barron, 1999; Aslan, 2006).

**Assumption 1.** *We assume that:*

*i There exists at least one parametrisation that corresponds to the true environment dynamics with:*

$$\left| \mathbb{E}_{s, a \sim \rho_\pi^\star, r, s' \sim P_{R,S}^\star(s, a)} \left[ \log p^\star(r, s'|s, a) \right] \right| < \infty$$

*and $|\ell^\star - \ell(\theta)|$ is bounded $P_\Theta$-almost surely.*

*ii $\ell_N(\theta)$ and $\ell(\theta)$ are $C^2$-continuous in $\theta$.*

*iii There are $K < \infty$ maximising points $\theta_i^\star$:*

$$\{\theta_1^\star, \theta_2^\star, \ldots \theta_K^\star\} = \arg\max_{\theta \in \Theta} \ell(\theta).$$

*For each maximiser $\theta_i^\star$, there exists a small region $\Theta_i^\star := \{\theta \in \Theta | \|\theta_i^\star - \theta\| \leq \epsilon\}$ for some $\epsilon > 0$ such that $\theta_i^\star$ is the unique maximiser in $\Theta_i^\star$, $\theta_i^\star$ is in the interior of $\Theta_i^\star$, $\nabla_\theta^2 \ell(\theta_i^\star)$ is negative definite, invertible and the regions are disjoint: $\bigcap_{i=1}^K \Theta_i^\star = \varnothing$.*

*iv The prior $p(\theta)$ is Lipschitz continuous in $\theta$ with support over $\Theta$.*

*v The sampling regime ensures that the strong law of large numbers holds for all maximisers $\theta_i^\star$ for the Hessian, and uniformly for $\theta \in \Theta$ for the likelihood, that is:*

$$\ell_N(\theta) \xrightarrow{Unif.\ a.s.} \ell(\theta), \quad \nabla_\theta^2 \ell_N(\theta_i^\star) \xrightarrow{a.s.} \nabla_\theta^2 \ell(\theta_i^\star).$$

*The central limit theorem applies to the gradient at each $\theta_i^\star$, that is:*

$$\sqrt{N} \nabla_\theta \ell_N(\theta_i^\star) \xrightarrow{d} \mathcal{N}(0, \Sigma_i^g),$$

*where $\Sigma_i^g = \mathbb{E}_{s,a \sim \rho_\pi^\star, r, s' \sim P_{R,S}^\star(s,a)} \left[ \nabla_\theta \log p(r, s'|s, a, \theta_i^\star) \nabla_\theta \log p(r, s'|s, a, \theta_i^\star)^\top \right]$ with $\|\Sigma_i^g\| < \infty$.*

Our assumptions are mild. Assumption 1i is our strictest assumption and is included for ease of exposition. We generalise our theory in Section D.4 to relax this assumption and also account for incomplete Bayes-optimal policy learning. Assumption 1ii ensures that a second order Taylor series expansion can be applied to obtain an asymptotic expansion around the maximising points. Assumption 1iii is much more general than most settings, which only consider problems with a single maximiser. The invertibility of the matrix can easily be guaranteed in Bayesian methods by the use of a prior that can re-condition a low rank matrix that may results from linearly dependent data. Assumption 1iv ensures that the prior places sufficient mass on the true parametrisation. The sampling and model would need to be very irregular for Assumption 1v not to hold; stochastic optimisation methods used to find statistics like the MAP will fail if this assumption did not hold. Assumption 1v holds automatically if sampling is either i.i.d. from $s, a \sim \rho_\pi^\star$ (see e.g. Bass (2013)) or from an aperiodic and irreducible Markov chain with stationary distribution $\rho_\pi^\star$ (see e.g. Roberts and Rosenthal (2004)). In both sample regimes, noting that $\mathbb{E}_{s,a \sim \rho_\pi^\star, r, s' \sim P_{R,S}^\star(s,a)} \left[ \nabla_\theta \log p(r, s'|s, a, \theta_i^\star) \right] = 0$, it's clear the (long run) covariance of $\nabla_\theta \log p(r, s'|s, a, \theta_i^\star)$ is $\Sigma_i^g$.

Our first lemma borrows techniques from Vaart (1998, Chapter 10). This approach is similar to asymptotic integral expansion approaches that apply Laplace's method (Lindley, 1980; Tierney and Kadane, 1986; Tierney et al., 1989; Kass et al., 1990) except we expand around the global maximising values of $\ell(\theta)$ rather than the maximising values of the likelihood $\ell_N(\theta)$ to obtain an asymptotic expression for the posterior:

**Lemma 3.** *Under Assumption 1 and using the notation introduced at the start of Section D.3:*

$$\frac{\int_{\Theta_i^\star} (\ell^\star - \ell(\theta) \exp(N\ell_N(\theta))) p(\theta) d\theta}{\int_{\Theta_i^\star} \exp(N\ell_N(\theta)) p(\theta) d\theta} = \mathcal{O}\left( \frac{d - g_{i,N}^{\star\top} H_i^{\star-1} g_{i,N}^\star}{N} \right),$$

*almost surely.*

*Proof.* We start by applying the transformation of variables $\theta' = f(\theta) := \sqrt{N}(\theta - \theta_i^\star)$ to integrals in the numerator and denominator with:

$$\theta = f^{-1}(\theta') = \theta_i^\star + \frac{1}{\sqrt{N}}\theta', \quad \left|\det \nabla_\theta f^{-1}(\theta')\right| = N^{-\frac{d}{2}}, \quad \Theta' := f(\Theta_i^\star),$$

yielding:

$$\frac{\int_{\Theta_i^\star} (\ell^\star - \ell(\theta)) \exp(N\ell_N(\theta)) p(\theta) d\theta}{\int_{\Theta^\star} \exp(N\ell_N(\theta)) p(\theta) d\theta}$$

$$= \frac{\int_{\Theta'} \left(\ell^\star - \ell\left(\theta = \theta^\star + \frac{1}{\sqrt{N}}\theta'\right)\right) \exp\left(N\ell_N\left(\theta = \theta_i^\star + \frac{1}{\sqrt{N}}\theta'\right)\right) p'(\theta') d\theta'}{\int_{\Theta'} \exp\left(N\ell_N\left(\theta = \theta_i^\star + \frac{1}{\sqrt{N}}\theta'\right)\right) p'(\theta') d\theta'}, \quad (15)$$

where $p'(\theta') := p\left(\theta = \theta_i^\star + \frac{1}{\sqrt{N}}\theta'\right)$. Now and making a Taylor series expansion of $\ell(\theta)$ about $\theta_i^\star$:

$$\ell(\theta) = \ell^\star + \underbrace{\nabla_\theta \ell(\theta_i^\star)}_{=0}^\top (\theta - \theta_i^\star) + (\theta - \theta_i^\star)^\top H_i^\star (\theta - \theta_i^\star) + \mathcal{O}\left(\|\theta - \theta_i^\star\|^3\right),$$

$$= \ell^\star + (\theta - \theta_i^\star)^\top H_i^\star (\theta - \theta_i^\star) + \mathcal{O}\left(\|\theta - \theta_i^\star\|^3\right),$$

hence:

$$\ell\left(\theta = \theta_i^\star + \frac{1}{\sqrt{N}}\theta'\right) = \ell^\star + \frac{1}{N}\theta'^\top H_i^\star \theta' + \mathcal{O}\left(N^{-\frac{3}{2}}\right).$$

Using the notation $H_N^\star := \nabla_\theta^2 \ell_N(\theta)\big|_{\theta=\theta_i^\star}$ and making a Taylor series expansion of $\ell_N(\theta)$ about $\theta_i^\star$:

$$\ell_N(\theta) = \ell_N(\theta_i^\star) + \nabla_\theta \ell_N(\theta_i^\star)^\top (\theta - \theta_i^\star) + (\theta - \theta_i^\star)^\top \nabla_\theta^2 \ell_N(\theta_i^\star)(\theta - \theta_i^\star) + \mathcal{O}\left(\|\theta - \theta_i^\star\|^3\right),$$

hence:

$$N\ell_N\left(\theta = \theta_i^\star + \frac{1}{\sqrt{N}}\theta'\right) = N\ell_N(\theta_i^\star) + \sqrt{N}\nabla_\theta \ell_N(\theta_i^\star)^\top \theta' + \theta'^\top \nabla_\theta^2 \ell_N(\theta_i^\star)\theta' + \mathcal{O}\left(\frac{1}{\sqrt{N}}\right).$$

Substituting into Eq. (15) yields:

$$\frac{\int_{\Theta^\star} (\ell^\star - \ell(\theta)) \exp(N\ell(\theta)) p(\theta)d\theta}{\int_{\Theta^\star} \exp(N\ell(\theta)) p(\theta)d\theta}$$

$$= -\frac{\int_{\Theta'} \theta'^\top H_i^\star \theta' \exp\left(N\ell_N(\theta_i^\star) + {g_{i,N}^\star}^\top \theta' + \theta'^\top H_N^\star \theta' + \mathcal{O}\left(\frac{1}{\sqrt{N}}\right)\right) p'(\theta')d\theta'}{\int_{\Theta'} \exp\left(N\ell_N(\theta_i^\star) + {g_{i,N}^\star}^\top \theta' + \theta'^\top H_N^\star \theta' + \mathcal{O}\left(\frac{1}{\sqrt{N}}\right)\right) p'(\theta')d\theta'}\mathcal{O}\left(\frac{1}{N}\right),$$

$$= -\frac{\int_{\Theta'} \theta'^\top H_i^\star \theta' \exp\left({g_{i,N}^\star}^\top \theta' + \theta'^\top H_N^\star \theta' + \mathcal{O}\left(\frac{1}{\sqrt{N}}\right)\right) p'(\theta')d\theta'}{\int_{\Theta'} \exp\left(N{g_{i,N}^\star}^\top \theta' + \theta'^\top H_N^\star \theta' + \mathcal{O}\left(\frac{1}{\sqrt{N}}\right)\right) p'(\theta')d\theta'}\mathcal{O}\left(\frac{1}{N}\right),$$

$$= -\frac{\int_{\Theta'} \theta'^\top H_i^\star \theta' \exp\left({g_{i,N}^\star}^\top \theta' + \theta'^\top H_N^\star \theta'\right) \exp\left(\mathcal{O}\left(\frac{1}{\sqrt{N}}\right)\right) p'(\theta')d\theta'}{\int_{\Theta'} \exp\left(N{g_{i,N}^\star}^\top \theta' + \theta'^\top H_N^\star \theta'\right) \exp\left(\mathcal{O}\left(\frac{1}{\sqrt{N}}\right)\right) p'(\theta')d\theta'}\mathcal{O}\left(\frac{1}{N}\right),$$

$$= \mathcal{O}\left(-\frac{1}{N}\frac{\int_{\Theta'} \theta'^\top H_i^\star \theta' \exp\left({g_{i,N}^\star}^\top \theta' + \theta'^\top H_N^\star \theta'\right) p'(\theta')d\theta'}{\int_{\Theta'} \exp\left(N{g_{i,N}^\star}^\top \theta' + \theta'^\top H_N^\star \theta'\right) p'(\theta')d\theta'}\right) \tag{16}$$

where we have multiplied top and bottom by $\exp\left(-N\ell_N(\theta_i^\star)\right)$ to derive the second equality and used the fact that $0 < \exp\left(\mathcal{O}\left(\frac{1}{\sqrt{N}}\right)\right) < \infty$ to derive the final line. Now, as the prior is Lipschitz, we make a Taylor series expansion about $\theta_i^\star$:

$$p(\theta) = p(\theta_i^\star) + \mathcal{O}\left(\|\theta - \theta_i^\star\|\right),$$

hence:

$$p'(\theta') = p\left(\theta = \theta_i^\star + \frac{1}{\sqrt{N}}\theta'\right) = p(\theta_i^\star) + \mathcal{O}\left(\frac{1}{\sqrt{N}}\right).$$

This allows us to find an asymptotic expression for Eq. (16):

$$\frac{\int_{\Theta'} \theta'^\top H_i^\star \theta' \exp\left({g_{i,N}^\star}^\top \theta' + \theta'^\top H_N^\star \theta'\right) p'(\theta')d\theta'}{\int_{\Theta'} \exp\left({g_{i,N}^\star}^\top \theta' + \theta'^\top H_N^\star \theta'\right) p'(\theta')d\theta'}$$

$$= \frac{\int_{\Theta'} \theta'^\top H_i^\star \theta' \exp\left({g_{i,N}^\star}^\top \theta' + \theta'^\top H_N^\star \theta'\right) p'(\theta_i^\star)d\theta'\left(1 + \mathcal{O}\left(\frac{1}{\sqrt{N}}\right)\right)}{\int_{\Theta'} \exp\left({g_{i,N}^\star}^\top \theta' + \theta'^\top H_N^\star \theta'\right) p'(\theta_i^\star)d\theta'\left(1 + \mathcal{O}\left(\frac{1}{\sqrt{N}}\right)\right)}$$

$$= \frac{\int_{\Theta'} \theta'^\top H_i^\star \theta' \exp\left({g_{i,N}^\star}^\top \theta' + \theta'^\top H_N^\star \theta'\right) p'(\theta_i^\star)d\theta'}{\int_{\Theta'} \exp\left({g_{i,N}^\star}^\top \theta' + \theta'^\top H_N^\star \theta'\right) p'(\theta_i^\star)d\theta'}\left(1 + \mathcal{O}\left(\frac{1}{\sqrt{N}}\right)\right)$$

$$= \frac{\int_{\Theta'} \theta'^\top H_i^\star \theta' \exp\left({g_{i,N}^\star}^\top \theta' + \theta'^\top H_N^\star \theta'\right) d\theta'}{\int_{\Theta'} \exp\left({g_{i,N}^\star}^\top \theta' + \theta'^\top H_N^\star \theta'\right) d\theta'}\left(1 + \mathcal{O}\left(\frac{1}{\sqrt{N}}\right)\right).$$

We re-write the exponential term to recover a quadratic form:

$$\exp\left(g_{i,N}^{\star}{}^{\top}\theta' + \theta'^{\top}H_N^{\star}\theta'\right)$$

$$= \exp\left(\left(\frac{1}{2}H_N^{\star}{}^{-1}g_{i,N}^{\star} + \theta'\right)^{\top}H_N^{\star}\left(\frac{1}{2}H_N^{\star}{}^{-1}g_{i,N}^{\star} + \theta'\right) - \frac{1}{4}g_{i,N}^{\star}{}^{\top}H_N^{\star}{}^{-1}g_{i,N}^{\star}\right).$$

Substituting yields:

$$\frac{\int_{\Theta'} \theta'^{\top}H_i^{\star}\theta' \exp\left(g_{i,N}^{\star}{}^{\top}\theta' + \theta'^{\top}H_N^{\star}\theta'\right)d\theta'}{\int_{\Theta'} \exp\left(g_{i,N}^{\star}{}^{\top}\theta' + \theta'^{\top}H_N^{\star}\theta'\right)d\theta'}$$

$$= \frac{\int_{\Theta'} \theta'^{\top}H_i^{\star}\theta' \exp\left(\left(\frac{1}{2}H_N^{\star}{}^{-1}g_{i,N}^{\star} + \theta'\right)^{\top}H_N^{\star}\left(\frac{1}{2}H_N^{\star}{}^{-1}g_{i,N}^{\star} + \theta'\right)\right)d\theta'}{\int_{\Theta'} \exp\left(\left(\frac{1}{2}H_N^{\star}{}^{-1}g_{i,N}^{\star} + \theta'\right)^{\top}H_N^{\star}\left(\frac{1}{2}H_N^{\star}{}^{-1}g_{i,N}^{\star} + \theta'\right)\right)d\theta'}.$$

In this form, we notice the expectation is that of a Gaussian $\mathcal{N}\left(\mu = -\frac{1}{2}H_N^{\star}{}^{-1}g_{i,N}^{\star}, \Sigma = -H_i^{\star}{}^{-1}\right)$ restricted to $\Theta'$. Noting that in the limit $\Theta' \xrightarrow[N\to\infty]{} \mathbb{R}^d$, hence:

$$\frac{\int_{\Theta'} \theta'^{\top}H_i^{\star}\theta' \exp\left(\left(\theta' + \frac{1}{2}H_N^{\star}{}^{-1}g_{i,N}^{\star}\right)^{\top}H_N^{\star}\left(\theta' + \frac{1}{2}H_N^{\star}{}^{-1}g_{i,N}^{\star}\right)\right)d\theta'}{\int_{\Theta'} \exp\left(\left(\theta' + \frac{1}{2}H_N^{\star}{}^{-1}g_{i,N}^{\star}\right)^{\top}H_N^{\star}\left(\theta' + \frac{1}{2}H_N^{\star}{}^{-1}g_{i,N}^{\star}\right)\right)d\theta'}$$

$$= \mathcal{O}\left(\frac{\int_{\mathbb{R}^d} \theta'^{\top}H_i^{\star}\theta' \exp\left(\left(\theta' + \frac{1}{2}H_N^{\star}{}^{-1}g_{i,N}^{\star}\right)^{\top}H_N^{\star}\left(\theta' + \frac{1}{2}H_N^{\star}{}^{-1}g_{i,N}^{\star}\right)\right)d\theta'}{\int_{\mathbb{R}^d} \exp\left(\left(\theta' + \frac{1}{2}H_N^{\star}{}^{-1}g_{i,N}^{\star}\right)^{\top}H_N^{\star}\left(\theta' + \frac{1}{2}H_N^{\star}{}^{-1}g_{i,N}^{\star}\right)\right)d\theta'}\right),$$

$$= \mathcal{O}\left(\mathbb{E}_{\theta'\sim\mathcal{N}\left(-\frac{1}{2}H_N^{\star}{}^{-1}g_{i,N}^{\star}, -H_N^{\star}{}^{-1}\right)}\left[\theta'^{\top}H_i^{\star}\theta'\right]\right). \tag{17}$$

Putting everything together, we have:

$$\frac{\int_{\Theta_i^{\star}} \left(\ell^{\star} - \ell(\theta)\exp\left(N\ell_N(\theta)\right)\right)p(\theta)d\theta}{\int_{\Theta_i^{\star}} \exp\left(N\ell_N(\theta)\right)p(\theta)d\theta}$$

$$= \mathcal{O}\left(-\frac{1}{N}\frac{\int_{\Theta'} \theta'^{\top}H_i^{\star}\theta' \exp\left(g_{i,N}^{\star}{}^{\top}\theta' + \theta'^{\top}H_N^{\star}\theta'\right)p'(\theta')d\theta'}{\int_{\Theta'} \exp\left(g_{i,N}^{\star}{}^{\top}\theta' + \theta'^{\top}H_N^{\star}\theta'\right)p'(\theta')d\theta'}\right), \quad \text{Eq. (16)}$$

$$= \mathcal{O}\left(-\frac{1}{N}\mathbb{E}_{\theta'\sim\mathcal{N}\left(-\frac{1}{2}H_N^{\star}{}^{-1}g_{i,N}^{\star}, -H_i^{\star}{}^{-1}\right)}\left[\theta'^{\top}H_i^{\star}\theta'\right]\right), \quad \text{Eq. (17)}$$

Using standard results for the multivariate Gaussian (Petersen and Pedersen, 2012) yields our desired result:

$$-\frac{1}{N}\mathbb{E}_{\theta'\sim\mathcal{N}\left(-\frac{1}{2}H_N^{\star}{}^{-1}g_{i,N}^{\star}, -H_i^{\star}{}^{-1}\right)}\left[\theta'^{\top}H_i^{\star}\theta'\right] = \frac{\text{Tr}\left(H_i^{\star}H_N^{\star}{}^{-1}\right) - \frac{1}{4}g_{i,N}^{\star}{}^{\top}H_N^{\star}{}^{-1}{}^{\top}H_i^{\star}H_N^{\star}{}^{-1}g_{i,N}^{\star}}{N},$$

$$= \mathcal{O}\left(\frac{\text{Tr}\left(I\right) - g_{i,N}^{\star}{}^{\top}H_i^{\star}{}^{-1}g_{i,N}^{\star}}{N}\right),$$

$$= \mathcal{O}\left(\frac{d - g_{i,N}^{\star}{}^{\top}H_i^{\star}{}^{-1}g_{i,N}^{\star}}{N}\right),$$

almost surely, where we have used the strong law of large numbers on the empirical Hessian from Assumption 1 to derive the second line. $\qquad\square$

In our final Lemma, we show that regions that are not close to the maximising points diminish exponentially in posterior probability as $N$ grows large.

**Lemma 4.** *Under Assumption 1, $\mathbb{E}_{\mathcal{D}_N \sim P_{Data}^\star}\left[P(\bar{\Theta}|\mathcal{D}_N)\right] = \mathcal{O}\left(\exp(-N)\right)$.*

*Proof.* We start by splitting the posterior expectation into integrals over $\bar{\Theta}$ and $\Theta \setminus \bar{\Theta}$:

$$P(\bar{\Theta}|\mathcal{D}_N) = \frac{\int_{\bar{\Theta}} \exp\left(N\ell_N(\theta)\right) p(\theta) d\theta}{\int_{\Theta} \exp\left(N\ell_N(\theta)\right) p(\theta) d\theta},$$

$$= \frac{\int_{\bar{\Theta}} \exp\left(N\ell_N(\theta)\right) p(\theta) d\theta}{\int_{\bar{\Theta}} \exp\left(N\ell_N(\theta)\right) p(\theta) d\theta + \int_{\Theta \setminus \bar{\Theta}} \exp\left(N\ell_N(\theta)\right) p(\theta) d\theta}.$$

Dividing top and bottom by $\int_{\bar{\Theta}} \exp\left(N\ell_N(\theta)\right) p(\theta) d\theta$:

$$P(\bar{\Theta}|\mathcal{D}_N) = \frac{1}{1 + \frac{\int_{\Theta \setminus \bar{\Theta}} \exp(N\ell_N(\theta)) p(\theta) d\theta}{\int_{\bar{\Theta}} \exp(N\ell_N(\theta)) p(\theta) d\theta}}.$$

Hence if we can show there exists some $N' < \infty$ and a function $C \exp(cN)$ with positive constants $c$ and $C$ that lower bounds the ratio:

$$C \exp(cN) \leq \frac{\int_{\Theta \setminus \bar{\Theta}} \exp\left(N\ell_N(\theta)\right) p(\theta) d\theta}{\int_{\bar{\Theta}} \exp\left(N\ell_N(\theta)\right) p(\theta) d\theta}$$

almost surely for all $N \geq N'$, then it follows:

$$\mathbb{E}_{\mathcal{D}_N \sim P_{Data}^\star}\left[P(\bar{\Theta}|\mathcal{D}_N)\right] = \mathcal{O}\left(\frac{1}{1 + \exp(N)}\right),$$

$$= \mathcal{O}\left(\exp(-N)\right).$$

From Assumption 1, each $\theta_i^\star$ maximises $\ell(\theta)$ with $\sup_{\theta \in \bar{\Theta}} \ell(\theta') < \ell(\theta_i^\star)$. As $\ell(\theta)$ is continuous, there thus exists a small, closed ball $B(\theta_j^\star, r) := \{\theta | \|\theta_i^\star - \theta\| \leq r\}$ of radius $r > 0$ centred on some $\theta_j^\star$ such that $\sup_{\theta' \in \bar{\Theta}} \ell(\theta') < \min_{\theta'' \in B(\theta_j^\star, r)} \ell(\theta'')$. From Assumption 1, the uniform strong law of large numbers holds with $\ell_N(\theta) \xrightarrow{\text{Unif. } a.s} \ell(\theta)$. By the definition of the limit and continuity of $\ell_N(\theta)$, there thus exists some finite $N'$ such that $\sup_{\theta' \in \bar{\Theta}} \ell_N(\theta') < \min_{\theta'' \in B(\theta_j^\star, \frac{r}{2})} \ell_N(\theta'')$ for all $N \geq N'$ almost surely, where $B(\theta_j^\star, \frac{r}{2})$ is a ball of half radius $\frac{r}{2}$. Noting that $B(\theta_j^\star, \frac{r}{2}) \subset \Theta \setminus \bar{\Theta}$ and $0 \leq \exp\left(N\ell_N(\theta)\right)$, this allows us to lower bound the integral:

$$\int_{\Theta \setminus \bar{\Theta}} \exp\left(N\ell_N(\theta)\right) p(\theta) d\theta \geq \int_{B(\theta_j^\star, \frac{r}{2})} \exp\left(N\ell_N(\theta)\right) p(\theta) d\theta,$$

$$\geq \exp\left(N \min_{\theta'' \in B(\theta_j^\star, \frac{r}{2})} \ell_N(\theta'')\right) \int_{B(\theta_j^\star, \frac{r}{2})} p(\theta) d\theta,$$

$$= \exp\left(N \min_{\theta'' \in B(\theta_j^\star, \frac{r}{2})} \ell_N(\theta'')\right) P\left(B\left(\theta_j^\star, \frac{r}{2}\right)\right).$$

We can also upper bound the integral:

$$\int_{\bar{\Theta}} \exp\left(N\ell_N(\theta)\right) p(\theta) d\theta \leq \exp\left(N \sup_{\theta' \in \bar{\Theta}} \ell_N(\theta')\right) \int_{\bar{\Theta}} p(\theta) d\theta,$$

$$= \exp\left(N \sup_{\theta' \in \bar{\Theta}} \ell_N(\theta')\right) P\left(\bar{\Theta}\right).$$

Using these results, we lower bound the ratio as:

$$\frac{\int_{\Theta \setminus \bar{\Theta}} \exp\left(N\ell_N(\theta)\right) p(\theta) d\theta}{\int_{\bar{\Theta}} \exp\left(N\ell_N(\theta)\right) p(\theta) d\theta} \geq \frac{\exp\left(N \min_{\theta'' \in B(\theta_j^\star, \frac{r}{2})} \ell_N(\theta'')\right) P\left(B\left(\theta_j^\star, \frac{r}{2}\right)\right)}{\exp\left(N \sup_{\theta' \in \bar{\Theta}} \ell_N(\theta')\right) P\left(\bar{\Theta}\right)},$$

$$= \exp\left(N\left(\min_{\theta'' \in B(\theta_j^\star, \frac{r}{2})} \ell_N(\theta'') - \sup_{\theta' \in \bar{\Theta}} \ell_N(\theta')\right)\right) \frac{P\left(B\left(\theta_j^\star, \frac{r}{2}\right)\right)}{P\left(\bar{\Theta}\right)}.$$

Let $\frac{P\left(B(\theta_j^\star, \frac{r}{2})\right)}{P(\bar{\Theta})} = C > 0$ from Assumption 1. As there exists some $N'$ such that $\min_{\theta'' \in B(\theta_j^\star, \frac{r}{2})} \ell_N(\theta'') > \sup_{\theta' \in \bar{\Theta}} \ell_N(\theta')$ for all $N > N'$, we have shown exists some positive constants $c > 0$ and $C > 0$ such that

$$C \exp\left(cN\right) \leq \frac{\int_{\Theta \setminus \bar{\Theta}} \exp\left(N\ell_N(\theta)\right) p(\theta) d\theta}{\int_{\bar{\Theta}} \exp\left(N\ell_N(\theta)\right) p(\theta) d\theta},$$

for all $N > N'$ almost surely, as required. $\qquad\square$

We now present our proof of Theorem 2. Here we split the posterior expectation up into small regions close to maximising points and regions away from maximising. We then apply our two lemmas to each region. Our result then follows by an application the central limit theorem under Assumption 1.

**Theorem 2.** *Let the data be drawn from the underlying true distribution $\mathcal{D}_N \sim P_{Data}^\star$. Under Assumption 1, there exists some constant $0 < C < \infty$ such that for sufficiently large $N$:*

$$\mathbb{E}_{\mathcal{D}_N \sim P_{Data}^\star}\left[\text{Regret}(\mathcal{M}^\star, \mathcal{D}_N)\right] \leq 2\mathcal{R}_{\max} \cdot \sqrt{1 - \exp\left(-\frac{Cd}{(1-\gamma)N}\right)}. \tag{18}$$

*Proof.* Using the notation introduced at the start of Section D.3, we write the PIL as:

$$\mathcal{I}_N^\pi := \mathbb{E}_{s,a \sim \rho_\pi^\star}\left[\mathbb{E}_{\theta \sim P_\Theta(\mathcal{D}_N)}\left[\text{KL}\left(P_{R,S}^\star(s,a) \| P_{R,S}(s,a,\theta)\right)\right]\right],$$

$$= \mathbb{E}_{\theta \sim P_\Theta(\mathcal{D}_N)}\left[\mathbb{E}_{s,a \sim \rho_\pi^\star, r, s' \sim P_{R,S}^\star(s,a)}\left[\log p(r,s'|s,a,\theta^\star) - \log p(r,s'|s,a,\theta)\right]\right],$$

$$= \mathbb{E}_{\theta \sim P_\Theta(\mathcal{D}_N)}\left[\ell^\star - \ell(\theta)\right],$$

Under this same notation, we write the posterior density as:

$$p(\theta|\mathcal{D}_N) = \frac{\exp\left(N\ell_N(\theta)\right) p(\theta)}{\int_\Theta \exp\left(N\ell_N(\theta)\right) p(\theta) d\theta}. \tag{19}$$

Now, under Assumption 1, we split the inner expectation into small regions $\Theta_i^\star$ around each maximising point $\theta_i^\star$ and the remainder of the parameter space $\bar{\Theta} := \Theta \setminus \bigcup_{i=1}^K \Theta_i^\star$:

$$\mathbb{E}_{\theta \sim P_\Theta(\mathcal{D}_N)}\left[\ell^\star - \ell(\theta)\right] = \sum_{i=1}^K \int_{\Theta_i^\star} \left(\ell^\star - \ell(\theta)\right) p(\theta|\mathcal{D}_N) d\theta + \int_{\bar{\Theta}} \left(\ell^\star - \ell(\theta)\right) p(\theta|\mathcal{D}_N) d\theta. \tag{20}$$

Using Eq. (19), we now re-write each integral in the summation term of Eq. (20) as:

$$\int_{\Theta_i^\star} \left(\ell^\star - \ell(\theta)\right) p(\theta|\mathcal{D}_N) d\theta = \frac{\int_{\Theta_i^\star} \left(\ell^\star - \ell(\theta)\right) \exp\left(N\ell_N(\theta)\right) p(\theta) d\theta}{\int_\Theta \exp\left(N\ell_N(\theta)\right) p(\theta) d\theta},$$

$$= \frac{\int_{\Theta_i^\star} \left(\ell^\star - \ell(\theta)\right) \exp\left(N\ell_N(\theta)\right) p(\theta) d\theta}{\int_\Theta \exp\left(N\ell_N(\theta)\right) p(\theta) d\theta} \cdot \frac{\int_{\Theta_i^\star} \exp\left(N\ell_N(\theta)\right) p(\theta) d\theta}{\int_{\Theta_i^\star} \exp\left(N\ell_N(\theta)\right) p(\theta) d\theta},$$

$$= \frac{\int_{\Theta_i^\star} \left(\ell^\star - \ell(\theta)\right) \exp\left(N\ell_N(\theta)\right) p(\theta) d\theta}{\int_{\Theta_i^\star} \exp\left(N\ell_N(\theta)\right) p(\theta) d\theta} \cdot \frac{\int_{\Theta_i^\star} \exp\left(N\ell_N(\theta)\right) p(\theta) d\theta}{\int_\Theta \exp\left(N\ell_N(\theta)\right) p(\theta) d\theta},$$

$$= \frac{\int_{\Theta_i^\star} \left(\ell^\star - \ell(\theta)\right) \exp\left(N\ell_N(\theta)\right) p(\theta) d\theta}{\int_{\Theta_i^\star} \exp\left(N\ell_N(\theta)\right) p(\theta) d\theta} \cdot P(\Theta_i^\star|\mathcal{D}_N),$$

$$\leq \frac{\int_{\Theta_i^\star} \left(\ell^\star - \ell(\theta)\right) \exp\left(N\ell_N(\theta)\right) p(\theta) d\theta}{\int_{\Theta_i^\star} \exp\left(N\ell_N(\theta)\right) p(\theta) d\theta}. \tag{21}$$

where we have used $0 \leq P(\Theta_i^\star|\mathcal{D}_N) \leq 1$ from Kolmogorov's axioms to bound the final line.

For the last term in Eq. (20), we note that $\ell^\star - \ell(\theta)$ is bounded $P_\Theta$-almost surely from Assumption 1, hence there exists some $\ell^\dagger < \infty$ such that:

$$\int_{\bar{\Theta}} \left(\ell^\star - \ell(\theta)\right) p(\theta|\mathcal{D}_N) d\theta \leq \int_{\bar{\Theta}} \ell^\dagger p(\theta|\mathcal{D}_N) d\theta,$$

$$= \ell^\dagger P(\bar{\Theta}|\mathcal{D}_N). \tag{22}$$

Using Ineqs. 21 and 22, we bound Eq. (20) as:

$$\mathbb{E}_{\theta \sim P_\Theta(\mathcal{D}_N)}\left[\ell^\star - \ell(\theta)\right] \le \sum_{i=1}^K \frac{\int_{\Theta_i^\star} (\ell^\star - \ell(\theta)) \exp\left(N\ell_N(\theta)\right) p(\theta) d\theta}{\int_{\Theta_i^\star} \exp\left(N\ell_N(\theta)\right) p(\theta) d\theta} + \ell^\dagger P(\bar{\Theta}|\mathcal{D}_N),$$

and hence the PIL can be bounded as:

$$\mathcal{I}_N^\pi \le \sum_{i=1}^K \frac{\int_{\Theta_i^\star} (\ell^\star - \ell(\theta)) \exp\left(N\ell_N(\theta)\right) p(\theta) d\theta}{\int_{\Theta_i^\star} \exp\left(N\ell_N(\theta)\right) p(\theta) d\theta} + \ell^\dagger P(\bar{\Theta}|\mathcal{D}_N).$$

Applying Lemma 3 and Lemma 4 under Assumption 1 yields:

$$\mathcal{I}_N^\pi = \sum_{i=1}^K \mathcal{O}\left(\frac{d - {g_{i,N}^\star}^\top H_i^{\star-1} g_{i,N}^\star}{N}\right) + \ell^\dagger \mathcal{O}\left(\exp(-N)\right),$$

$$= \mathcal{O}\left(\frac{d - \sum_{i=1}^K {g_{i,N}^\star}^\top H_i^{\star-1} g_{i,N}^\star}{N}\right). \tag{23}$$

almost surely. As $f(x) := 2\mathcal{R}_{\max} \cdot \sqrt{1 - \exp\left(-\frac{x}{(1-\gamma)}\right)}$ is monotonic in $x$ and $\frac{d - \sum_{i=1}^K {g_{i,N}^\star}^\top H_i^{\star-1} g_{i,N}^\star}{N} \ge 0$, Eq. (23) implies there exists some positive $0 < C < \infty$ such that:

$$\text{Regret}(\mathcal{M}^\star, \mathcal{D}_N) \le 2\mathcal{R}_{\max} \cdot \sqrt{1 - \exp\left(-C \frac{d - \sum_{i=1}^K {g_{i,N}^\star}^\top H_i^{\star-1} g_{i,N}^\star}{(1-\gamma)N}\right)},$$

almost surely for large enough $N$. Under Assumption 1, $g_{i,N}^\star \xrightarrow{d} \mathcal{N}(0, \Sigma_i^g)$. As $f(x)$ is also a bounded, continuous function and concave, we can apply the Portmanteau Theorem (see for example Bass (2013, Chapter 21.7)) followed by Jensen's inequality to yield:

$$\mathbb{E}_{\mathcal{D}_N \sim P_{\text{Data}}^\star}\left[\text{Regret}(\mathcal{M}^\star, \mathcal{D}_N)\right] \le 2\mathcal{R}_{\max} \cdot \mathbb{E}_{g_i \sim \mathcal{N}(0, \Sigma_i^g)}\left[\sqrt{1 - \exp\left(-C \frac{d - \sum_{i=1}^K g_i^\top H_i^{\star-1} g_i}{(1-\gamma)N}\right)}\right],$$

$$\le 2\mathcal{R}_{\max} \cdot \sqrt{1 - \exp\left(-C \frac{d - \sum_{i=1}^K \mathbb{E}_{g_i \sim \mathcal{N}(0, \Sigma_i^g)}\left[g^\top H_i^{\star-1} g\right]}{(1-\gamma)N}\right)},$$

$$= 2\mathcal{R}_{\max} \cdot \sqrt{1 - \exp\left(-C \frac{d - \sum_{i=1}^K \text{Tr}\left(\Sigma_i^g H_i^{\star-1}\right)}{(1-\gamma)N}\right)}. \tag{24}$$

Now, examining the Hessian:

$$H(\theta) = \nabla_\theta^2 \mathbb{E}_{s,a \sim \rho_\pi^\star, r, s' \sim P_{R,S}^\star(s,a)}\left[\log p(r, s'|s, a, \theta)\right]$$

$$= \nabla_\theta \mathbb{E}_{s,a \sim \rho_\pi^\star, r, s' \sim P_{R,S}^\star(s,a)}\left[\nabla_\theta \log p(r, s'|s, a, \theta)\right],$$

$$= \nabla_\theta \mathbb{E}_{s,a \sim \rho_\pi^\star, r, s' \sim P_{R,S}^\star(s,a)}\left[\frac{\nabla_\theta p(r, s'|s, a, \theta)}{p(r, s'|s, a, \theta)}\right],$$

$$= \mathbb{E}_{s,a \sim \rho_\pi^\star, r, s' \sim P_{R,S}^\star(s,a)}\left[\nabla_\theta \frac{\nabla_\theta p(r, s'|s, a, \theta)}{p(r, s'|s, a, \theta)}\right],$$

$$= \mathbb{E}_{s,a \sim \rho_\pi^\star, r, s' \sim P_{R,S}^\star(s,a)}\left[\frac{\nabla_\theta^2 p(r, s'|s, a, \theta)}{p(r, s'|s, a, \theta)}\right]$$

$$- \mathbb{E}_{s,a \sim \rho_\pi^\star, r, s' \sim P_{R,S}^\star(s,a)}\left[\frac{\nabla_\theta p(r, s'|s, a, \theta)}{p(r, s'|s, a, \theta)} \frac{\nabla_\theta p(r, s'|s, a, \theta)^\top}{p(r, s'|s, a, \theta)}\right], \tag{25}$$

Hence at $\theta = \theta_i^\star$, the first term of Eq. (25) is:

$$\mathbb{E}_{s,a\sim\rho_\pi^\star,r,s'\sim P_{R,S}^\star(s,a)}\left[\frac{\nabla_\theta^2 p(r,s'|s,a,\theta)}{p(r,s'|s,a,\theta_i^\star)}\right] = \mathbb{E}_{s,a\sim\rho_\pi^\star,r,s'\sim P_{R,S}^\star(s,a)}\left[\frac{\nabla_\theta^2 p(r,s'|s,a,\theta)|_{\theta=\theta_i^\star}}{p^\star(r,s'|s,a)}\right],$$

$$= \mathbb{E}_{s,a\sim\rho_\pi^\star}\left[\int_{\mathbb{R}\times\mathcal{S}} \nabla_\theta^2 p(r,s'|s,a,\theta)|_{\theta=\theta_i^\star} d(r,s')\right],$$

$$= \mathbb{E}_{s,a\sim\rho_\pi^\star}\left[\nabla_\theta^2 \int_{\mathbb{R}\times\mathcal{S}} p(r,s'|s,a,\theta)d(r,s')|_{\theta=\theta_i^\star}\right],$$

$$= \nabla_\theta^2 1|_{\theta=\theta_i^\star},$$

$$= 0,$$

hence:

$$H(\theta_i^\star) = 0 - \mathbb{E}_{s,a\sim\rho_\pi^\star,r,s'\sim P_{R,S}^\star(s,a)}\left[\frac{\nabla_\theta p(r,s'|s,a,\theta_i^\star)}{p(r,s'|s,a,\theta_i^\star)}\frac{\nabla_\theta p(r,s'|s,a,\theta_i^\star)^\top}{p(r,s'|s,a,\theta_i^\star)}\right],$$

$$= -\mathbb{E}_{s,a\sim\rho_\pi^\star,r,s'\sim P_{R,S}^\star(s,a)}\left[\nabla_\theta \log p(r,s'|s,a,\theta_i^\star)\nabla_\theta \log p(r,s'|s,a,\theta_i^\star)^\top\right],$$

$$= -\Sigma_i^g.$$

Using this result, each $\mathrm{Tr}\left(\Sigma_i^g H_i^{\star-1}\right) = \mathrm{Tr}\left(-I\right) = -d$. Substituting y ields:

$$\mathbb{E}_{\mathcal{D}_N\sim P_{\mathrm{Data}}^\star}\left[\mathrm{Regret}(\mathcal{M}^\star,\mathcal{D}_N)\right] \leq 2\mathcal{R}_{\max}\cdot\sqrt{1-\exp\left(-C\frac{(k+1)d}{(1-\gamma)N}\right)},$$

$$\leq 2\mathcal{R}_{\max}\cdot\sqrt{1-\exp\left(-C'\frac{d}{(1-\gamma)N}\right)},$$

for some $0 < C' < \infty$ and sufficiently large $N$, as required. $\qquad\square$

We note that Theorem 2 applies to the Gaussian world model introduced in Section 4.4 with neural network mean functions with $C^2$-continuous activations (tanh, identity, sigmoid, softplus, SiLU, SELU, GELU...) using a Gaussian or uniform prior truncated to a compact parameter space and similarly well-behaved parametric models. The resulting differences in performance only arise from the choice of prior, model representability and coverage of dataset, which affect Bayesian and frequentist methods equally. The information rate coincides with the optimal 'minimax' convergence rate of frequentist parametric density estimators (Yang and Barron, 1999; Bilodeau et al., 2021). Similar results for the information rate have been found for nonparametric models such a Gaussian processes (van der Vaart and van Zanten, 2011).

Using our result in Theorem 2, we plot the normalised regret bound (i.e. taking $\mathcal{R}_{\max} = 0.5$) in Ineq. 18 for increasing dimensionality (blue) and decreasing $\gamma$ (copper) in Fig. 5. Our bound reveals an S-shaped curve with three distinct phases as number of data points $N$ increases: an initial plateau, a sudden decrease in regret follow by a slow exponential decay towards a regret of zero. The plateau indicates that a minimum amount of data is needed before any benefit can be realised in terms of regret. This is to be expected because initially the only information about the parameter values is given by the prior, which has no guarantee of accuracy under our analysis. Once a threshold of data points has been reached, the data can start to overwhelm

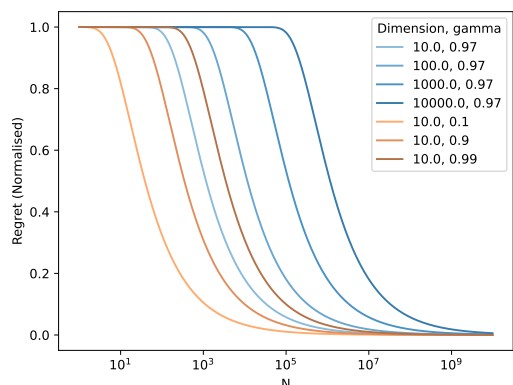

Figure 5: Normalised Regret Curves for $C = 1$

the prior, resulting in a sudden decrease in regret. The higher the dimensionality of the model, the greater this data limit is - represented in Fig. 5 by the plateau length increasing with greater $d$ (blue curves). Due to overspecification in models, this limit is likely to be set by the effective dimension of

the problem (which may be much lower than $d$) as many parameters will be redundant, however the effective dimension is typically not possible to ascertain a priori. Finally, we observe that increasing the discount factor $\gamma$ leads to a longer regret plateau (copper curves) due to any error in the model dynamics being compounded over a longer horizon at test time.

### D.4 Extensions for Model Misspecification and Sub-optimal Policy Learning

We now generalise our theorems to include the effects of model misspecification, that is models that cannot fully represent the true environment dynamics, and sub-optimal Bayesian policy learning, that is the effect of using a policy that does not fully optimise the Bayesian RL objective. We use the dagger notation to denote the maximum cross entropy parametrisation:

$$\theta_i^\dagger \in \arg\max_{\theta \in \Theta} \ell(\theta) = \arg\min_{\theta \in \Theta} \mathbb{E}_{s,a\sim\rho_\pi^\star,r,s'\sim P_{R,S}^\star(s,a)} \left[ \mathrm{KL}\left( P_{R,S}^\star(s,a) | P_{R,S}(s,a,\theta) \right) \right].$$

To characterise the degree of model misspecification, we use the KL divergence:

$$\epsilon_{\mathrm{miss}} := \min_{\theta\in\Theta} \mathbb{E}_{s,a\sim\rho_\pi^\star,r,s'\sim P_{R,S}^\star(s,a)} \left[ \mathrm{KL}\left( P_{R,S}^\star(s,a) | P_{R,S}(s,a,\theta) \right) \right].$$

We also introduce the following simplifying dagger notation for the expected cross entropy and corresponding gradients and Hessian under the optimal parameter:

$$\ell^\dagger := \max_{\theta\in\Theta} \ell(\theta) = \mathbb{E}_{s,a\sim\rho_\pi^\star,r,s'\sim P_{R,S}^\star(s,a)} \left[ \log p(r, s'|s, a, \theta_i^\dagger) \right],$$

$$g_{i,N}^\dagger := \sqrt{N} \nabla_\theta \ell_N(\theta)\big|_{\theta=\theta_i^\dagger},$$

$$H_i^\dagger := \nabla_\theta^2 \ell(\theta)\big|_{\theta=\theta_i^\dagger}.$$

We now relax Assumption 2 to allow for model misspecification:

**Assumption 2.** *We assume that:*

  *i  The maximum likelihood is finite $\left| \ell^\dagger \right| < \infty$ and $\left| \ell^\dagger - \ell(\theta) \right|$ is bounded $P_\Theta$-almost surely.*

 *ii  $\ell_N(\theta)$ and $\ell(\theta)$ are $C^2$-continuous in $\theta$.*

*iii  There are $K < \infty$ maximising points $\theta_i^\dagger$:*

$$\{\theta_1^\dagger, \theta_2^\dagger, \ldots \theta_K^\dagger\} = \arg\max_{\theta\in\Theta} \ell(\theta).$$

  *For each maximiser $\theta_i^\dagger$, there exists a small region $\Theta_i^\dagger := \{\theta \in \Theta | \|\theta_i^\dagger - \theta\| \leq \epsilon\}$ for some $\epsilon > 0$ such that $\theta_i^\dagger$ is the unique maximiser in $\Theta_i^\dagger$, $\theta_i^\dagger$ is in the interior of $\Theta_i^\dagger$, $\nabla_\theta^2 \ell(\theta_i^\dagger)$ is negative definite, invertible and the regions are disjoint: $\bigcap_{i=1}^K \Theta_i^\dagger = \varnothing$.*

 *iv  The prior $p(\theta)$ is Lipschitz continuous in $\theta$ with support over $\Theta$.*

  *v  The sampling regime ensures that the strong law of large numbers holds for all maximisers $\theta_i^\dagger$ for the Hessian, and uniformly for $\theta \in \Theta$ for the likelihood, that is:*

$$\ell_N(\theta) \xrightarrow{Unif.\ a.s.} \ell(\theta), \quad \nabla_\theta^2 \ell_N(\theta_i^\dagger) \xrightarrow{a.s.} \nabla_\theta^2 \ell(\theta_i^\dagger).$$

  *The central limit theorem applies to the gradient at each $\theta_i^\dagger$, that is:*

$$\sqrt{N} \nabla_\theta \ell_N(\theta_i^\dagger) \xrightarrow{d} \mathcal{N}(0, \Sigma_i^g),$$

  *where $\Sigma_i^g = \mathbb{E}_{s,a\sim\rho_\pi^\star,r,s'\sim P_{R,S}^\star(s,a)} \left[ \nabla_\theta \log p(r, s'|s, a, \theta_i^\dagger) \nabla_\theta \log p(r, s'|s, a, \theta_i^\dagger)^\top \right]$ with $\|\Sigma_i^g\| < \infty$.*

Finally, we account for let the Bayes sub-optimality be defined as

$$\epsilon_{\mathrm{Bayes}} := \left| J_{\mathrm{Bayes}}^{\pi_{\mathrm{Bayes}}^\star}(P_\Phi(\mathcal{D}_N)) - J_{\mathrm{Bayes}}^{\hat{\pi}}(P_\Phi(\mathcal{D}_N)) \right|. \tag{26}$$

**Lemma 5.** *Let* $\mathcal{R}_{\max} := \frac{(r_{max} - r_{min})}{1-\gamma}$ *denote the maximum possible regret for the MDP and the Bayes sub-optimality* $\epsilon_{Bayes}$ *be defined as in Eq.* (26). *For a prior* $P_\Theta(\mathcal{D}_N)$, *the true regret can be bounded as:*

$$\text{Regret}(\mathcal{M}^\star, \mathcal{D}_N) \leq 2\mathcal{R}_{\max} \cdot \sup_\pi \mathbb{E}_{i \sim \mathcal{G}(\gamma)} \left[ TV\left( P^\star_{i+1, \pi} \| P^\pi_{i+1}(\mathcal{D}_N) \right) \right] + \epsilon_{\text{Bayes}}.$$

*Proof.* We start from the definition of the true regret under Bayes sub-optimality:

$$
\begin{aligned}
\text{Regret}(\mathcal{M}^\star, \mathcal{D}_N) &:= J^{\pi^\star}(\mathcal{M}^\star) - J^{\hat{\pi}}(\mathcal{M}^\star, \mathcal{D}_N), \\
&= J^{\pi^\star}(\mathcal{M}^\star) - J^{\pi^\star}_{\text{Bayes}}(P_\Phi(\mathcal{D}_N)) + J^{\pi^\star}_{\text{Bayes}}(P_\Phi(\mathcal{D}_N)) - J^{\hat{\pi}}(\mathcal{M}^\star, \mathcal{D}_N), \\
&\leq J^{\pi^\star}(\mathcal{M}^\star) - J^{\pi^\star}_{\text{Bayes}}(P_\Phi(\mathcal{D}_N)) + J^{\pi^\star_{\text{Bayes}}}_{\text{Bayes}}(P_\Phi(\mathcal{D}_N)) - J^{\hat{\pi}}(\mathcal{M}^\star, \mathcal{D}_N), \\
&= J^{\pi^\star}(\mathcal{M}^\star) - J^{\pi^\star}_{\text{Bayes}}(P_\Phi(\mathcal{D}_N)) + J^{\hat{\pi}}_{\text{Bayes}}(P_\Phi(\mathcal{D}_N)) - J^{\hat{\pi}}(\mathcal{M}^\star, \mathcal{D}_N) \\
&\quad + J^{\pi^\star_{\text{Bayes}}}_{\text{Bayes}}(P_\Phi(\mathcal{D}_N)) - J^{\hat{\pi}}_{\text{Bayes}}(P_\Phi(\mathcal{D}_N)), \\
&\leq \sup_\pi \left| J^\pi(\mathcal{M}^\star) - J^\pi_{\text{Bayes}}(P_\Theta(\mathcal{D}_N)) \right| + \sup_\pi \left| J^\pi_{\text{Bayes}}(P_\Theta(\mathcal{D}_N)) - J^\pi(\mathcal{M}^\star) \right| \\
&\quad + \left| J^{\pi^\star_{\text{Bayes}}}_{\text{Bayes}}(P_\Phi(\mathcal{D}_N) - J^{\hat{\pi}}_{\text{Bayes}}(P_\Phi(\mathcal{D}_N) \right|, \\
&= 2 \sup_\pi \left| J^\pi(\mathcal{M}^\star) - J^\pi_{\text{Bayes}}(P_\Theta(\mathcal{D}_N)) \right| + \epsilon_{\text{Bayes}},
\end{aligned}
$$

where the second line follows from $J^{\pi^\star}_{\text{Bayes}}(P_\Phi(\mathcal{D}_N)) \leq J^{\pi^\star_{\text{Bayes}}}_{\text{Bayes}}(P_\Phi(\mathcal{D}_N))$ by definition. We then bound $\left| J^\pi(\mathcal{M}^\star) - J^\pi_{\text{Bayes}}(P_\Theta(\mathcal{D}_N)) \right|$ using Lemma 1 to obtain our desired result. $\square$

**Theorem 3.** *Let the data be drawn from the underlying true distribution* $\mathcal{D}_N \sim P^\star_{Data}$. *Under Assumption 2, there exists some constant* $0 < C < \infty$ *such that for sufficiently large* $N$:

$$\mathbb{E}_{\mathcal{D}_N \sim P^\star_{Data}} \left[ \text{Regret}(\mathcal{M}^\star, \mathcal{D}_N) \right] \leq 2\mathcal{R}_{\max} \cdot \exp\left( 1 - \sqrt{C \left( \frac{d}{(1-\gamma)N} + \frac{\epsilon_{\text{miss}}}{1-\gamma} \right)} \right) + \epsilon_{\text{Bayes}}.$$

*Proof.* Starting with Lemma 5, we obtain:

$$\text{Regret}(\mathcal{M}^\star, \mathcal{D}_N) \leq 2\mathcal{R}_{\max} \cdot \sqrt{1 - \exp\left( -C \left( \frac{d^2}{(1-\gamma)N} + \frac{\epsilon_{\text{miss}}}{(1-\gamma)} \right) \right)} + \epsilon_{\text{Bayes}}.$$

Next we apply Theorem 1 to bound the first term, obtaining:

$$\text{Regret}(\mathcal{M}^\star, \mathcal{D}_N) \leq 2\mathcal{R}_{\max} \cdot \sup_\pi \sqrt{1 - \exp\left( -\frac{\mathcal{I}^\pi_N}{1-\gamma} \right)} + \epsilon_{\text{Bayes}}.$$

We write the PIL to include misspecification as:

$$\mathcal{I}_N^\pi := \mathbb{E}_{s,a\sim\rho_\pi^\star}\left[\mathbb{E}_{\theta\sim P_\Theta(\mathcal{D}_N)}\left[\mathrm{KL}\left(P_{R,S}^\star(s,a)\|P_{R,S}(s,a,\theta)\right)\right]\right],$$

$$=\mathbb{E}_{\theta\sim P_\Theta(\mathcal{D}_N)}\left[\mathbb{E}_{s,a\sim\rho_\pi^\star,r,s'\sim P_{R,S}^\star(s,a)}\left[\log p(r,s'|s,a,\theta^\star)-\log p(r,s'|s,a,\theta)\right]\right],$$

$$=\mathbb{E}_{\theta\sim P_\Theta(\mathcal{D}_N)}\Big[\mathbb{E}_{s,a\sim\rho_\pi^\star,r,s'\sim P_{R,S}^\star(s,a)}[\log p(r,s'|s,a,\theta^\star)-\log p(r,s'|s,a,\theta_i^\dagger)$$
$$+\log p(r,s'|s,a,\theta_i^\dagger)-\log p(r,s'|s,a,\theta)]\Big],$$

$$=\mathbb{E}_{\theta\sim P_\Theta(\mathcal{D}_N)}\Big[\mathbb{E}_{s,a\sim\rho_\pi^\star,r,s'\sim P_{R,S}^\star(s,a)}\left[\log p(r,s'|s,a,\theta^\star)-\log p(r,s'|s,a,\theta_i^\dagger)\right]$$
$$+\mathbb{E}_{s,a\sim\rho_\pi^\star,r,s'\sim P_{R,S}^\star(s,a)}\left[\log p(r,s'|s,a,\theta_i^\dagger)-\log p(r,s'|s,a,\theta)\right]\Big]\Big],$$

$$=\mathbb{E}_{s,a\sim\rho_\pi^\star,r,s'\sim P_{R,S}^\star(s,a)}\left[\log p(r,s'|s,a,\theta^\star)-\log p(r,s'|s,a,\theta_i^\dagger)\right]$$
$$+\mathbb{E}_{\theta\sim P_\Theta(\mathcal{D}_N)}\left[\mathbb{E}_{s,a\sim\rho_\pi^\star,r,s'\sim P_{R,S}^\star(s,a)}\left[\log p(r,s'|s,a,\theta_i^\dagger)-\log p(r,s'|s,a,\theta)\right]\right],$$

$$=\mathbb{E}_{s,a\sim\rho_\pi^\star,r,s'\sim P_{R,S}^\star(s,a)}\left[\mathrm{KL}\left(P_{R,S}^\star(s,a)|P_{R,S}(s,a,\theta_i^\dagger)\right)\right]$$
$$+\mathbb{E}_{\theta\sim P_\Theta(\mathcal{D}_N)}\left[\mathbb{E}_{s,a\sim\rho_\pi^\star,r,s'\sim P_{R,S}^\star(s,a)}\left[\log p(r,s'|s,a,\theta_i^\dagger)-\log p(r,s'|s,a,\theta)\right]\right],$$

$$=\mathbb{E}_{\theta\sim P_\Theta(\mathcal{D}_N)}\left[\ell^\dagger-\ell(\theta)\right]+\mathbb{E}_{s,a\sim\rho_\pi^\star,r,s'\sim P_{R,S}^\star(s,a)}\left[\mathrm{KL}\left(P_{R,S}^\star(s,a)|P_{R,S}(s,a,\theta_i^\dagger)\right)\right],$$

$$=\mathbb{E}_{\theta\sim P_\Theta(\mathcal{D}_N)}\left[\ell^\dagger-\ell(\theta)\right]+\min_\theta\mathbb{E}_{s,a\sim\rho_\pi^\star,r,s'\sim P_{R,S}^\star(s,a)}\left[\mathrm{KL}\left(P_{R,S}^\star(s,a)|P_{R,S}(s,a,\theta)\right)\right],$$

$$=\mathbb{E}_{\theta\sim P_\Theta(\mathcal{D}_N)}\left[\ell^\dagger-\ell(\theta)\right]+\epsilon_{\mathrm{miss}}.$$

To bound the first term $\mathbb{E}_{\theta\sim P_\Theta(\mathcal{D}_N)}\left[\ell^\dagger-\ell(\theta)\right]$, we follow the remainder of the proof of Theorem 2 to Ineq. 24, replacing $\star$ notion with $\dagger$ to yield:

$$\mathcal{I}_N^\pi=\epsilon_{\mathrm{miss}}+\mathcal{O}\left(\frac{d-\sum_{i=1}^K g_{i,N}^{\dagger\,\top}H_i^{\dagger\,-1}g_{i,N}^\dagger}{N}\right). \tag{27}$$

almost surely. As $f(x):=2\mathcal{R}_{\max}\cdot\sqrt{1-\exp\left(-\frac{x}{(1-\gamma)}\right)}$ is monotonic in $x$ and $\frac{d-\sum_{i=1}^K g_{i,N}^{\dagger\,\top}H_i^{\dagger\,-1}g_{i,N}^\dagger}{N}+\epsilon_{\mathrm{miss}}\geq 0$, Eq. (27) implies there exists some positive $0<C<\infty$ such that:

$$\mathrm{Regret}(\mathcal{M}^\dagger,\mathcal{D}_N)\leq 2\mathcal{R}_{\max}\cdot\sqrt{1-\exp\left(-C\left(\frac{d-\sum_{i=1}^K g_{i,N}^{\dagger\,\top}H_i^{\dagger\,-1}g_{i,N}^\dagger}{(1-\gamma)N}+\frac{\epsilon_{\mathrm{miss}}}{(1-\gamma)}\right)\right)},$$

almost surely for large enough $N$. Under Assumption 2, $g_{i,N}^\dagger\xrightarrow{d}\mathcal{N}(0,\Sigma_i^g)$. As $f(x)$ is also a bounded, continuous function and concave, we can apply the Portmanteau Theorem (see for example Bass (2013, Chapter 21.7)) followed by Jensen's inequality to yield:

$$\mathbb{E}_{\mathcal{D}_N\sim P_{\mathrm{Data}}^\dagger}\left[\mathrm{Regret}(\mathcal{M}^\dagger,\mathcal{D}_N)\right]$$

$$\leq 2\mathcal{R}_{\max}\cdot\mathbb{E}_{g_i\sim\mathcal{N}(0,\Sigma_i^g)}\left[\sqrt{1-\exp\left(-C\left(\frac{d-\sum_{i=1}^K g_i^\top H_i^{\dagger\,-1}g_i}{(1-\gamma)N}++\frac{\epsilon_{\mathrm{miss}}}{(1-\gamma)}\right)\right)}\right],$$

$$\leq 2\mathcal{R}_{\max}\cdot\sqrt{1-\exp\left(-C\left(\frac{d-\sum_{i=1}^K\mathbb{E}_{g_i\sim\mathcal{N}(0,\Sigma_i^g)}\left[g^\top H_i^{\dagger\,-1}g\right]}{(1-\gamma)N}+\frac{\epsilon_{\mathrm{miss}}}{(1-\gamma)}\right)\right)},$$

$$=2\mathcal{R}_{\max}\cdot\sqrt{1-\exp\left(-C\left(\frac{d-\sum_{i=1}^K\mathrm{Tr}\left(\Sigma_i^g H_i^{\dagger\,-1}\right)}{(1-\gamma)N}+\frac{\epsilon_{\mathrm{miss}}}{(1-\gamma)}\right)\right)}.$$

Now $\text{Tr}\left(\Sigma_i^g H_i^{\star-1}\right) = \mathcal{O}(d^2)$, hence

$$\mathbb{E}_{\mathcal{D}_N \sim P_{\text{Data}}^\dagger}\left[\text{Regret}(\mathcal{M}^\dagger, \mathcal{D}_N)\right] \leq 2\mathcal{R}_{\max} \cdot \sqrt{1 - \exp\left(-C\left(\frac{(k+1)d^2}{(1-\gamma)N} + \frac{\epsilon_{\text{miss}}}{(1-\gamma)}\right)\right)},$$

$$\leq 2\mathcal{R}_{\max} \cdot \sqrt{1 - \exp\left(-C'\left(\frac{d^2}{(1-\gamma)N} + \frac{\epsilon_{\text{miss}}}{(1-\gamma)}\right)\right)},$$

for some $0 < C' < \infty$ and sufficiently large $N$, as required. $\qquad\square$

Taking the limit $N \to \infty$, we see the residual term due to misspecification and sub-optimal policy learning is:

$$\mathbb{E}_{\mathcal{D}_N \sim P_{\text{Data}}^\dagger}\left[\text{Regret}(\mathcal{M}^\dagger, \mathcal{D}_N)\right] \leq 2\mathcal{R}_{\max} \cdot \sqrt{1 - \exp\left(-\frac{\epsilon_{\text{miss}}C}{1-\gamma}\right)} + \epsilon_{\text{Bayes}}.$$

We compare against prior work such as MOReL (Kidambi et al., 2020, Corollary 2), where the residual misspecification term is:

$$\frac{4\gamma\mathcal{R}_{\max}}{1-\gamma}\epsilon_{TV}$$

where $\epsilon_{TV}$ characterises the misspecification in terms of total variational distance instead of KL divergence of our method. Crucially, we see that our bound is much less sensitive to $\gamma$; our bound is $\mathcal{O}(\frac{1}{\sqrt{1-\gamma}})$ in comparison to $\mathcal{O}(\frac{\gamma}{1-\gamma})$ of MOReL meaing our bound is tighter as $\gamma \to 1$.

# E  FURTHER RESULTS

## E.1  TOReL

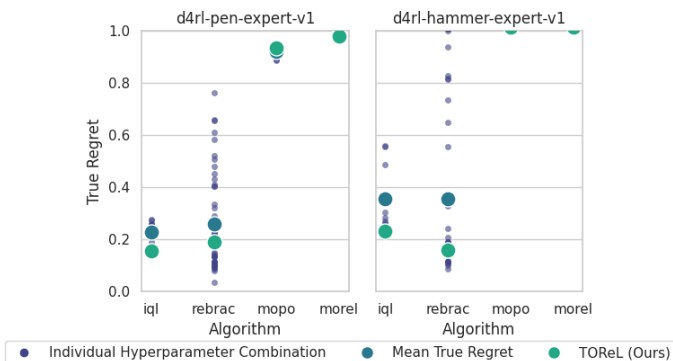

Figure 6: TOReL-hyperparameter regret versus mean hyperparameter regret for Adroit (lower is better).

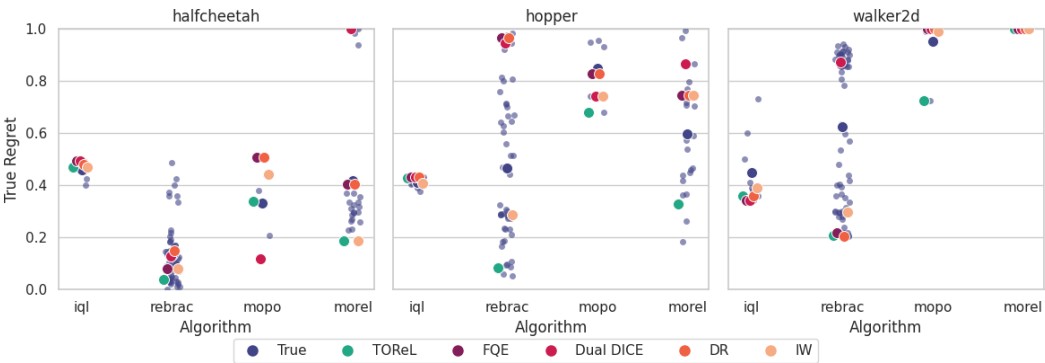

Figure 7: Hyperparameters selected by different OPE metrics for d4rl-halfcheetah-medium-expert-v2, d4rl-hopper-medium-v2 and d4rl-walker-medium-replay-v2 (lower is better).

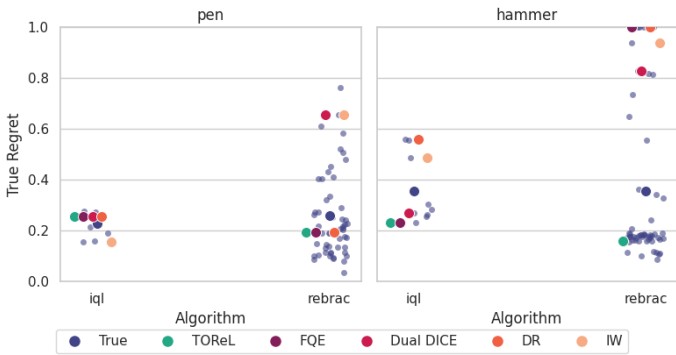

Figure 8: Hyperparameters selected by different OPE metrics for d4rl-pen-expert-v1 and d4rl-hammer-expert-v1 (lower is better).

| Task | | IQL | ReBRAC | MOPO | MOReL |
|------|---|-----|--------|------|-------|
| **brax-** | True | 0.203 | 0.312 | 0.403 | 0.751 |
| **halfcheetah-** | Oracle | 0.186 | 0.089 | 0.133 | 0.641 |
| **full-replay** | TOReL | *0.186* | *0.089* | *0.133* | *0.648* |
| **brax-** | True | 0.550 | 0.534 | 0.558 | 0.575 |
| **hopper-** | Oracle | 0.377 | 0.070 | 0.082 | 0.243 |
| **full-replay** | TOReL | *0.397* | *0.070* | *0.086* | *0.282* |
| **brax-** | True | 0.374 | 0.357 | 0.342 | 0.625 |
| **walker-** | Oracle | 0.304 | 0.000 | 0.243 | 0.415 |
| **full-replay** | TOReL | *0.331* | *0.000* | 0.384 | *0.554* |
| | True | 0.469 | 0.134 | 0.331 | 0.418 |
| | Oracle | 0.400 | 0.000 | 0.116 | 0.187 |
| **d4rl-** | TOReL | 0.469 | *0.036* | 0.339 | *0.187* |
| **halfcheetah-** | FQE | 0.492 | 0.079 | 0.507 | 0.402 |
| **medium-expert-v2** | DICE | 0.492 | 0.127 | 0.116 | 1.000 |
| | DR | 0.480 | 0.147 | 0.507 | 0.402 |
| | IW | 0.469 | 0.079 | 0.441 | 0.187 |
| | True | 0.411 | 0.467 | 0.848 | 0.595 |
| | Oracle | 0.375 | 0.053 | 0.681 | 0.183 |
| **d4rl-** | TOReL | 0.428 | *0.083* | *0.681* | *0.327* |
| **hopper-** | FQE | 0.432 | 0.967 | 0.829 | 0.744 |
| **medium-v2** | DICE | 0.432 | 0.945 | 0.743 | 0.865 |
| | DR | 0.432 | 0.967 | 0.829 | 0.744 |
| | IW | 0.406 | 0.285 | 0.743 | 0.744 |
| | True | 0.450 | 0.625 | 0.952 | 1.000 |
| | Oracle | 0.339 | 0.204 | 0.724 | 1.000 |
| **d4rl-** | TOReL | *0.358* | *0.204* | *0.724* | **1.000** |
| **walker2d-** | FQE | 0.339 | 0.217 | 1.000 | 1.000 |
| **medium-replay-v2** | DICE | 0.341 | 0.872 | 1.000 | 1.000 |
| | DR | 0.358 | 0.204 | 1.000 | 1.000 |
| | IW | 0.388 | 0.295 | 0.990 | 1.000 |

Table 2: True, oracle, TOReL and OPE regrets across tasks. Bold indicates where TOReL identifies the oracle hyperparameters, while italic indicates where TOReL identifies hyperparameters with a regret lower than the true regret. ReBRAC+TOReL outperforms all algorithms on every dataset. Green indicates where TOReL is the best OPE metric. Orange indicates where another OPE metric beats or ties with TOReL. OPE metrics for the brax datasets are undetermined, as the brax datasets do not contain full trajectories (required for DR and IW) or initial state flags (required for DICE). Given TOReL's strong performance on the brax datasets (with ReBRAC + TOReL achieving the Oracle hyper-parameters for each dataset), FQE could only have performed on par with it.

| Task | | IQL | ReBRAC | MOPO | MOReL |
|------|------|------|--------|------|-------|
| **d4rl-pen-expert-v1** | True | 0.226 | 0.258 | 0.919 | 0.985 |
| | Oracle | 0.154 | 0.033 | 0.885 | 0.968 |
| | TOReL | 0.254 | *0.192* | | |
| | FQE | 0.254 | 0.192 | | |
| | DICE | 0.254 | 0.654 | | |
| | DR | 0.254 | 0.192 | | |
| | IW | 0.154 | 0.654 | | |
| **d4rl-hammer-expert-v1** | True | 0.355 | 0.354 | 1.000 | 1.000 |
| | Oracle | 0.229 | 0.086 | 1.000 | 1.000 |
| | TOReL | ***0.229*** | *0.159* | | |
| | FQE | 0.229 | 1.000 | | |
| | DICE | 0.268 | 0.827 | | |
| | DR | 0.558 | 1.000 | | |
| | IW | 0.485 | 0.938 | | |

Table 3: True, oracle, TOReL and OPE regrets across tasks. Bold indicates where TOReL identifies the oracle hyperparameters, while italic indicates where TOReL identifies hyperparameters with a regret lower than the true regret. Green indicates where TOReL is the best OPE metric. Orange indicates where another OPE metric beats or ties with TOReL. All metrics are unreliable for the pen and hammer datasets due to the quality of the data. We refer the reader to the corresponding scatter-plots.

| Task | | | IQL | ReBRAC | MOPO | MOReL |
|---|---|---|---|---|---|---|
| **brax-halfcheetah-full-replay** | TOReL | r | 0.29 | 0.92 | 0.98 | 0.93 |
| | | p | 0.448 | 0.000 | 0.000 | 0.000 |
| **brax-hopper-full-replay** | TOReL | r | 0.18 | 0.98 | 1.00 | 0.98 |
| | | p | 0.635 | 0.000 | 0.00 | 0.000 |
| **brax-walker-full-replay** | TOReL | r | 0.32 | nan | $-0.68$ | 0.99 |
| | | p | 0.406 | nan | 0.133 | 0.000 |
| **d4rl-halfcheetah-medium-expert-v2** | TOReL | r | 0.29 | 0.90 | 0.02 | 0.98 |
| | | p | 0.442 | 0.000 | 0.975 | 0.000 |
| | FQE | r | $-0.21$ | 0.66 | $-0.32$ | 0.97 |
| | | p | 0.591 | 0.000 | 0.530 | 0.000 |
| | DICE | r | $-0.47$ | 0.19 | 0.74 | 0.65 |
| | | p | 0.204 | 0.166 | 0.094 | 0.001 |
| | DR | r | $-0.24$ | 0.32 | $-0.27$ | 0.96 |
| | | p | 0.537 | 0.017 | 0.602 | 0.000 |
| | IW | r | $-0.33$ | $-0.44$ | 0.07 | 0.97 |
| | | p | 0.393 | 0.001 | 0.901 | 0.000 |
| **d4rl-hopper-medium-v2** | TOReL | r | 0.29 | 0.98 | 0.98 | 0.94 |
| | | p | 0.443 | 0.000 | 0.001 | 0.000 |
| | FQE | r | 0.16 | 0.27 | 0.87 | 0.37 |
| | | p | 0.674 | 0.044 | 0.025 | 0.0078 |
| | DICE | r | 0.19 | 0.34 | 0.31 | 0.09 |
| | | p | 0.623 | 0.009 | 0.554 | 0.673 |
| | DR | r | 0.26 | 0.20 | 0.78 | 0.34 |
| | | p | 0.499 | 0.135 | 0.065 | 0.099 |
| | IW | r | $-0.39$ | 0.27 | 0.24 | 0.24 |
| | | p | 0.301 | 0.042 | 0.647 | 0.264 |
| **d4rl-walker2d-medium-replay-v2** | TOReL | r | 0.65 | 0.78 | 1.00 | $-0.53$ |
| | | p | 0.057 | 0.000 | 0.000 | 0.008 |
| | FQE | r | 0.41 | 0.89 | $-0.48$ | $-0.05$ |
| | | p | 0.276 | 0.000 | 0.334 | 0.810 |
| | DICE | r | 0.52 | $-0.04$ | $-0.14$ | $-0.07$ |
| | | p | 0.148 | 0.700 | 0.785 | 0.744 |
| | DR | r | 0.19 | 0.82 | $-0.29$ | $-0.06$ |
| | | p | 0.631 | 0.000 | 0.571 | 0.780 |
| | IW | r | $-0.06$ | 0.86 | 0.40 | 0.96 |
| | | p | 0.875 | 0.000 | 0.427 | 0.000 |

Table 4: Pearson correlation ($r$) and statistical significance ($p$) between TOReL regret metrics and true regrets for different hyperparameter combinations. Even where no strong positive correlation is observed (possibly due to limited hyperparameter coverage), the TOReL regret is lower than the true regret averaged over those tasks. Green indicates where TOReL has strong ($r > |0.5|$), statically significant ($p < 0.05$) positive correlation, while orange indicates where other OPE metrics have strong, statistically significant correlation.

| Task | | | IQL | ReBRAC |
|---|---|---|---|---|
| **d4rl-pen-expert-v1** | TOReL | r | nan | nan |
| | | p | nan | nan |
| | FQE | r | nan | nan |
| | | p | nan | nan |
| | DICE | r | 0.49 | 0.00 |
| | | p | 0.178 | 0.980 |
| | DR | r | nan | 0.26 |
| | | p | nan | 0.056 |
| | IW | r | 0.26 | $-0.66$ |
| | | p | 0.497 | 0.000 |
| **d4rl-hammer-expert-v1** | TOReL | r | 0.70 | 0.30 |
| | | p | 0.035 | 0.023 |
| | FQE | r | 0.74 | 0.11 |
| | | p | 0.023 | 0.422 |
| | DICE | r | $-0.12$ | $-0.14$ |
| | | p | 0.755 | 0.313 |
| | DR | r | $-0.35$ | $-0.04$ |
| | | p | 0.349 | 0.745 |
| | IW | r | $-0.40$ | $-0.66$ |
| | | p | 0.287 | 0.000 |

Table 5: Pearson correlation ($r$) and statistical significance ($p$) between TOReL regret metrics and true regrets for different hyperparameter combinations. Green indicates where TOReL has strong ($r > |0.5|$), statically significant ($p < 0.05$) positive correlation, while orange indicates where other OPE metrics have strong, statistically significant correlation. All OPE metrics perform poorly due due to the nature of the datasets.

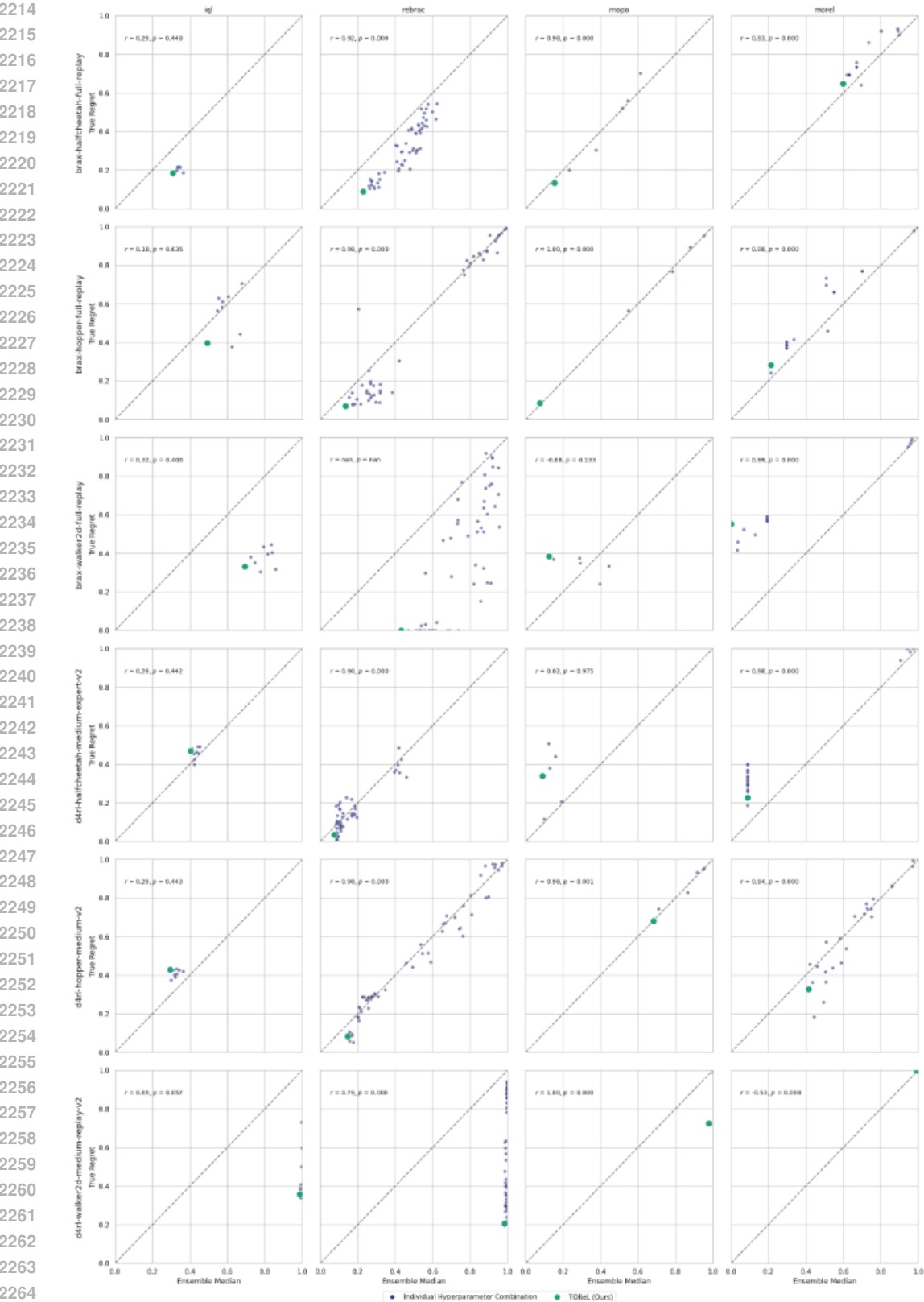

Figure 9: Scatter plots to visualise the positive correlation between the TOReL regret metric and the true regret.

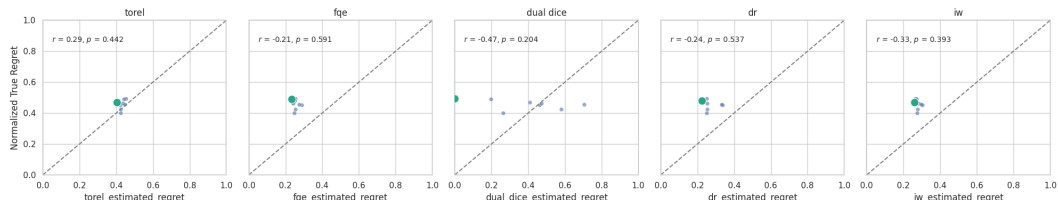

Figure 10: OPE metric scatter-plots: IQL-trained policies, d4rl-halfcheetah-medium-expert-v2.

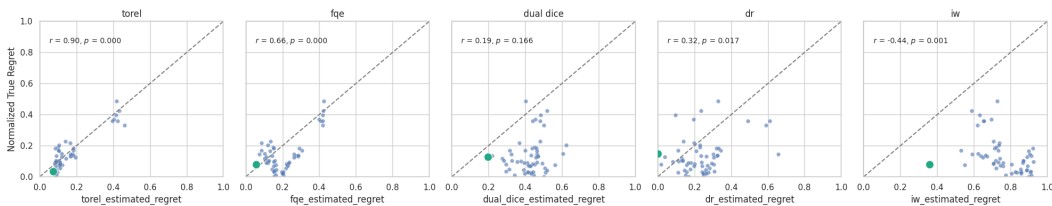

Figure 11: OPE metric scatter-plots: ReBRAC-trained policies, d4rl-halfcheetah-medium-expert-v2.

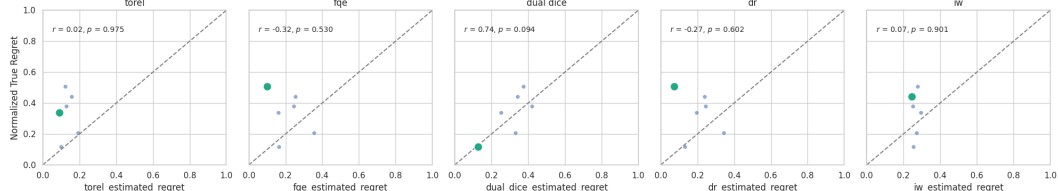

Figure 12: OPE metric scatter-plots: MOPO-trained policies, d4rl-halfcheetah-medium-expert-v2.

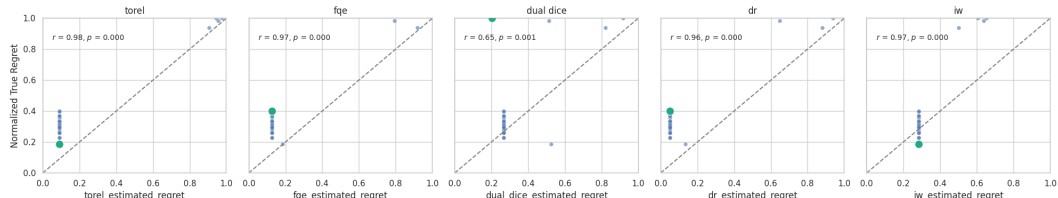

Figure 13: OPE metric scatter-plots: MOReL-trained policies, d4rl-halfcheetah-medium-expert-v2.

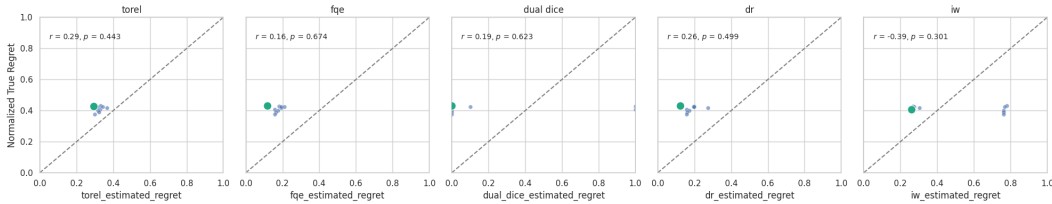

Figure 14: OPE metric scatter-plots: IQL-trained policies, d4rl-hopper-medium-v2.

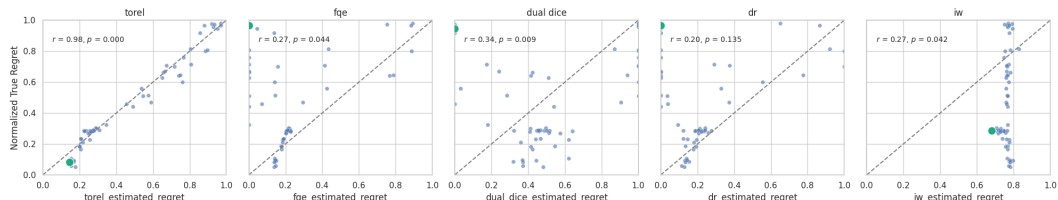

Figure 15: OPE metric scatter-plots: ReBRAC-trained policies, d4rl-hopper-medium-v2.

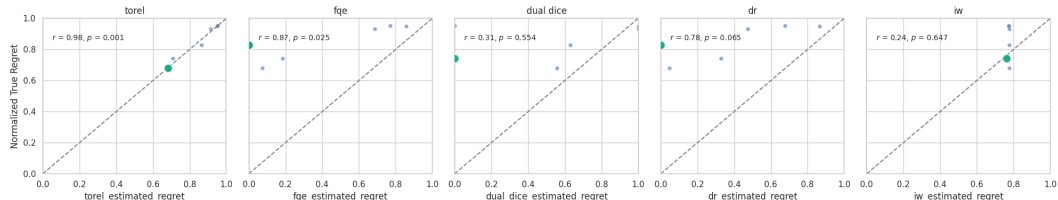

Figure 16: OPE metric scatter-plots: MOPO-trained policies, d4rl-hopper-medium-v2.

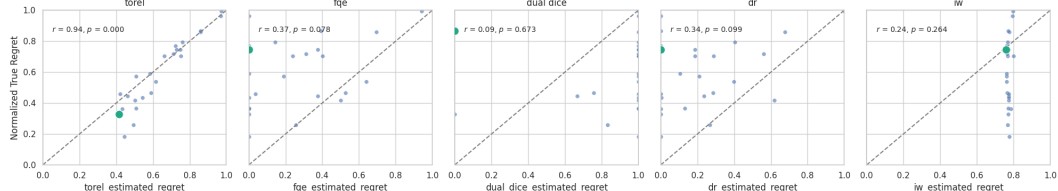

Figure 17: OPE metric scatter-plots: MOReL-trained policies, d4rl-hopper-medium-v2.

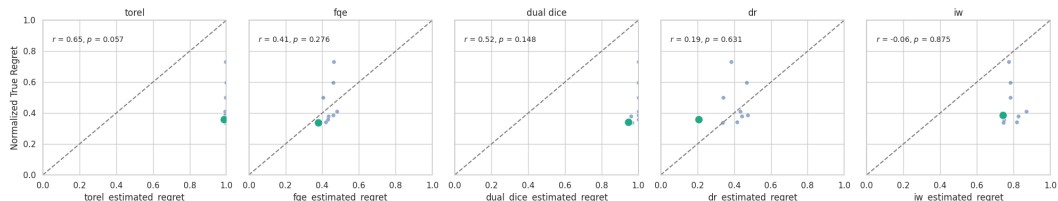

Figure 18: OPE metric scatter-plots: IQL-trained policies, d4rl-walker-medium-replay-v2.

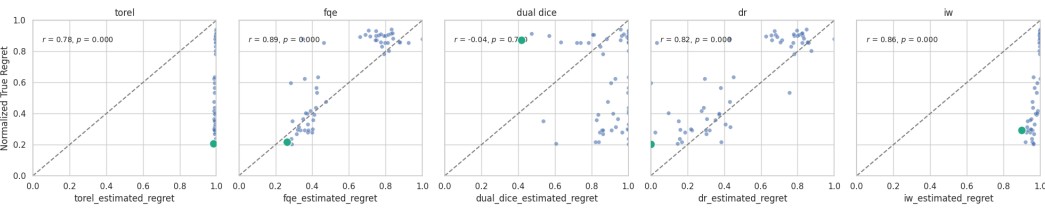

Figure 19: OPE metric scatter-plots: ReBRAC-trained policies, d4rl-walker-medium-replay-v2.

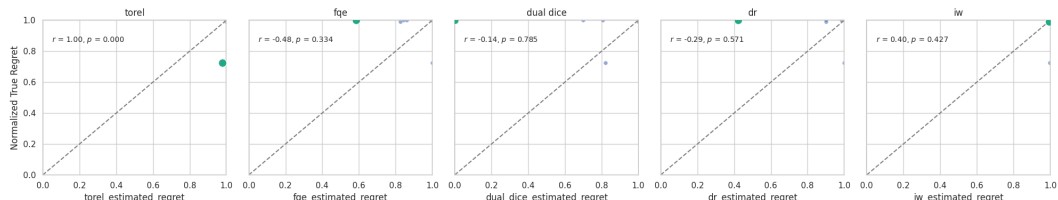

Figure 20: OPE metric scatter-plots: MOPO-trained policies, d4rl-walker-medium-replay-v2.

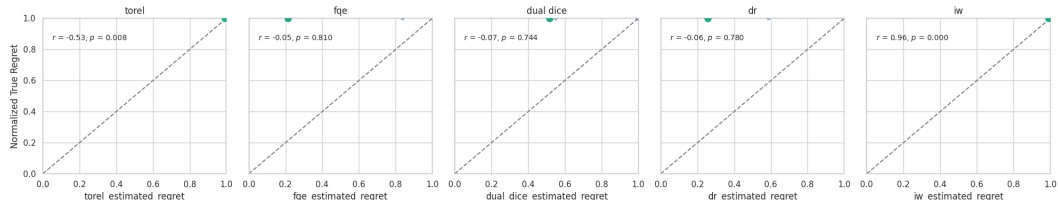

Figure 21: OPE metric scatter-plots: MOReL-trained policies, d4rl-walker-medium-replay-v2.

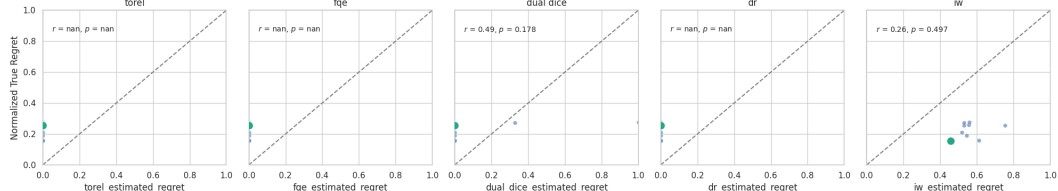

Figure 22: OPE metric scatter-plots: IQL-trained policies, d4rl-pen-expert-v1.

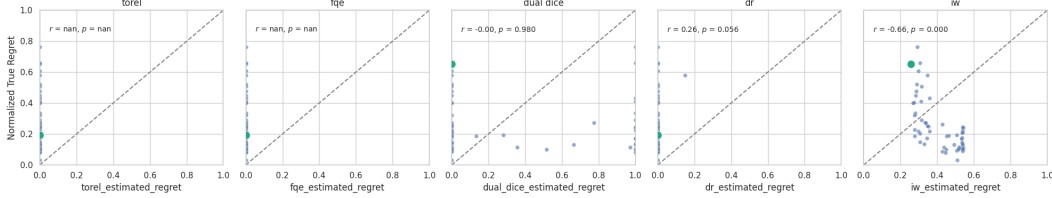

Figure 23: OPE metric scatter-plots: ReBRAC-trained policies, d4rl-pen-expert-v1.

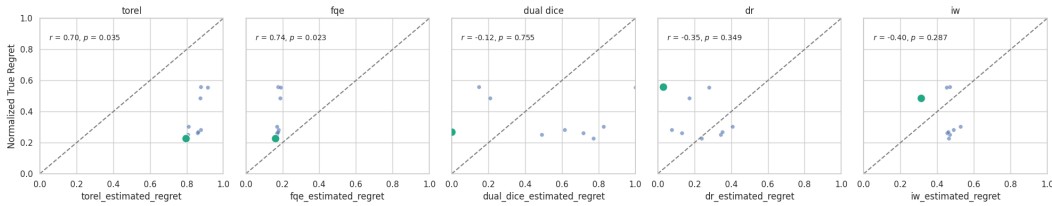

Figure 24: OPE metric scatter-plots: IQL-trained policies, d4rl-hammer-expert-v1.

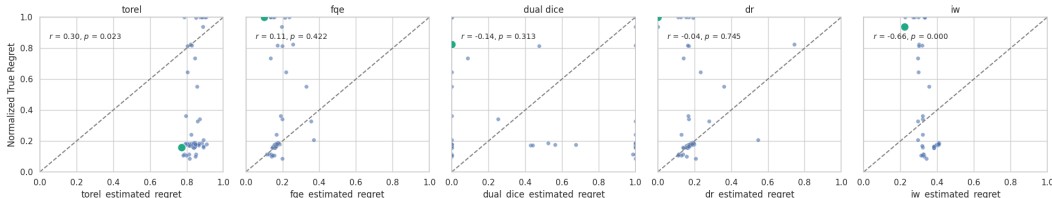

Figure 25: OPE metric scatter-plots: ReBRAC-trained policies, d4rl-hammer-expert-v1.

## E.2 SOREL

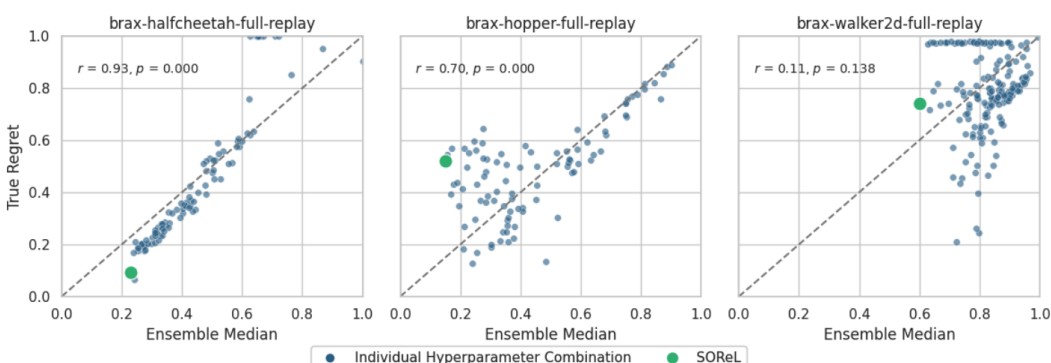

Figure 26: SOReL BAMDP hyperparameter sweeps (tuning set $\phi_{III}$) for 200,000 randomly sampled transitions of the brax datasets. The plots correspond to $\phi_{III} \leftarrow \arg\min_{\phi_{III}} \text{RegretMetric}(\phi_I, \phi_{II}, \phi_{III}, \mathcal{D}_N)$ in Algorithm 1. SOReL selects the BAMDP hyperparameters that yield the lowest approximate regret (green). For Walker2d the high approximate regret for all hyperparameter combinations ($R_N > 0.6$) suggests that in Algorithm 1 $R_N > \mathcal{R}_{\text{Deploy}}$ has not been satisfied: the practitioner should change the model or approximate inference method to obtain a lower PIL before re-tuning the BAMDP hyperparameters. Alternatively, the practitioner could consider collecting more data - though the high approximate regret highlights the associated risk. While we report the true regret to validate the approach, in practice only the policy with the lowest approximate regret would be deployed.

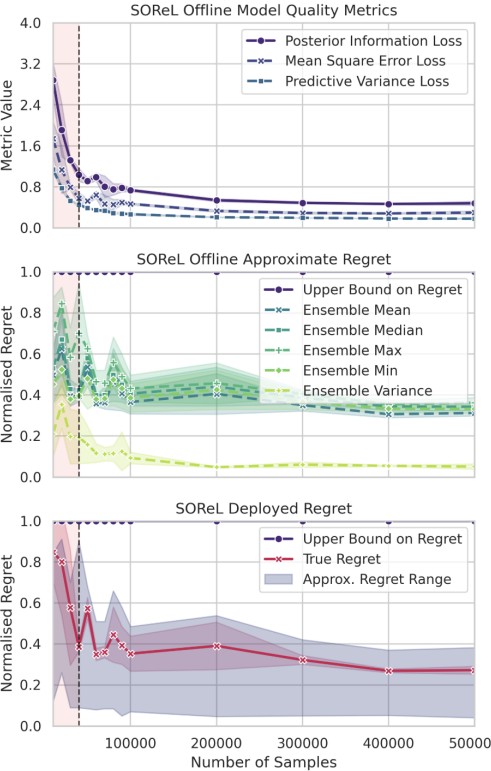

Figure 27: Simplified version of SOReL on brax-halfcheetah-full-replay. The plot showing only the ensemble median as the approximate regret is given in the main body of the paper. Shaded purple shows the approximate regret range across all the metrics (with varying degrees of conservatism). Shaded red indicates where $\mathcal{E}(\mathcal{D}_N, \mathcal{M}^\star) \not\approx \mathcal{V}(\mathcal{D}_N)$ (for a threshold of 0.25), and hence the approximate regret may be unreliable. Mean and standard deviation given over 3 seeds.

The only environment for which the world model is sufficiently accurate relative to its discount factor, resulting in a non-trivial upper bound, is pendulum-v1 (Fig. 28b). This is due to two factors: (i) the other environments use a lower discount factor (0.998 vs. 0.995), and (ii) learning accurate world models with low MSE is inherently more challenging in high-dimensional settings. We note that this is a supervised-learning problem, and orthogonal to our line of work.

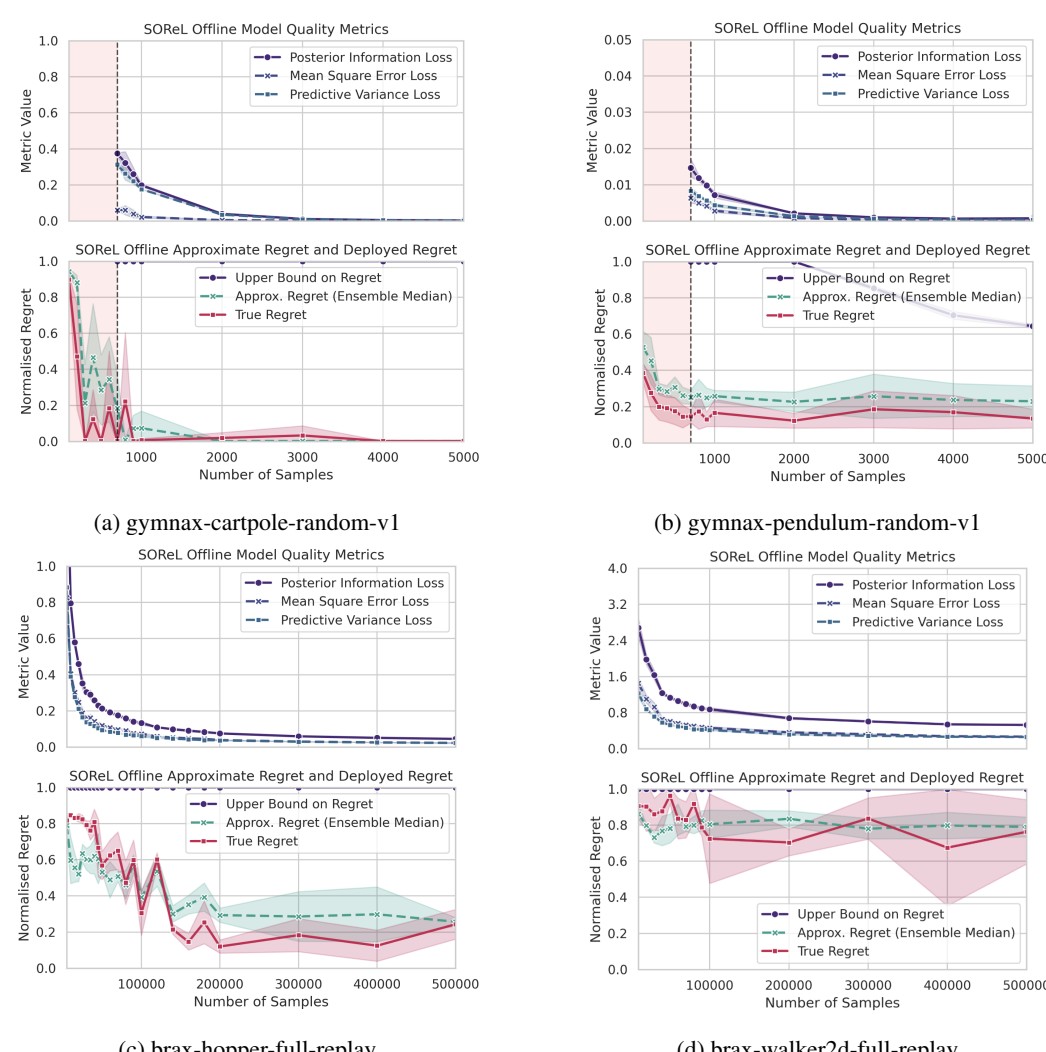

(a) gymnax-cartpole-random-v1

(b) gymnax-pendulum-random-v1

(c) brax-hopper-full-replay

(d) brax-walker2d-full-replay

Figure 28: Simplified version of SOReL applied to various tasks. For the gymnax environments, $N < 700$ is shaded red because the PIL is undefined in this region (validation set < batch size): without being able to ensure $\mathcal{E}(\mathcal{D}_N, \mathcal{M}^\star) <\approx \mathcal{V}(\mathcal{D}_N)$ the practitioner would have been unable to determine whether to trust the approximate regret. Mean and standard deviation given over 3 seeds.

# F IMPLEMENTATION DETAILS

Our implementations of SOReL and TOReL, along with all of the code to reproduce the experiments, are made publicly available with this work.

## F.1 DIVERSE FULL-REPLAY DATASETS

As mentioned in Section 6.2, without a model prior, the offline dataset must include transitions from poor, medium and expert regions of performance. For the brax environments, we collect our own full-replay datasets to ensure that this is the case. We arbitrarily choose the hyperparameters, simply requiring that the agent spends sufficient time in all three regions of performance. The training curves

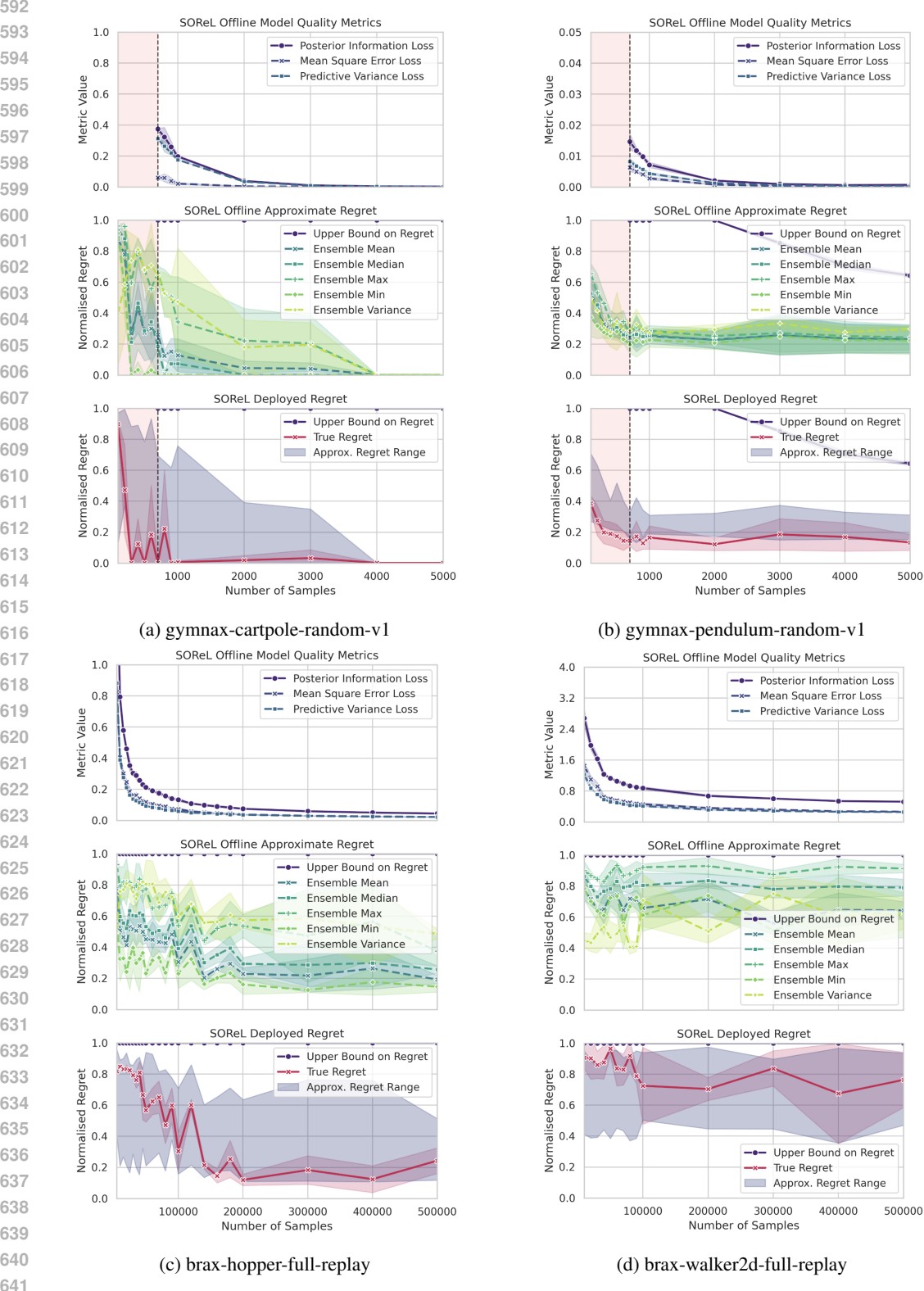

(a) gymnax-cartpole-random-v1

(b) gymnax-pendulum-random-v1

(c) brax-hopper-full-replay

(d) brax-walker2d-full-replay

Figure 29: Simplified version of SOReL applied to various tasks. Shaded purple shows the approximate regret range across all the metrics (with varying degrees of conservatism). For the gymnax environments, $N < 700$ is shaded red because the PIL is undefined in this region (validation set < batch size): without being able to ensure $\mathcal{E}(\mathcal{D}_N, \mathcal{M}^\star) <\approx \mathcal{V}(\mathcal{D}_N)$ the practitioner would have been unable to determine whether to trust the approximate regret. Mean and standard deviation given over 3 seeds.

obtained while collecting the offline datasets are given in Figure 30. The gymnax environments are simple enough that collecting a dataset using an ensemble of randomly initialised policies leads to sufficient coverage across all three regions of performance.

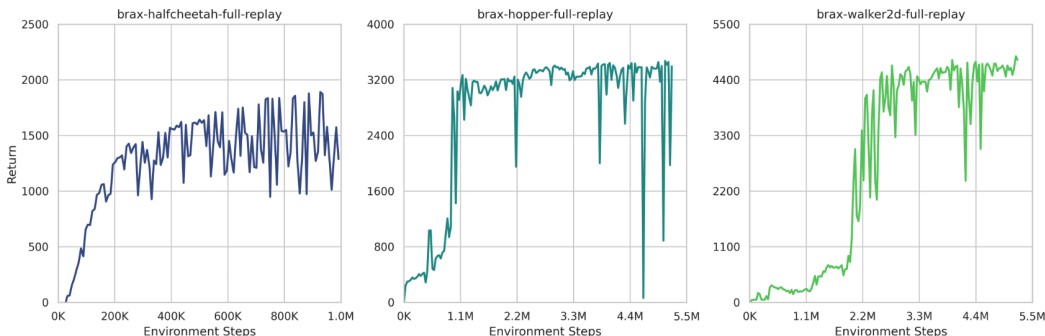

Figure 30: Training curves while collecting the brax full-replay offline datasets. We ensure that the agent spend sufficient time in poor, medium and expert regions of performance such that the offline dataset captures diverse transitions.

### F.2 WORLD MODEL AND RP APPROXIMATE INFERENCE

To ensure compatibility with Unifloral implementations (Jackson et al., 2025), our world model is a variation of the Gaussian World Model presented in 4.4, but amended to predict the change in state $\Delta := s' - s$ rather than the absolute next state $s'$. We also allow the model to characterise its uncertainty with variance functions $\sigma_{r,\theta}^2(s, a)$ and $\sigma_{\Delta,\theta}^2(s, a)$. The Gaussian reward and state transition models then have the form:

$$P_R(s, a, \theta) = \mathcal{N}(r_\theta(s, a), \sigma_{r,\theta}^2(s, a)), \quad P_\Delta(s, a, \theta) = \mathcal{N}(\Delta_\theta(s, a), \sigma_{\Delta,\theta}^2(s, a)),$$

with mean reward function $r_\theta(s, a)$ and mean state transition function $\Delta_\theta(s, a)$, as before. Let $r(s, a, \mathcal{D}_N) := \mathbb{E}_{\theta \sim P_\Theta(\mathcal{D}_N)}[r_\theta(s, a)]$ and $\Delta(s, a, \mathcal{D}_N) := \mathbb{E}_{\theta \sim P_\Theta(\mathcal{D}_N)}[\Delta_\theta(s, a)]$ denote the Bayesian mean reward and state transition functions and $r^\star(s, a)$ and $\Delta^\star(s, a)$ denote the true mean reward and state transition functions. We define the normalised mean squared error between the true and Bayesian mean functions as:

$$\mathcal{E}(\mathcal{D}_N, \mathcal{M}^\star) := \mathbb{E}_{(s,a) \sim \rho_\pi^\star} \left[ \frac{\|r(s, a, \mathcal{D}_N) - r^\star(s, a)\|_2^2}{2\sigma_r^2(\mathcal{D}_N)} + \frac{\|\Delta(s, a, \mathcal{D}_N) - \Delta^\star(s, a)\|_2^2}{2\sigma_\Delta^2(\mathcal{D}_N)} \right],$$

and the normalised predictive variance using the law of total variance as:

$$\mathcal{V}(\mathcal{D}_N) := \mathbb{E}_{(s,a) \sim \rho_\pi^\star} \left[ \mathbb{E}_{\theta \sim P_\Theta(\mathcal{D}_N)} \left[ \frac{\|r(s, a, \mathcal{D}_N) - r_\theta(s, a)\|_2^2}{2\sigma_r^2(\mathcal{D}_N)} + \|\sigma_{r,\theta}^2(s, a)\|_2^2 \right. \right.$$
$$\left. \left. + \frac{\|\Delta(s, a, \mathcal{D}_N) - \Delta_\theta(s, a)\|_2^2}{2\sigma_\Delta^2(\mathcal{D}_N)} + \|\sigma_{\Delta,\theta}^2(s, a)\|_2^2 \right] \right],$$

where $\sigma_r^2(\mathcal{D}_N)$ and $\sigma_\Delta^2(\mathcal{D}_N)$ denote the variance of the reward and change in state over the offline dataset.

When rolling out sequences of trajectories on which to train our (Bayes-Optimal) policy, we uniformly sample a model from the ensemble of elite models and then sample the transition from the corresponding Gaussian output distribution.

Our ensemble consists of multilayer perceptrons (MLPs) with ReLU activation, which we train using negative log-likelihood loss derived in Section C.1. Training the models in parallel allows us to simultaneously optimise maximum and minimum (log) variance parameters for each dimension across the model ensemble, which we use to soft-clamp the (log) variances output by the individual models. This prevents any individual model becoming overly confident or too uncertain in one dimension. All models in our ensemble have identical structure, but are initialised differently using LeCun

LeCun et al. (1998) initialisation. The maximum and minimum log-variance terms are initialised at constants. The exact loss function and ensemble dynamics model are the same as the one implemented by Jackson et al. (2025), but we use an Adam optimiser Kingma and Ba (2014) with cosine learning rate schedule rather than constant learning rate. A percentage of the available offline dataset is used as a validation set to calculate the PIL. At the end of training, only a subset of elite models are retained, based on their validation MSE. Although the current implementation uses hard-coded reset and termination conditions during model rollouts, the dynamics model could naturally be extended to learn reset and termination heads. When sampling transitions, we conservatively clip the rewards to remain within the support of the offline dataset distribution.

| Hyperparameter | gymnax-cartpole-random-v1 | gymnax-pendulum-random-v1 | brax-halfcheetah-full-replay | brax-hopper-full-replay | brax-walker2d-full-replay |
|---|---|---|---|---|---|
| Num. layers | 3 | 3 | 3 | 3 | 3 |
| Layer Size | 200 | 200 | 200 | 200 | 200 |
| Activation | ReLU | ReLU | ReLU | ReLU | ReLU |
| Num. Ensemble Models | 7 | 7 | 7 | 10 | 10 |
| Num. Elite Models | 5 | 5 | 5 | 8 | 8 |
| Log Var. Diff. Coeff. | 0.01 | 0.01 | 0.01 | 0.01 | 0.01 |
| Batch Size | 64 | 64 | 256 | 256 | 256 |
| Num. Epochs | 400 | 400 | 400 | 400 | 400 |
| Learning Rate | 0.001 | 0.001 | 0.001 | 0.001 | 0.001 |
| Learning Rate Schedule | cosine | cosine | cosine | cosine | cosine |
| Final Learning Rate % | 10 | 10 | 10 | 10 | 10 |
| Weight Decay | 2.5e-05 | 2.5e-05 | 2.5e-05 | 2.5e-05 | 2.5e-05 |
| Validation Split | 0.1 | 0.1 | 0.1 | 0.1 | 0.1 |

Table 6: World model ensemble dynamics hyperparameters.

### F.3 BAMDP SOLVER

We use the RNN-PPO implementation of Lu et al. (2022a), which we amend to be compatible with continuous action spaces. We sweep over the hyperparameters given in Table 7.

| Hyperparameter | Value / Sweep Values |
|---|---|
| Learning rate | [0.0001, 0.0003] |
| Anneal learning rate | True |
| Number of environments | [4, 64, 128, 256, 512] |
| Steps per environment | [32, 64, 128] |
| Total timesteps | Set to 500,000, 1,000,000 or 50,000,000 |
| Update epochs | [2, 4, 8] |
| Number of minibatches | [2, 4, 8, 16] |
| Discount factor ($\gamma$) | [0.99, 0.995, 0.998] |
| GAE lambda | [0.8, 0.9, 0.95] |
| Clip $\epsilon$ | [0.2, 0.3] |
| Entropy coefficient | [0.000, 0.001, 0.010] |
| Value function coefficient | 0.5 |
| Max gradient norm | [0.5, 1.0] |
| Layer Size | 256 |
| Activation function | tanh |
| RNN size | Set to 64, 128 or 256 |
| Burn-in Percentage | 25 |

Table 7: RNN-PPO hyperparameters swept over.

| Hyperparameter | gymnax-cartpole-random-v1 | gymnax-pendulum-random-v1 | brax-halfcheetah-full-replay | brax-hopper-full-replay | brax-walker2d-full-replay |
|---|---|---|---|---|---|
| Learning rate | 0.0003 | 0.0003 | 0.0003 | 0.0003 | 0.0003 |
| Anneal learning rate | True | True | True | True | True |
| Number of environments | 4 | 128 | 512 | 512 | 512 |
| Steps per environment | 128 | 64 | 64 | 32 | 64 |
| Total timesteps | 500,000 | 1,000,000 | 50,000,000 | 50,000,000 | 50,000,000 |
| Update epochs | 4 | 8 | 8 | 2 | 4 |
| Number of minibatches | 4 | 16 | 16 | 8 | 8 |
| Gamma | 0.99 | 0.99 | 0.99 | 0.998 | 0.995 |
| GAE lambda | 0.95 | 0.95 | 0.95 | 0.8 | 0.95 |
| Clip $\epsilon$ | 0.2 | 0.2 | 0.2 | 0.3 | 0.2 |
| Entropy coefficient | 0.01 | 0.003 | 0.003 | 0.001 | 0.001 |
| Value function coefficient | 0.5 | 0.5 | 0.5 | 0.5 | 0.5 |
| Max gradient norm | 0.5 | 0.5 | 0.5 | 1.0 | 0.5 |
| Layer Size | 256 | 256 | 256 | 256 | 256 |
| Activation function | tanh | tanh | tanh | tanh | tanh |
| RNN size | 64 | 128 | 256 | 256 | 256 |
| Burn-in Percentage | 25 | 25 | 25 | 25 | 25 |

Table 8: RNN-PPO hyperparameters for gymnax and brax environments. For computational reasons, we sweep over the hyperparameters of each task once, for a fixed dataset size (1000 datapoints for the gymnax tasks and 200,000 datapoints for brax tasks) and choose the hyperparameters corresponding to the lowest approximate regret, which we then use to train the policy of all other dataset sizes. Ideally, we would sweep over all hyperparameters for each dataset size.

## F.4 ORL IMPLEMENTATIONS

We use Jackson et al. (2025)'s implementations of the ORL algorithms. We use their default hyperparameters and sweep over their suggested hyperparameters, which we summarise in Table 9.

| | Hyperparameter | Value / Sweep Values |
|---|---|---|
| **Generic Optimisation** | Discount factor $\gamma$ | 0.99 |
| | Polyak averaging coefficient | 0.005 |
| **IQL** | Learning rate | 0.0003 |
| | Batch size | 256 |
| | Beta | [0.5, 3.0, 10.0] |
| | $\tau$ (expectile) | [0.5, 0.7, 0.9] |
| | Advantage clip | 100.0 |
| **ReBRAC** | Learning rate | 0.001 |
| | Batch size | 1024 |
| | Critic BC coefficient | [0, 0.0001, 0.0005, 0.001, 0.005, 0.01, 0.1] |
| | Actor BC coefficient | [0.0005, 0.001, 0.002, 0.003, 0.03, 0.1, 0.3, 1.0] |
| | Critic layer norm | true |
| | Actor layer norm | false |
| | Observation normalization | false |
| | Noise clip | 0.5 |
| | Policy noise | 0.2 |
| | Num critic updates per step | 2 |
| **MOPO** | Learning rate | 0.0001 |
| | Batch size | 256 |
| | Model retain epochs | 5 |
| | Number of critics | 10 |
| | Rollout batch size | 50000 |
| | Rollout interval | 1000 |
| | Rollout length | [1, 3, 5] |
| | Dataset sample ratio | 0.05 |
| | Step penalty coefficient | [1.0, 5.0] |
| **MOReL** | Learning rate | 0.0001 |
| | Batch size | 256 |
| | Model retain epochs | 5 |
| | Number of critics | 10 |
| | Rollout batch size | 50000 |
| | Rollout interval | 1000 |
| | Rollout length | 5 |
| | Dataset sample ratio | 0.01 |
| | Threshold coefficient | [0, 5, 10, 15, 20, 25] |
| | Termination penalty offset | [-30, -50, -100, -200] |

Table 9: Hyperparameters and sweep ranges for IQL, ReBRAC, MOPO, and MOReL.

For the sample efficiency experiments in Fig. 3, we use the default UCB bandit-based hyperparameter-tuning algorithm hyperparameters. We rescale the regret into a score to be maximised, normalised between 0 and 100 $(100 \cdot (1 - \text{regret}))$, enabling fair comparison and compatibility with the algorithm.

## F.5 REGRET NORMALISATION

We normalise the regret using the (known) minimum and maximum returns ($R_{min}$ and $R_{max}$) that an online-learnt policy would achieve in the true environment. In the absence of access to these values, we suggest estimating $R_{max}$ or $R_{min}$ as $\frac{r_{min}}{1-\gamma}$ and $\frac{r_{max}}{1-\gamma}$ respectively, where $r_{min}$ and $r_{max}$ denote the 2.5th and 97.5th percentiles of episode rewards in the offline dataset. These thresholds are

suggested to avoid unrealistic assumptions, such as that the best possible policy consistently receives the 100th percentile reward at every time step. Such unrealistic assumptions may significantly distort results, especially in reward distributions with heavy tails where an inflated $R_{max} - R_{min}$ would compress the true and expected regrets.

| | gymnax-cart pole-random-v1 | gymnax-pend ulum-random-v1 | brax-half cheetah-full-replay | brax-hop per-full-replay | brax-walker 2d full-replay | d4rl-halfcheetah medium-expert-v2 | d4rl-hopper med ium-v2 | d4rl-walker2d medium-replay-v2 |
|---|---|---|---|---|---|---|---|---|
| $r_{\min}$ | 0 | -13.3 | -0.50 | 0.00 | 0.00 | -0.28 | -0.20 | 0.00 |
| $r_{\max}$ | 1 | -0.2 | 3.50 | 3.50 | 3.50 | 12.14 | 3.23 | 4.59 |
| $P_{2.5}$ | 1 | -13.2 | -0.84 | 1.08 | 0.01 | 1.99 | 1.25 | -0.19 |
| $P_{97.5}$ | 1 | -0.16 | 3.44 | 4.34 | 5.65 | 12.39 | 4.73 | 4.92 |

Table 10: Top half of table: for the gymnax and brax tasks, we define $r_{min}$ and $r_{max}$ using approximate known minimum and maximum returns Jesson et al. (2024), and dividing by the episode length. For the D4RL tasks, we divide the given D4RL minimum and maximum reference scores by the episode length to find $r_{min}$ and $r_{max}$. Bottom half of table: the suggested normalisation values if expert and random scores were unknown, and determined using the $95^{th}$ percentile of the offline dataset. Based on the above datasets, for all datasets apart from cartpole-v1, normalisation using the $95^{th}$ percentile approximation would lead to a more conservative approximate regret. We note that as long as we use the same normalisation constants for both the approximate and true regrets, the absolute value of the normalisation constants is arbitrary. cartpole-v1 is an exception, as the reward is constant for each step (episode returns vary only due to early termination).

We calculate the infinite horizon discounted return from the finite horizon discounted return as follows:

$$R_{inf} = R_{fin} \cdot \left(1 + \frac{\gamma^s}{1 - \gamma^s}\right),$$

where $s$ represents the maximum number of episode steps. The normalised regret is then calculated from the infinite horizon discounted return using:

$$\text{Regret} = \frac{R_{max} - R_{inf}}{R_{max} - R_{min}}.$$

F.6  EXPERIMENT COMPUTE RESOURCES

All of our experiments were run within a week using four L40S GPUs.

