# OpenReview forum: "SOReL and TOReL: Two Methods for Fully Offline Reinforcement Learning"
_ICLR.cc/2026/Conference — Submitted to ICLR 2026_

### Official Review · Reviewer_ApZA · 2025-10-29

**Soundness:** 2
**Presentation:** 2
**Contribution:** 2
**Rating:** 2
**Confidence:** 3

**Summary:**

Current offline RL methods rely on extensive online interactions for hyperparameter tuning, and have no reliable bound on their initial online performance. To address these two issues, the authors introduce two algorithms. Firstly, SOReL: an algorithm for safe offline reinforcement learning. Using only offline data their Bayesian approach infers a posterior over environment dynamics to obtain a reliable estimate of the online performance via the posterior predictive uncertainty. Crucially, all hyperparameters are also tuned fully offline. Secondly, they introduce TOReL: a tuning for offline reinforcement learning algorithm that extends their information rate based offline hyperparameter tuning methods to general offline RL approaches. Lastly, they conduct empirical evaluations.

**Strengths:**

1. The problem of fully offline RL is interesting and important.
2. The authors propose to solve the problem through a Bayesian view, which is an interesting idea.
3. There are empirical evidence supporting the theoretical findings.

**Weaknesses:**

1. My main concern is about Theorem 2, since the last equation in line 274 is not correct. As N becomes sufficiently large, LHS is a constant while RHS converges to 0. If this is a typo, please provide the correct form.

2. Discussions about Theorem 2 and its assumption is missing in the main part. Therefore it is hard to understand the implication of the Theorem. Some high-level discussions about the assumption in the main part would be helpful.

3. The algorithms tune the hyperparameters fully offline, while it is unclear whether the approach is efficient. The PIL function includes multiple expectations and the posterior distribution, which seems hard to optimize directly. Are the optimization problems in both algorithms efficient in general?

**Questions:**

Please refer to the weakness

---

> ### Author Response · Authors · 2025-11-17
>
> We thank the reviewer for their comments. We address their concerns and questions below:
>
> ### Equation Line 274:
> The reviewer is right to point out the typo in the inequality on line 274. This should read: $$\le 2\mathcal{R}_{\max}\cdot\sqrt{1-\exp\left(-\frac{Cd}{(1-\gamma)N}\right)}$$
> as derived on line 1699 of the proof in the Appendix. Many thanks for spotting this.
>
> ### Discussions about assumptions:
> Given the extra page limit for the camera-ready version, we can move the discussion from the Appendix to the main text.
>
> ### Efficiency:
>  TOReL and SOReL adapt the efficient ensembling methods commonplace in offline RL for posterior marginalisation, making the algorithms no less tractable than methods such as MOReL and MOPO. We make this point clearly on line 233: `a Bayesian approach... scales as well as existing
> state-of-the-art offline RL methods (Jackson et al., 2025) which also rely on ensembling methods for uncertainty quantification.' Like these methods, posterior uncertainty is characterized through ensemble variance and evaluating posterior expectations involves drawing ensemble members uniformly from the set of ensembles.

---

### Official Review · Reviewer_sfCQ · 2025-10-30

**Soundness:** 3
**Presentation:** 2
**Contribution:** 2
**Rating:** 4
**Confidence:** 4

**Summary:**

The paper introduces SOReL and TOReL, two new algorithms that enable safe and reliable offline reinforcement learning by providing offline hyperparameter tuning and regret estimation, a problem so far underexplored in prior literature, thereby improving sample efficiency and performance guarantees without needing online interactions and moving offline RL closer to practical applicability.

**Strengths:**

- the theoretical considerations & derivation of the algorithms, leveraging PIL to select hyperparameters for the model appear novel and significant
- the method appears to be able to well approximate the true regret, enabling fully offline hyperparameter tuning and thus bringing offline RL closer to real world applicability
- the method is very flexible and can be used in principle with any existing offline RL algorithm of a users choice
- the authors demonstrate accurate regret estimation on a variety of tasks / datasets

**Weaknesses:**

While I find the proposed method highly appealing in many ways, I find the paper has a couple of major weaknesses:

1) A key contribution appears to be the PIL-based tuning of the model hyperparameters. I would argue however, that this is the "easy" part of offline RL hyperparameter tuning, since we can resort to simple supervised learning techniques like holding out 10% of the training data & measuring prediction performance on this set so select hyperparameters. I would argue the proposed method is much more sophisticated (directly tuning for regret minimization instead of dynamics prediction performance), and works likely better - but since this is a large part of the paper's contribution, I'd argue we still need a comparison against such a simple baseline.

2) Apart from the PIL-based model parameter tuning, this kind of approach (bayesian model + sampling from posterior to make policy e.g. safe) has been studied in many prior works, such as [1,2] (consider adding in related work). No need for a direct comparison in experiments, but I'd argue the novelty in that sense it at least somewhat limited.

3) Probably the biggest one: I would agree that so far no method has really solved the offline hyperparameter tuning / selection problem & your work towards achieving this appears highly promising, but I would not agree to your assessment in the related work section, that your method is the first to "carry out all hyperparameter tuning [...] using only offline data". Other methods have been developed for fully offline policy evaluation / hyperparameter selection (see e.g. the baselines in [3] and all the derivatives that were developed). While you compare your algorithm's performance against an online tuning algorithm and an oracle, the truly interesting thing to understand (at least to me it seems that way), would be whether the proposed method performs better than other offline tuning / selection methods - without these experiments, I believe the key merit of your method is not empirically validated.

[1] Depeweg, Stefan, et al. "Learning and policy search in stochastic dynamical systems with bayesian neural networks." ICLR 2017
[2] Kaiser, Markus, et al. "Bayesian decomposition of multi-modal dynamical systems for reinforcement learning." Neurocomputing 416 (2020): 352-359.
[3] Fu, Justin, et al. "Benchmarks for deep off-policy evaluation." ICLR (2021).

**Questions:**

- please clarify if I am mistaken: PIL is only used for model hyperparameter tuning - afterwards, to tune policy, we simply use the final selected model to quantify regret (via median return), correct?
- if I am not mistaken, you refer to PIL with many different terms (posterior inforamtion loss, posterior information distance, information rate, ...) - this is a bit confusing, could you elaborate / align?

---

> ### Author Response · Authors · 2025-11-17
>
> We thank the reviewer for their comments and appreciate the reviewer recognises our contribution to solving the offline hyperparameter tuning and regret estimation problems. We identify and respond to the three points raised by the reviewer.
>
> ### The PIL:
> The reviewer suggests that our contribution for PIL-based tuning is the same as a simple supervised learning technique like holding out training data and measuring predictive performance on that set. This is not true and is a mischaracterisation of our work. One of our key contributions is a detailed theoretical analysis which finds that the PIL controls the regret for the Bayesian offline RL problem setting, not just using it for tuning as the reviewer suggests. This provides a formal justification for using it to monitor regret rate during offline training and the derivation is non-trivial (see Appendix for proof). Making this connection is an entirely novel stand-alone contribution. We then derive the PIL for the specific Gaussian setting, which we find reduces to a mean squared error (MSE) term *and* a predictive variance term. The `easy' part that the reviewer is claiming our method is corresponds to estimating the MSE term. But, importantly, this does *not* estimate the PIL as we *also* need to evaluate the predictive variance (which is unique to our Bayesian approach). We discuss this distinction clearly in Section 4.4. Moreover, ensuring that the predictive variance and MSE terms in the PIL decay in unison is a core message and contribution of our paper (see Section 5.1, Algorithm 1 and Algorithm 2, where we discuss this).
>
> ### Bayesian model and sampling from the posterior:
> We assume by Bayesian model and sampling from the posterior, the reviewer means solving a model-based offline Bayesian RL problem (i.e. solving a BAMDP using offline data), which is the problem setting we study. The referenced papers have little relevance to our setting as they focus on online RL and do not solve the Bayesian RL problem: states are not augmented with beliefs and policies are static - this is clear from the fact that policies only condition on recent state $s_t$ and have no capacity for adaptive belief updating, hence cannot solve a BAMDP. Regardless, we do not claim novelty in formulating the offline model-based Bayesian RL problem (see related work where we discuss other approaches that use the same formulation). What we do claim novelty for is providing a regret analysis of the Bayesian offline RL problem and providing a way to estimate regret for the (approximately) Bayes-optimal policy via SOReL.
>
> ### Offline policy Evaluation:
> The paper referenced by the reviewer  benchmarks offline policy evaluation (OPE) and the baselines they refer to are all OPE methods. We disagree with the reviewer's claim that these methods carry out fully offline hyperparameter selection as none of them carry out offline tuning for the hyperparameters associated with the policy planner. Offline planning is a key part of solving the offline RL problem. These methods are designed to estimate the value of a given (fixed) target policy from logged data and, with the exception of the model-based method by Zhang et al. [4] (MB using notation from [3]), do not account for the planning problem needed to learn optimal policies. In addition, many tune other parts of their parameter set online (the Variational Power Method (VPM) for example needs online interactions for learning rates and regularisation weights).  The model-based OPE method MB does account for planning but needs online interactions with Cartpole to tune the policy optimisation planner hyperparmaters: 'hyperparameter tuning for the planning process is conducted on the “cartpole swingup” task. The hyperparameters
> used in the planning process are $M = 3, N = 16, H = 10, \beta = 0.1,$ and $\sigma = 0.01$'. In comparison, the hyperparameters in SOReL/TOReL's policy planners are tuned offline.
>
>  We test using Unifloral's (Jackson et al. 2025) rigorous benchmarking protocol, which is designed to expose prior methods that frequently employ extensive, undocumented online evaluation for hyperparameter tuning as this complicates method comparisons. Key to this is a transparent evaluation protocol that explicitly quantifies online tuning budgets. TOReL is designed for zero-shot deployment, and so must be strictly evaluated with no online interaction. Comparing to MB (which is the only feasible approach as it accounts for policy optimisation) is not a sound benchmark as it requires online samples. As MB is not designed for zero-shot deployment in the offline RL setting, making this comparison would only serve to add to the opaqueness that Unifloral seeks to resolve.
>
> [4] Zhang et al. 2021, Autoregressive Dynamics Models for Offline Policy Evaluation and Optimization

---

> ### Author Response · Authors · 2025-11-17
> **Response to Questions**
>
> ### Q1:
> The PIL is used to tune both the model architecture and the hyper-parameters of the approximate posterior. As described in lines 326–330, our goal is to minimise the PIL while ensuring that the mean squared error (MSE) matches the predicted variance of the ensemble. We want to avoid the MSE from collapsing below the ensemble variance, which would indicate model overfitting. In other words, we minimise the MSE while maintaining the ensemble variance at a similar magnitude, so that the model uncertainty can be accurately quantified and the regret can be approximated (lines 335–337).
> Once the posterior hyper-parameters are fixed, we train the policy. The policy hyper-parameters are tuned using the approximate regret derived from the fixed posterior. Concretely, this means that each policy is evaluated on all members of the world-model ensemble, and the hyperparameter configuration that achieves the highest median return across the ensemble is selected (lines 352-355).
>
> ### Q2:
> Posterior information loss (PIL) (defined on line 247) is the distance, and information rate (defined on line 258) is the rate at which the PIL decays. The posterior information distance in Theorem 1 is a typo, it should read posterior information loss. Thank you for spotting that.

---

### Official Review · Reviewer_iime · 2025-10-31

**Soundness:** 3
**Presentation:** 3
**Contribution:** 3
**Rating:** 6
**Confidence:** 3

**Summary:**

This paper addresses two critical issues in offline reinforcement learning that have largely been overlooked: the lack of offline hyperparameter tuning metrics, and the absence of reliable offline regret approximation methods . The authors propose a Bayesian framework for offline RL, leveraging the Posterior Information Loss (PIL), the expected KL divergence between a learned model and true environment dynamics as an offline tuning signal. From this framework, they introduce two methods: SOReL (Safe Offline RL), which provides reliable offline regret estimates using predictive uncertainty to enable safe deployment, and TOReL (Tuning for Offline RL), which extends SOReL's offline hyperparameter tuning methodology to general model-based and model-free offline RL algorithms. The paper provides rigorous theoretical analysis showing that regret is bounded by the PIL (Theorem 1) and that Bayesian offline RL achieves the optimal convergence rate (Theorem 2). Empirical evaluations on D4RL, Adroit, and proprietary Brax datasets demonstrate that TOReL identifies near-oracle hyperparameters using only offline data, saving 20K to >200K online samples compared to existing online hyperparameter tuning methods, while SOReL accurately approximates true regret across diverse offline datasets.

**Strengths:**

**Well-motivated problem formulation.** The paper identifies and articulates two concrete, practically important issues in offline RL that have been underexplored: the reliance on online interactions for hyperparameter tuning and the lack of performance guarantees before deployment. The motivating examples (healthcare, robotics) and the cycle diagram (Figure 1) effectively communicate why these issues matter for real-world applications. This clear problem framing strengthens the contribution's significance.

**Rigorous theoretical analysis with frequentist justification.** Theorems 1 and 2 provide formal regret bounds in terms of PIL, with Theorem 2 establishing the optimal convergence rate for Bayesian offline RL under local asymptotic normality assumptions. The frequentist justification for a Bayesian approach is valuable and non-obvious. Proposition 1  decomposes PIL for Gaussian world models into interpretable terms (MSE and predictive variance), facilitating practical implementation and understanding.

**Comprehensive empirical validation.** The paper includes extensive experiments across multiple environments (gymnax, Brax, D4RL) with careful ablations and statistical testing (e.g., Pearson correlation in Table 3). The comparison against online UCB-based hyperparameter tuning (Figure 3) provides compelling evidence of sample efficiency gains. Both SOReL and TOReL experiments demonstrate practical effectiveness, with particular strength in the near-oracle performance of ReBRAC+TOReL across diverse datasets (Table 1).

**Weaknesses:**

**Gap between theory and practice.** The theoretical regret bound (Theorem 1) is noted to be "too conservative" and is not used in practice; instead, the posterior predictive median (Eq. 5) is employed as a heuristic. This disconnect undermines the theoretical contribution's direct utility. The paper acknowledges that the bound's tightness depends critically on model accuracy relative to discount factor
\gamma, a constraint difficult to satisfy in practice (e.g., only pendulum-v1 yields non-trivial bounds in Figure 10b).

**Missing comparisons with related work.** The paper does not compare against other offline hyperparameter tuning methods beyond citing them (e.g., Wang et al. 2022, Paine et al. 2020, or any related work of the same nature). Direct empirical comparisons would strengthen the work. There are some numerical experiments, but visual trends of training/eval curves would be helpful.

**Questions:**

**Q1:** Can the theoretical bound (Theorem 1) be tightened? Given that the bound is often too conservative in practice, are there domain-specific refinements or tighter bounds for structured environments that could improve practical utility?

**Q2:** Why use the posterior predictive median rather than other quantiles (e.g., mean, upper confidence bound)? The choice of median is justified as a "compromise" but lacks formal justification. Have you tested other regret approximation metrics?

---

> ### Author Response · Authors · 2025-11-17
>
> We thank the reviewer for their comments and for recognising our theoretical contributions and extensive empirical validation. We address their concerns below:
>
> ### Utility of Analysis.
> The reviewer claims that the theoretical contribution is undermined by its utility. This is not correct for two reasons. Firstly, our first result (Theorem 1) proves that regret is controlled by the PIL. Making this connection yields a powerful method for hyperparameter tuning that we use in practice (see Section 4.4, Section 5.1, PIL monitoring, Algorithm 1 and Algorithm 2). PIL monitoring is an essential component of our method that arises from our theoretical contribution.  Secondly, the purpose of the regret analysis in Theorem 2 is to determine the worst-case regret rate of an algorithm — the bound holds uniformly over all environments that satisfy the assumptions. Because of this, regret bounds provide a theoretical tool to compare sample efficiency of different algorithms. Algorithms that achieve $\mathcal{O}(\frac{1}{\sqrt{N}})$ are asymptotically rate-equivalent. Our second result (Theorem 2) proves that offline Bayes-optimal policies achieve the $\mathcal{O}(\frac{1}{\sqrt{N}})$ regret rate achieved by approaches such as MOPO and MOReL under the usual assumptions. From a frequentist perspective, this is an essential result to prove for any offline RL method (hence why they are ubiquitous in the literature) with significant practical implications. If Bayesian methods could be shown to be lower bounded by a slower rate under the same assumptions (which our analysis rules out), this would be a good reason to disregard them practically in favour of existing methods that achieve faster worse-case rates. Finally, we remark our analysis is not just for one algorithm, but is a more general analysis of Bayesian offline RL as a whole. This is an important result that has been missing from the literature, making our theoretical result significant stand-alone contributions.
>
> ### Missing Comparisons.
> Our algorithms are designed for zero-shot offline RL and so must be strictly evaluated with no prior online interaction. i.e. no interactions with the environment are allowed during training or hyper-parameter tuning. None of the methods proposed by the reviewer are suitable for this. As we discuss in related work, Wang et al. 2022 is a method for offline to online RL - it does not learn a deployable policy offline - and Paine et al. 2020 requires online samples to tune its OPE method. The authors even state this in their conclusion: `An important remaining challenge is how to choose hyperparameters for FQE. Although we do show that it is relatively robust to one (number of learner steps), in general this is an open problem.'
>
> ### Q1. Tighter Bounds.
> We assume that the reviewer means the bound in Theorem 2. Our bound is statistically tight as no algorithm can guarantee a smaller asymptotic regret rate under the same assumptions (Agarwal et al. 2022). By definition, regret bounds inherently consider the worst case scenario, and thus are always conservative by construction. As discussed in Section 5 and above, this makes them powerful tools for comparing the sample efficiency of algorithms, but they are not designed to be used to estimate the actual regret. This motivates the development of approximate regret metrics, which do estimate the actual regret rather than upper bounding it to determine asymptotic rates across all MDPs.
>
> ### Q2. Choice of Median:
> The reviewer's claim that our choice of predictive median is not formally justified and that other regret approximation metrics have not been tested is incorrect. We provide a derivation of 5 regret estimation metrics in Section C.2 (including the median). We clearly reference this in Section 5.2:  'full details and an overview of alternative metrics that can be derived from the posterior to approximate the regret with varying degrees of conservatism are found in Section C.2'. Moreover, all 5 metrics are evaluated across 5 domains (see Fig. 9 and Figs. 11 (a)-(d)). Again, we clearly reference this in Section 6.2: 'More details on a non-trivial upper-bound, along with results for gymnax and the remaining brax environments and ablations of different ensemble metrics that can be used to approximate regret with varying degrees of conservatism are found in Section E' (Figures 9 and 11). As we state, a practitioner using this algorithm is free to select the regret approximation that best suits their application’s safety-criticality, such as opting for a more conservative approximate in high-risk domains.

---

> > ### Comment · Reviewer_iime · 2025-11-21
> >
> > Thank you very much for taking the time to provide a thorough response. I appreciate the effort you placed in addressing weaknesses and questions and would like to apologize for any mistakes or misunderstandings regarding the question on choice of median.
> >
> > However, given my assessment on the entire paper, my score will remain unchanged at this time.

---

### Official Review · Reviewer_CiFA · 2025-10-31

**Soundness:** 2
**Presentation:** 3
**Contribution:** 3
**Rating:** 4
**Confidence:** 4

**Summary:**

This paper considers two salient problems with offline RL: (a) what offline metrics are useful for tuning hyper-parameters of an offline RL policy search procedure, (b) how to estimate the online regret associated with deploying a policy outputted by an offline RL procedure? Towards this, the paper utilizes Bayesian perspective to consider the use of posterior information loss (PIL) for purposes of solving the aforementioned problems, developing TOReL (for tuning) and SOReL (for estimating regret).

**Strengths:**

- The paper addresses important problems that plague the design of offline RL algorithms and these issues limit the applicability of offline RL in general.
- The empirical results appear pretty compelling as well.

**Weaknesses:**

- The paper can do a better job with presenting connections with lowerbounds in offline RL connecting the regret estimation against hardness of policy evaluation; for instance, see [1], but there are probably other works that build on this. Why does the proposed procedure actually work in light of some of these stark negative results?

[1] Wang et al: What are the Statistical Limits of Offline RL with Linear Function Approximation? 2020.

**Questions:**

- Regarding estimators proposed by this paper: can you mention again how one can rely on point estimates of these quantities and whether they can be strengthened using estimation procedures building on [2]?

[2] Agarwal et al: Deep Reinforcement Learning at the Edge of the Statistical Precipice, 2021.

---

> ### Author Response · Authors · 2025-11-17
> **Rebuttal to Reviewer CiFA**
>
> We thank the Reviewer for their comments. The first point raised is interesting, and we will include a discussion in an accepted paper. The lower bound analysis by Agarwal et al. is a worst-case analysis. It does not state that for all MDPs the sample complexity is exponential; rather, it states that there exists at least one MDP where the sample complexity is exponential. Crucially, the result says nothing about the average MDP, or the  proportion of MDPs where the sample complexity is exponential, which would be needed to rule out the possibility of a regret estimation method working in practice. Moreover, our method offers two methods of protection against pathological cases: as our method is Bayesian, it outputs predictive variance, which can be used to determine the certainty in the regret estimation (which we plot shaded on our graphs). As we discuss in Section 5, a second indicator of when regret estimates may be inaccurate is when $\mathcal{E}(\mathcal{D}_N,\mathcal{M}^\star)\ne \mathcal{V}(\mathcal{D}_N)$. This allows us to produce a shaded red region on our graphs where the approximate regret may be unreliable.
>
> We do not understand the reviewer’s claim that our method relies on point estimates—this is incorrect—and we do not see the direct relevance of [2] to our work, which focuses on evaluation metrics for reporting RL performance. Like all fully Bayesian approaches, we infer a posterior that characterises both epistemic and aleatoric uncertainty, rather than relying on a single point estimate. The posterior variance enables us to tune the hyper-parameters of the posterior by tracking the components of the PIL (MSE vs. predictive variance), ensuring that the posterior variance accurately reflects true uncertainty. This, in turn, guarantees that the approximate regret is a faithful proxy for the true regret, allowing us to use it to tune the policy hyper-parameters. Tuning using the predictive variance would not be possible if our method used only point estimates. See lines 294, 320, 326, and 353 for explanations, and the bottom of Figure 9 and Figures 11a–d for visualisations of the posterior predictive variance (the approximate regret range).

---

### Author Response · Authors · 2025-12-03
**Comment to AC I**

We are disappointed with the reviewing process for our submission to ICLR this year, and don't believe that the reviews reflect a meaningful assessment of our work.

### Factually Incorrect Claims about our Work

All reviewers mischaracterise our work, making statements that are not true. It would not be possible to make these claims if reviewers had carefully read the main body of the paper.  Reviewer CiFA incorrectly claims our method relies on point estimates. Our method relies on inferring an (approximate) posterior, and so by definition does not rely on point estimates. We explain this clearly in lines 294, 320, 326, and 353. Moreover, we explain this at the bottom of Figure 9 and Figures 11a–d where we provide visualisations of the posterior predictive variance (which would not be possible for a point estimate method). Reviewer iime has responded and recognised that they made incorrect claims about our work regarding the derivation and use of median and we appreciate this engagement.  Reviewer sfCQ makes incorrect claims about our method, claiming that we only use the MSE for model tuning. This is not true, as our method requires the MSE **and** predictive variance for model tuning. This is clear from a cursory read of the paper (See Section 4.4, Section 5.1, Algorithm 1 and 2). The related works they highlight have no relevance to ours as they do not not solve an offline RL problem, let alone a Bayesian offline RL problem. Moreover, their claim that other OPE based methods can be used for fully offline hyperparameter tuning is not correct - as we highlight in our response, none of the methods the reviewer proposed can be used for fully offline hyperparmater tuning (see also further experiments below). Finally Reviewer ApZA incorrectly claims our methods do not scale due to posterior expectations. As we responded in our rebuttal, we adapt the efficient ensembling methods commonplace in offline RL for posterior marginalisation, making the algorithms no less tractable than methods such as MOReL and MOPO. We make this point clearly on line 233.

### Lack of Engagement

Despite our timely rebuttal, only one reviewer (Reviewer iime) responded. Whilst we appreciate they admitted that we addressed their perceived weaknesses, and their criticisms were based on incorrect assumptions, they did not raise their score beyond a 6. No other reviewer has acknowledged our response, which we find appalling.

### Further Experiments

Reviewer sfCQ asked for a comparison to OPE methods as baselines. We re-iterate that this is not a like-for-like comparison as all of these methods **still require online tuning of part of their hyperparamater set**. The authors of FQE [1] even quote this: `Finding a good way to tune FQE hyperparameters remains an open problem for future research'. Despite this, we have implemented the suggested methods and we used the hyperparameter values provided by the authors. Results can be found in Appendix E.1 of the updated PDF and we will include them in the main body conditioned on acceptance given the additional space provided.
Our results show that TOReL consistently outperforms these OPE methods, despite not relying on the online hyperparameter tuning they require. This highlights the strength of our fully offline approach. We have included the absolute error and rank correlation (evaluation criteria recommended for Off-Policy Evaluation by [2]) as well as scatter plots comparing each regret metric against true regret for each D4RL dataset. On D4RL-halfcheetah-medium-expert-v2, D4RL-hopper-medium-v2, and D4RL-walker-medium-replay-v2, TOReL outperforms all baselines, both in identifying the lowest-regret policy and in the magnitude and statistical significance of its correlation with true regret. On D4RL-pen-expert-v1 and D4RL-hammer-expert-v1, all algorithms perform equally poorly, which we attribute to the nature (quality and coverage) of the datasets. We again reiterate that these comparisons are not like-for-like: for example, we could not evaluate the OPE baselines on the brax datasets because these datasets do not contain full trajectories, which are required for regret metrics in BC-based methods such as DR and IW, nor do they include initial-state flags, which are required for DICE. Finally, we use the hyperparameter settings provided in [2] for all comparison methods, which we reiterate include hyperparameter that have been tuned online. We will release our JAX re-implementations of these OPE-metrics with the RL community.


[1] Hyperparameter Selection for Offline Reinforcement Learning, Le Paine et al., NeurIPS 2020

[2] Benchmarks for Deep Off-Policy Evaluation, Fu et al., ICLR 2021

---

> ### Author Response · Authors · 2025-12-03
> **Comment to AC II**
>
> ### The only legitimate criticism raised is a typo.
>
> The only valid criticism of our work was related to a typo in Theorem 2 (as identified by Reviewer ApZA). We explicitly acknowledge this as a typo and have corrected it in the latest draft; we do not consider this sufficient grounds for rejecting the paper.
>
> In light of this, we respectfully ask the AC to re-assess the merit of our submission. All reviewers recognise the importance of our contribution— developing a method for fully offline hyperparameter tuning and accurate regret approximation constitutes a meaningful and impactful advance for the RL community. As the perceived weaknesses of our work are factually incorrect or have been thoroughly addressed, we maintain that there are no grounds for rejection. Our additional ablations clearly position our method as the strongest, and indeed **only**, approach for **fully offline hyperparameter tuning**, particularly given that it outperforms alternative OPE-based methods that **still require online samples to tune part of their hyperparameter set**.

---

### Meta-Review · Area_Chair_VJPP · 2026-01-05

**Summary:**

This work studies offline RL, and propose two approaches including a Bayesian approach for environment inferences and learning, and another algorithm for offline parameter finetuning.

**Reviewer Concerns:**

During the review process, the authors provided clarifications that largely mitigated the concerns of Reviewer sfCQ (regarding comparisons to prior 'not purely offline' works) and Reviewer iime (regarding the connection between theory and the median selection method).

However, the issues raised by Reviewer CiFA remain only partially resolved. Specifically, the tension between the paper's main results and the established lower bounds (or hard instances) has not been fully reconciled. The authors' rebuttal deferred the discussion of this disconnect, but a rigorous explanation of how these positive results coexist with known hardness results is essential for the paper's validity.

I believe this submission has merit, but it requires a revision (which is not provided during rebuttal) to explicitly discuss these theoretical comparisons and to more clearly highlight the contributions against the existing landscape. Consequently, the paper would benefit from a fresh review cycle after these updates.

**Reviewer Scores:**

I believe reviewers iime and sfCQ's concerns might be addressed, yet the questions of CiFA remain partially unsolved.

---

### Decision · Program_Chairs · 2026-01-26

Reject